# Adipose tissue retains an epigenetic memory of obesity after weight loss

Laura C. Hinte[1,2,12], Daniel Castellano-Castillo[1,8,12], Adhideb Ghosh[1,3,4], Kate Melrose[1,2], Emanuel Gasser[1], Falko Noé[1,3,4], Lucas Massier[5,9], Hua Dong[4,10], Wenfei Sun[4,11], Anne Hoffmann[6], Christian Wolfrum[4], Mikael Rydén[5], Niklas Mejhert[5], Matthias Blüher[6,7] & Ferdinand von Meyenn[1✉]

Reducing body weight to improve metabolic health and related comorbidities is a primary goal in treating obesity[1,2]. However, maintaining weight loss is a considerable challenge, especially as the body seems to retain an obesogenic memory that defends against body weight changes[3,4]. Overcoming this barrier for long-term treatment success is difficult because the molecular mechanisms underpinning this phenomenon remain largely unknown. Here, by using single-nucleus RNA sequencing, we show that both human and mouse adipose tissues retain cellular transcriptional changes after appreciable weight loss. Furthermore, we find persistent obesity-induced alterations in the epigenome of mouse adipocytes that negatively affect their function and response to metabolic stimuli. Mice carrying this obesogenic memory show accelerated rebound weight gain, and the epigenetic memory can explain future transcriptional deregulation in adipocytes in response to further high-fat diet feeding. In summary, our findings indicate the existence of an obesogenic memory, largely on the basis of stable epigenetic changes, in mouse adipocytes and probably other cell types. These changes seem to prime cells for pathological responses in an obesogenic environment, contributing to the problematic 'yo-yo' effect often seen with dieting. Targeting these changes in the future could improve long-term weight management and health outcomes.

Obesity and its related comorbidities represent substantial health risks[1]. A primary clinical objective in managing obesity is to achieve appreciable weight loss (WL), typically through rigorous dietary and lifestyle interventions, pharmaceutical treatments or bariatric surgery (BaS)[2]. Strategies relying on behavioural and dietary changes frequently only result in short-term WL and are susceptible to the 'yo-yo' effect, in which individuals regain weight over time[3,5,6]. This recurrent pattern may be partially attributable to an (obesogenic) metabolic memory that persists even after notable WL[4,7–10] or metabolic improvements[11–13]. Indeed, lasting phenotypic changes from previous metabolic states, that is, metabolic memory, have been reported in mouse adipose tissue (AT) or the stromal vascular fraction (SVF)[14–16], whereas in liver these were reversible[15–17]. Persistent alterations after WL in the immune compartment[18], and transcriptional and functional memory of obesity in endothelial cells of many organs[19–22], have also been reported.

Epigenetic mechanisms and modifications are essential for development, differentiation and identity maintenance of adipocytes in vitro and in vivo[23–27], but are also expected to be crucial contributors to the cellular memory of obesity[4,7]. For example, lasting chromatin accessibility changes have been associated with pathological memory of obesity in mouse myeloid cells[28] and, also, cold exposure studies have indicated the existence of (epigenetic) cellular memory[26,29]. Hitherto, most human studies have focused on DNA methylation analysis in bulk tissues or whole blood to assess putative cellular memory[30–33]. These reports might be confounded by variations in cell type composition, which are poorly characterized in the AT during WL, and therefore serve foremost as indicators of cellular epigenetic memory.

In summary, it remains unresolved whether individual cells retain a metabolic memory and whether it is conferred through epigenetic mechanisms. Here, we set out to address this by first performing single-nucleus RNA sequencing (snRNA-seq) of AT from individuals living with obesity before and after significant WL, as well as lean, obese and formerly obese mice, confirming the presence of retained transcriptional changes, and, second, by characterizing the epigenome of mouse adipocytes, which revealed the long-term persistence of an epigenetic obesogenic memory.

[1]Laboratory of Nutrition and Metabolic Epigenetics, Institute of Food, Nutrition and Health, Department of Health Sciences and Technology, ETH Zurich, Zurich, Switzerland. [2]Biomedicine Programme, Life Science Zurich Graduate School, Zurich, Switzerland. [3]Functional Genomics Center Zurich, ETH Zurich and University Zurich, Zurich, Switzerland. [4]Laboratory of Translational Nutrition Biology, Institute of Food, Nutrition and Health, Department of Health Sciences and Technology, ETH Zurich, Zurich, Switzerland. [5]Department of Medicine Huddinge, Karolinska Institutet, Karolinska University Hospital Huddinge, Stockholm, Sweden. [6]Helmholtz Institute for Metabolic, Obesity and Vascular Research (HI-MAG), Helmholtz Zentrum München, University of Leipzig and University Hospital Leipzig, Leipzig, Germany. [7]Medical Department III – Endocrinology, Nephrology, Rheumatology, University of Leipzig Medical Center, Leipzig, Germany. [8]Present address: Medical Oncology Department, Virgen de la Victoria University Hospital, Málaga Biomedical Research Institute (IBIMA)-CIMES-UMA, Málaga, Spain. [9]Present address: Helmholtz Institute for Metabolic, Obesity and Vascular Research (HI-MAG), Helmholtz Zentrum München, University of Leipzig and University Hospital Leipzig, Leipzig, Germany. [10]Present address: Stem Cell Bio Regenerative Med Institute, Stanford University, Stanford, CA, USA. [11]Present address: Department of Bioengineering, Stanford University, Stanford, CA, USA. [12]These authors contributed equally: Laura C. Hinte, Daniel Castellano-Castillo. ✉e-mail: ferdinand.vonmeyenn@hest.ethz.ch

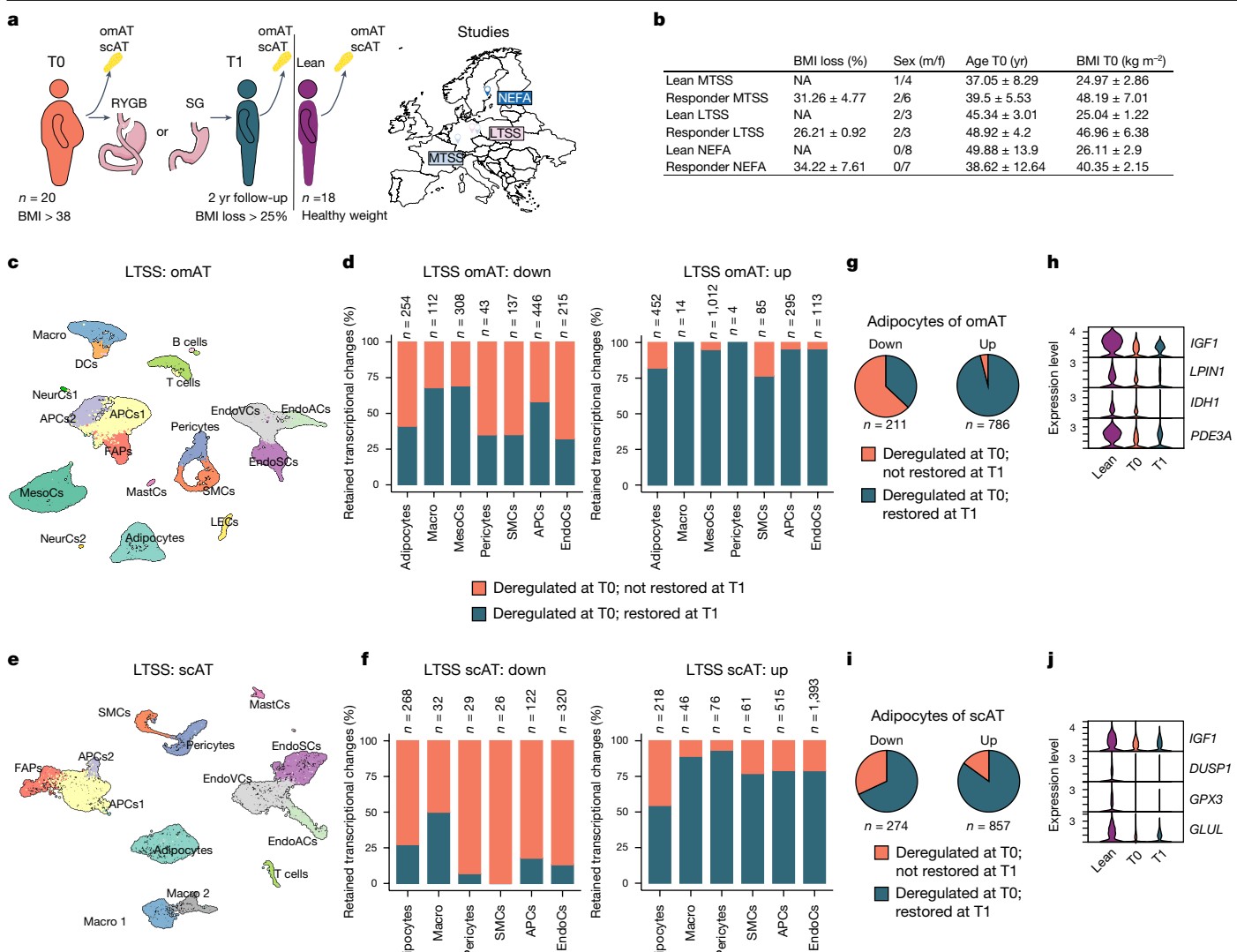

**Fig. 1 | Human AT retains cellular transcriptional changes after BaS-induced WL. a**, omAT and scAT biopsies were collected from people living with obesity during BaS (T0) and 2 yr post-surgery (T1). Only individuals that had lost at least 25% of BMI compared with T0 were included. omAT and scAT biopsies were collected from healthy weight/lean individuals from the same studies (MTSS, LTSS and NEFA). **b**, Sex, age, starting BMI and BMI loss of lean donors and donors with obesity. **c**, Uniform manifold approximation and projection (UMAP) of 22,742 nuclei representing omAT pools from lean subjects ($n = 5$; 2 males, 3 females) and paired omAT from T0 and T1 ($n = 5$ each; 2 males, 3 females) from LTSS. **d**, Proportion of retained transcriptional changes in highly abundant cell types of LTSS omAT. **e**, UMAP of 15,347 nuclei representing scAT pools from lean subjects ($n = 5$; 2 males, 3 females) and paired scAT from T0 and T1 ($n = 5$ each; 2 males, 3 females) from LTSS. **f**, Proportion of retained transcriptional changes in highly abundant cell types of LTSS scAT. **g**, Proportion of retained transcriptional changes in integrated omAT adipocytes of LTSS and MTSS omAT. **h**, Normalized expression of selected memory DEGs in omAT adipocytes. **i**, Proportion of retained transcriptional changes in integrated omAT adipocytes of LTSS and NEFA scAT. **j**, Normalized expression of selected memory DEGs in scAT adipocytes. Wilcoxon rank-sum test with adjusted $P < 0.01$ by the Bonferroni correction method, and $\log_2$ fold change ($\log_2$FC) > ±0.5 was used for DEG identification in **d**, **f**, **g**, **h**, **i** and **j**. DCs, dendritic cells; EndoCs, endothelial cells; EndoACs, arteriolar EndoCs; EndoSCs, stalk EndoCs; EndoVCs, venular EndoCs; LECs, lymphatic EndoCs; FAPs, fibro-adipogenic progenitors; Macro, macrophages; MastCs, mast cells; MesoCs, mesothelial cells; NeurCs, neuronal-like cells; SMCs, (vascular) smooth muscle cells; NA, not applicable; m/f, male/female. Credit: **a**, Copyright 2017−Simplemaps.com (https://simplemaps.com/resources/svg-maps).

## Transcriptional changes in human AT

To explore whether signatures of previous obesogenic states persist in humans after appreciable WL, we obtained subcutaneous AT (scAT) and omental AT (omAT) biopsies from individuals with healthy weight who have never had obesity (called healthy weight here) and people living with obesity (but without diabetes) before (T0) and 2 yr after (T1) BaS from multiple independent studies (Fig. 1a). The omAT samples were from the multicentre two-step surgery (MTSS) study ($n = 5$ lean individuals, 1 male, 4 females; $n = 8$ individuals with obesity, 2 males, 6 females) and Leipzig two-step surgery (LTSS) study ($n = 5$ lean

individuals, 2 males, 3 females; $n = 5$ individuals with obesity, 2 males, 3 females). Only patients exhibiting a minimum of 25% body mass index (BMI) reduction were included into our study (Fig. 1a,b and Extended Data Table 1). We performed snRNA-seq on pooled omAT per group and could annotate, on the basis of published data[34,35], 18 cell clusters in the omAT samples (Fig. 1c and Extended Data Figs. 1a and 2a–d), including adipocytes, adipocyte progenitor cells (APCs), mesothelial cells, immune cells and endothelial cells. Although we did not observe consistent cellular composition differences between T0 and T1 in omAT, we observed inter-individual cellular composition variations after single nucleotide polymorphism (SNP)-based demultiplexing, possibly also

affected by sampling during surgery (Extended Data Fig. 2e,f). Notably, cell type-specific gene expression analysis revealed that many differentially expressed genes (DEGs) at T0 (obese versus healthy weight) were also deregulated at T1 in both studies (Fig. 1d and Extended Data Fig. 1b,c). We next performed the same analysis with scAT biopsies from the LTSS study (n = 5 lean individuals, 2 males, 3 females; n = 5 individuals with obesity, 2 males, 3 females) and NEFA trial (ClinicalTrials.gov registration no. NCT01727245; n = 8 lean individuals, all female; n = 7 individuals with obesity, all female), including only patients exhibiting a minimum of 25% BMI reduction (Fig. 1a,b and Extended Data Table 1). We annotated 13 cell clusters for scAT (Fig. 1e and Extended Data Fig. 1d), including APCs, adipocytes, endothelial cells and immune cells, on the basis of published markers[34–37] (Extended Data Fig. 3a–d). We did not observe consistent cellular composition differences between T0 and T1 in scAT (Extended Data Fig. 3e,f). However, similar to omAT we found in both studies that many cell types retained transcriptional differences from T0 to T1 (Fig. 1f and Extended Data Fig. 1e,f). A further detailed analysis of cell type-specific gene expression changes in omAT and scAT showed that transcriptional deregulation during obesity was most pronounced in adipocytes, APCs and endothelial cells (Extended Data Fig. 1g–j). In line with this observation, the absolute number of retained DEGs from T0 to T1 was highest in these cell types as well (Extended Data Fig. 1k). Given that adipocytes showed strong retainment of transcriptional differences in each individual sample, we integrated the snRNA-seq data of all adipocytes from the omAT and scAT studies, respectively (Extended Data Fig. 1l,m), and performed differential gene expression analysis. Pooled omAT adipocytes displayed a strong retention of downregulated DEGs (Fig. 1g), including relevant metabolic genes[38–41] such as *IGF1*, *LPIN1*, *IDH1* or *PDE3A* (Fig. 1h). Similarly, the retention of downregulated DEGs in scAT adipocytes was pronounced (Fig. 1i) and included relevant metabolic genes[38,42–44] such as *IGF1*, *DUSP1*, *GPX3* and *GLUL* (Fig. 1j). Gene set enrichment analysis (GSEA) of retained DEGs in adipocytes of each study showed persistent downregulation of pathways linked to adipocyte metabolism and function (Extended Data Fig. 4a–d) and persistent upregulation of pathways linked to fibrosis (related to *TGFβ* signalling) and apoptosis (Extended Data Fig. 4e–h). These results indicate that obesity induces cellular and transcriptional (obesogenic) changes in the AT, which are not resolved following significant WL.

## Pathophysiology mostly resolves after WL

To investigate the molecular mechanisms and pathophysiological importance of this putative metabolic memory of obesity, we assessed WL in an experimental animal model (Fig. 2a). The 6-week-old male mice were fed a high-fat diet (HFD; 60% kcal from fat) or low-fat chow diet (10% kcal from fat) for 12 (H and C) or 25 weeks (HH and CC_l). Subsequently, we switched the diet to a standard chow diet (HC, CC_s, HHC, CCC), leading to weight normalization in 4–8 weeks (Fig. 2b,c). Glucose tolerance was impaired in H but not in HH mice (compared with age-matched controls), whereas insulin sensitivity was lower in HH but not in H mice (Extended Data Fig. 5a,b). Fasting blood glucose levels were greater in both groups (Extended Data Fig. 5c). WL restored insulin sensitivity in HHC mice, whereas HC mice still showed impaired glucose tolerance (Extended Data Fig. 5d,e). Fasting glucose levels were normalized by WL in both groups, matching those of control mice (Extended Data Fig. 5f). After WL, hyperinsulinemia was resolved in HC mice, but only diminished in HHC mice (Extended Data Fig. 5g–i). Leptin levels, which were elevated in obese mice, returned to control levels after WL (Extended Data Fig. 5j). Energy expenditure and food intake showed no differences between HC and CC_s mice after WL (Extended Data Fig. 5k,l). Liver triglyceride accumulation was normalized (to control levels) in HC, and most HHC, mice. (Extended Data Fig. 5m,n). Similarly, C and H mice, and CC_s and HC mice, did not differ in the amount of lean mass nor did HC mice lose lean mass (Extended Data Fig. 5o). Obese

H mice had larger subcutaneous inguinal AT (ingAT), epididymal AT (epiAT) and brown AT (BAT) depots than corresponding control mice (Extended Data Fig. 5p,q). ingAT and BAT depot sizes normalized after WL. In line with a recent report, epiAT of HC mice was smaller than that of controls after WL[18]. Interestingly, the phenomenon of epiAT shrinkage was already observed during obesity in 25-week HFD-fed (HH) mice, as previously reported[45], and maintained after WL in HHC mice (Extended Data Fig. 5r–v). Adipocyte sizes varied between depots, and adipocytes were enlarged in ingAT of H and HH mice and normalized after WL in HC, but not in HHC, mice (Extended Data Fig. 5w,x). epiAT adipocytes were also enlarged and shrunk to normal sizes in H and HC mice, respectively, whereas in HH and HHC mice adipocytes were of equal size, probably owing to the tissue shrinkage (Extended Data Fig. 5v,y). The epiAT of obese mice (H and HH) showed immune cell infiltration and apical fibrosis, which partially improved after WL in HC, but not HHC, mice (Extended Data Fig. 5t–v). Masson's trichrome staining showed more collagen deposition in epiAT after WL (Extended Data Fig. 5z). Overall, after WL, only a few mild metabolic impartments persisted, including glucose intolerance in HC mice, hyperinsulinemia and slight liver steatosis in HHC mice and a notable decrease in epiAT depot size after WL in both groups.

## Transcriptional obesogenic memory in mice

Considering our observations of persistent transcriptional changes in human AT, we examined mouse epiAT cellular changes throughout obesity and WL using snRNA-seq. We annotated 15 key cell populations using common marker genes[34,36,46], including APCs, immune cells, adipocytes, mesothelial cells, endothelial cells and epithelial cells (Fig. 2d and Extended Data Fig. 6a,b). Consistent with previous findings[16,18,46], macrophage cell number in epiAT was higher in obese conditions (H and HH), and was not fully normalized after WL, especially in HHC mice (Fig. 2e). Resident macrophages in control mice (C, CC and CCC) primarily consisted of perivascular macrophages and non-perivascular macrophages. Notably, during obesity mainly lipid-associated macrophage (LAM) and non-perivascular macrophage cell numbers increased in the epiAT, altering the macrophage population composition persistently (Fig. 2f and Extended Data Fig. 6c,d).

Motivated by our own observation of persistent transcriptional changes in human AT (Fig. 1 and Extended Data Fig. 1) and corresponding recent reports in endothelial and immune cells[18,19], we next investigated transcriptional retention ('memory') in the mouse epiAT. On the basis of the number of DEGs in each cell type, we found stronger transcriptional deregulation in obesity and after WL in adipocytes, APCs, endothelial cells, epithelial cells and macrophages than in other cell types (Extended Data Fig. 7a), corroborating the existence of persistent, cell-specific transcriptional changes in mouse epiAT. Indeed, across cell types many DEGs from the obesity time point remained deregulated after WL (Fig. 2g,h and Extended Data Fig. 7b,c). GSEA of retained DEGs in adipocytes, APCs, endothelial cells, LAMs, non-perivascular macrophages, perivascular macrophages and mesothelial cells showed persistent upregulation in HC and HHC mice of genes related to lysosome activity, apoptosis and other inflammatory pathways (Extended Data Fig. 7c,d), indicating endoplasmic reticulum and cellular stress. Persistently downregulated retained DEGs in HC and HHC mice were mainly related to metabolic AT pathways, such as fatty acid omega oxidation, fatty acid biosynthesis, adipogenesis or peroxisome proliferator-activated receptor signalling (Extended Data Fig. 7e,f), pointing to potential dysfunction in the AT after WL.

Focusing specifically on adipocytes, we identified three distinct patterns of DEGs (Fig. 2i): a group that failed to restore normal expression after WL in HC or HHC (for example, *Maob* or *Ctsd*); another group that restored expression in HC but not in HHC (for example, *Cyp2e1* or *Runx2*); and a third group that restored normal expression after WL in both HC and HHC mice (for example, *Gpam* or *Tyrobp*).

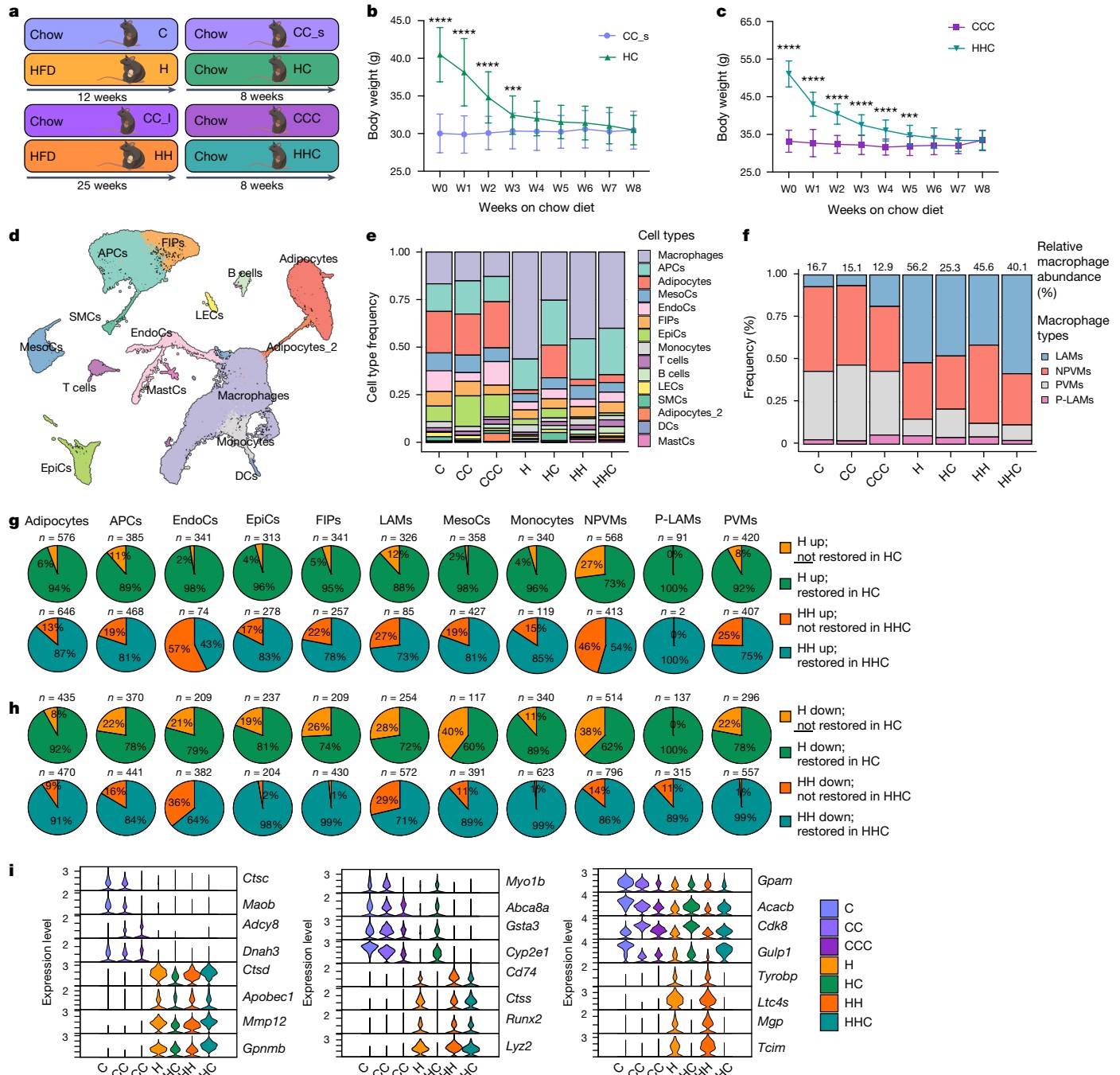

**Fig. 2 | Transcriptional changes persist WL induced (partial) remodelling of epiAT. a**, Experimental setup of the WL study. **b**,**c**, 18-week-old (**b**) or 31-week-old (**c**) diet-induced obesity or age-matched control male mice were fed chow diet for 8 weeks. Body weight (*n* = 20 each; data from two experiments). **d**, UMAP of 48,046 nuclei representing integrated epiAT pools (*n* = 5 pooled mice each) from C, CC, CCC, H, HC, HH and HHC mice. **e**, Relative abundance of cell types/clusters per condition. **f**, Relative abundance of macrophage subclusters per condition as percentage of total macrophages. **g**,**h**, Proportion of retained upregulated (**g**) or downregulated (**h**) transcriptional changes in different cell types. **i**, Normalized expression of selected DEGs in adipocytes across all conditions that did not restore expression profile (left), restored only in HC adipocytes (middle) or restored expression profile (right) (Wilcoxon rank-sum test, adjusted *P* < 0.05 by the Bonferroni correction method; FC > ±0.5). Significance for **b** and **c** was calculated using unpaired, multiple *t*-tests with Benjamini, Krieger and Yekutieli post-hoc test for multiple comparisons. ***FDR < 0.001, ****FDR < 0.0001. Exact *P* values are in the Source Data. EpiCs, epithelial cells; FDR, false discovery rate; FIPs, fibro-inflammatory progenitors; NPVMs, non-perivascular macrophages; PVMs, perivascular macrophages; P-LAMs, proliferating LAMs; W, week.

Notably, we did not identify any DEGs that exclusively restored normal expression after WL in HHC mice but not in HC mice, suggesting that longer durations of obesity or relatively shorter WL periods exert a stronger influence on retainment of a transcriptional memory. In summary, after WL, adipocytes from mice maintained an upregulation of inflammatory- and extracellular matrix remodelling-related pathways, whereas adipocyte-specific metabolic pathways remained downregulated (Extended Data Fig. 7g,h), mirroring our findings from human adipocytes (Fig. 1h,j and Extended Data Fig. 4).

## Epigenetic obesogenic memory in mice

Having established the persistence of obesity-associated transcriptional changes after WL in human AT and mouse epiAT, our attention

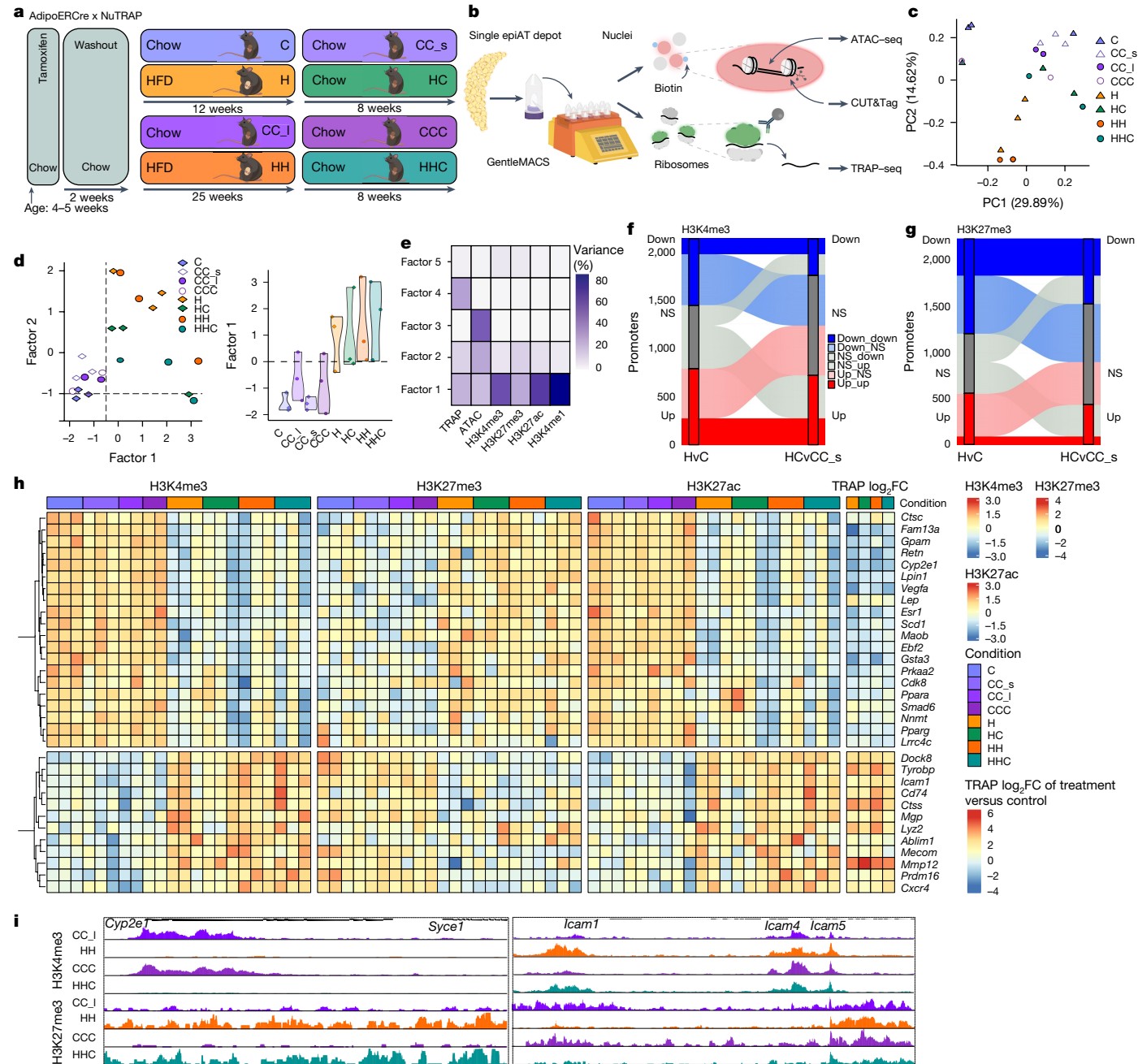

**Fig. 3 | Adipocyte promoters retain an epigenetic memory. a**, Experimental setup of the WL study in AdipoERCre x NuTRAP mice. **b**, Workflow of paired CUT&Tag, ATAC–seq and TRAP–seq from one AT depot. Biotinylated nuclei and GFP-tagged ribosomes are isolated from frozen tissue, pulled down and subjected to CUT&Tag, ATAC–seq (nuclei) and TRAP–seq (ribosomes). **c**, PCA of translatome (TRAP–seq) of labelled adipocytes from C, CC_s, CC_l, CCC, H, HH, HC and HHC. Each dot represents an individual biological replicate. **d**, MOFA plots showing the sample clustering along latent Factors 1 and 2 (left) and Factor 1 value distribution (right) across labelled adipocytes. Each dot corresponds to one biological replicate. For each replicate all six modalities are represented in one dot. **e**, Percentage of variance explained by each MOFA factor across one of six modalities. **f**, Dynamics of differentially H3K4me3-marked promoters (*y* axis) from H to HC. **g**, Dynamics of differentially H3K27me3-marked promoters (*y* axis) from H to HC. **h**, Scaled enrichment of H3K4me3 (left), H3K27me3 (middle) and H3K27ac (right) at selected promoters of genes and the $\log_2$FC of TRAP–seq from comparisons against controls for the same genes. **i**, Distribution of normalized reads of H3K4me3 and H3K27me3 at the *Cyp2e1* and *Icam1* loci across conditions. Scaling of reads was performed per hPTM. NS, not significant; v, versus.

shifted towards exploring the underlying mechanisms conferring this putative memory. We decided to focus on adipocytes given their post-mitotic nature, immobility, long lifespan and central position in AT biology[47]. We conducted an epigenetic analysis of adipocytes derived from mouse epiAT. Considering the inherent difficulties in studying epigenetic signatures in heterogenous cell populations, we crossed tamoxifen-inducible AdipoERCre mice with NuTRAP reporter mice, and thereby labelled adipocyte nuclei with biotin and GFP-tagged ribosomes before HFD feeding (Fig. 3a). We then developed a protocol to assay multiple modalities from labelled adipocytes of the same epiAT depot (Fig. 3b) and performed paired analysis of the translatome using targeted purification of polysomal messenger RNA (translating ribosome affinity purification followed by RNA sequencing technology (TRAP–seq)), chromatin accessibility using assay for transposase-accessible chromatin (ATAC) with sequencing (ATAC–seq) and four histone post-translational modifications (hPTMs) using cleavage under targets and tagmentation (CUT&Tag). In essence, we generated extensive epigenetic datasets from adipocytes of each epiAT sample (Extended

Data Fig. 8a,b) encompassing H3K27me3 (a polycomb-mediated repressive hPTM), H3K4me3 (which marks active transcription start sites (TSS)), H3K4me1 (indicative of active or poised enhancers) and H3K27ac (which marks active enhancers and other candidate *cis*-regulatory elements)[48,49]. We observed strong correlation between the transcriptional profiles of labelled adipocytes and the adipocyte clusters identified by snRNA-seq (Extended Data Fig. 8c). Consistent with our observation from the snRNA-seq, we also noted a restoration of the translational profile in adipocytes from HC and HHC mice (Fig. 3c).

Next, to identify sources of biological variability (factors) in our datasets on the basis of all modalities across all conditions we used multi-omics factor analysis (MOFA)[50]. This enables unsupervised integration and clustering of our paired multi-omic (epigenetic) datasets to overcome potential limitations of modality-specific analyses. HC and HHC samples clustered closer to H and HH samples than to controls along Factor 1, indicating that WL did not induce complete normalization of the adipocyte epigenome (Fig. 3d). MOFA inferred Factor 1 as the main source of data variability between the conditions, which was predominantly influenced by active hPTMs (Fig. 3e).

Motivated by our MOFA findings, we investigated promoters marked by H3K4me3 or H3K27me3 to identify differentially marked promoters for these hPTMs (Extended Data Fig. 8d). We examined the dynamics of these modifications between adipocytes from obese and WL mice. More than 1,000 promoters showed differential enrichment of H3K4me3 in H and HC mice (H: 1,475; HC: 1,094), with a majority showing increased H3K4me3 levels (Fig. 3f). Similarly, 859 promoters were differentially marked in HH and HHC mice (Extended Data Fig. 8e). Overall, many promoters remained activated after WL that were less actively marked in controls, and vice versa. In contrast to H3K4me3, overall, more promoters lost than gained H3K27me3 in obese mice, and a substantial number of these promoters remained repressed or did not regain trimethylation at K27 after WL compared with controls (Fig. 3g and Extended Data Fig. 8e).

We next performed a functional analysis of differentially marked promoters. The activity status of many promoters switched, transitioning from active (H3K4me3 and/or H3K27ac) to repressed (H3K27me3), or vice versa, in obese and WL conditions, compared with control samples. Many of these epigenetic changes were also reflected in the translatome (Fig. 3h) and nuclear transcriptome (Fig. 2g,h). Promoters that remained repressed (high H3K27me3 and low H3K4me3 and/or H3K27ac) were linked to adipocyte function-related genes (for example, *Gpam*, *Cyp2e1* or *Acacb*), whereas promoters that remained active (that is, high H3K4me3 and/or H3K27ac and low H3K27me3) were related to genes involved in extracellular matrix remodelling and inflammatory signalling (for example, *Icam1*, *Lyz2* or *Tyrobp*) (Fig. 3h,i). By GSEA, we confirmed that H3K4me3 persistence in adipocytes from H/HC and HH/HHC mice was associated with chemokine and inflammatory processes (Extended Data Fig. 8f,g). Persistent H3K4me3 loss in H/HC-affected genes included those involved in adipocyte functions (for example, adipogenesis, triacylglyceride synthesis, peroxisome proliferator-activated receptor signalling, leptin and adiponectin signalling) (Extended Data Fig. 8f), whereas adipogenesis-related genes were repressed by H3K27me3 gain and H3K4me3 loss in adipocytes from HHC/HH mice (Extended Data Fig. 8g), suggesting a persistently impaired adipocyte function. Notably, the expression of relevant epigenetic modifiers was not deregulated in HC or HHC adipocytes (Extended Data Fig. 8h).

Enhancers are key drivers of cellular identity and cell fate[51,52]. MOFA (Fig. 3d) indicated that active (H3K27ac) and enhancer (H3K4me1) hPTMs, together with chromatin accessibility (ATAC), were the modalities mostly explaining data variability across all conditions. An analysis of the correlation coefficients of hPTM signatures in each condition against an aggregated control (composed of averaged healthy young controls) revealed large deviations between H3K27ac and H3K4me1 in obese or WL conditions from control mice (Fig. 4a and Extended Data Fig. 9a). We generated adipocyte-specific enhancer annotations for each condition on the basis of our data (Extended Data Fig. 9b–e) and analysed enhancer dynamics in obese and WL mice. Next, we performed differential enrichment analysis of H3K4me1, H3K27ac and ATAC–seq in enhancers. By principal component analysis (PCA), we found that HC and HHC samples clustered closer to H and HH than to controls for H3K4me1, ATAC and H3K27ac (Fig. 4b and Extended Data Fig. 9f–h). H3K4me1 separated H/HH and HC/HHC from controls, indicating that not only active but also poised enhancers could drive persistent epigenetic alterations. We then analysed the dynamic behaviour of enhancers between the obese and WL adipocytes. Several thousand enhancers were differentially marked by H3K4me1 during obesity (H: 4,255, HH: 3,237) and/or after WL (HC: 3,439, HHC: 6,589), and remained altered from H to HC ($n$ = 848) and from HH to HHC ($n$ = 857) (Fig. 4c).

We termed enhancers that gained (and maintained) H3K4me1 in obesity and WL 'new enhancers'. Most of these 'new enhancers' were also active (that is, marked by H3K27ac) during obesity and/or WL (Fig. 4d). We then annotated the enhancers to their closest gene and performed a GSEA. In agreement with the promoter GSEA above, we found that the 'new active enhancers' were related to inflammatory signalling, lysosome activity and extracellular matrix remodelling (Fig. 4e and Extended Data Fig. 9i), indicating a persistent shift of adipocytes towards a more inflammatory and less adipogenic identity. Corroborating these results, Roh et al. had analysed H3K27ac in adipocytes of obese mice and reported impaired identity maintenance during obesity[25].

To combine our findings regarding retained translational changes and epigenetic memory, we investigated whether epigenetic mechanisms, such as differentially marked promoters or enhancers, could explain the persistent translational obesity-associated changes after WL. Notably, 57–62% of downregulated and 68–75% of upregulated persistent translational DEGs after WL could be accounted for by one or more of the analysed epigenetic modalities (Fig. 4f). Overall, these results strongly suggest the presence of stable cellular, epigenetic and transcriptional memory in mouse adipocytes that persists after WL.

## Metabolic memory primes adipocytes

We then asked whether this persistent memory primed mature adipocytes to respond differently to nutritional stimuli than non-primed controls. We collected mature epiAT and ingAT adipocytes from WL and control mice, cultured them for 48 h and then assessed glucose and palmitate uptake. Adipocytes from WL epiAT showed increased glucose and palmitate uptake compared with controls (Fig. 5a,b). ingAT adipocytes from HHC mice displayed significantly increased glucose uptake compared with controls, and for HC adipocytes we observed a trend towards an increased uptake (Extended Data Fig. 10a). Assessing adipogenesis capacity, we found that the SVF from epiAT of HC and HHC mice accumulated lipids in response to insulin but failed to differentiate, unlike controls (Extended Data Fig. 10b). Adipogenesis was slightly impaired in the SVF from ingAT of WL mice compared with controls (Extended Data Fig. 10c). These findings indicate that persistent cellular memory confers phenotypic consequence ex vivo.

Next, we investigated the response of WL and control mice to 4 weeks of HFD feeding. HC mice gained weight faster than CC_s mice (called HCH and CCH, respectively, here) (Fig. 5c). Fasting blood glucose levels and postprandial insulin levels were elevated in HCH mice (Fig. 5d,e), but neither glucose tolerance nor insulin sensitivity was impaired when compared with CCH mice (Extended Data Fig. 10d–g). Leptin levels in HCH mice returned to H mice levels, whereas CCH mice did not show a significant increase (Fig. 5f). Adipocytes in epiAT from HCH mice were larger on average, resembling the adipocyte size distribution of H mice, whereas epiAT adipocytes from CCH mice were similar to those in CC_s mice (Extended Data Fig. 10h). HCH mice exhibited larger ingAT, BAT and epiAT depots compared with CCH mice (Fig. 5g,h) and showed increased triglyceride accumulation and hepatic steatosis (Extended Data Fig. 10i–k).

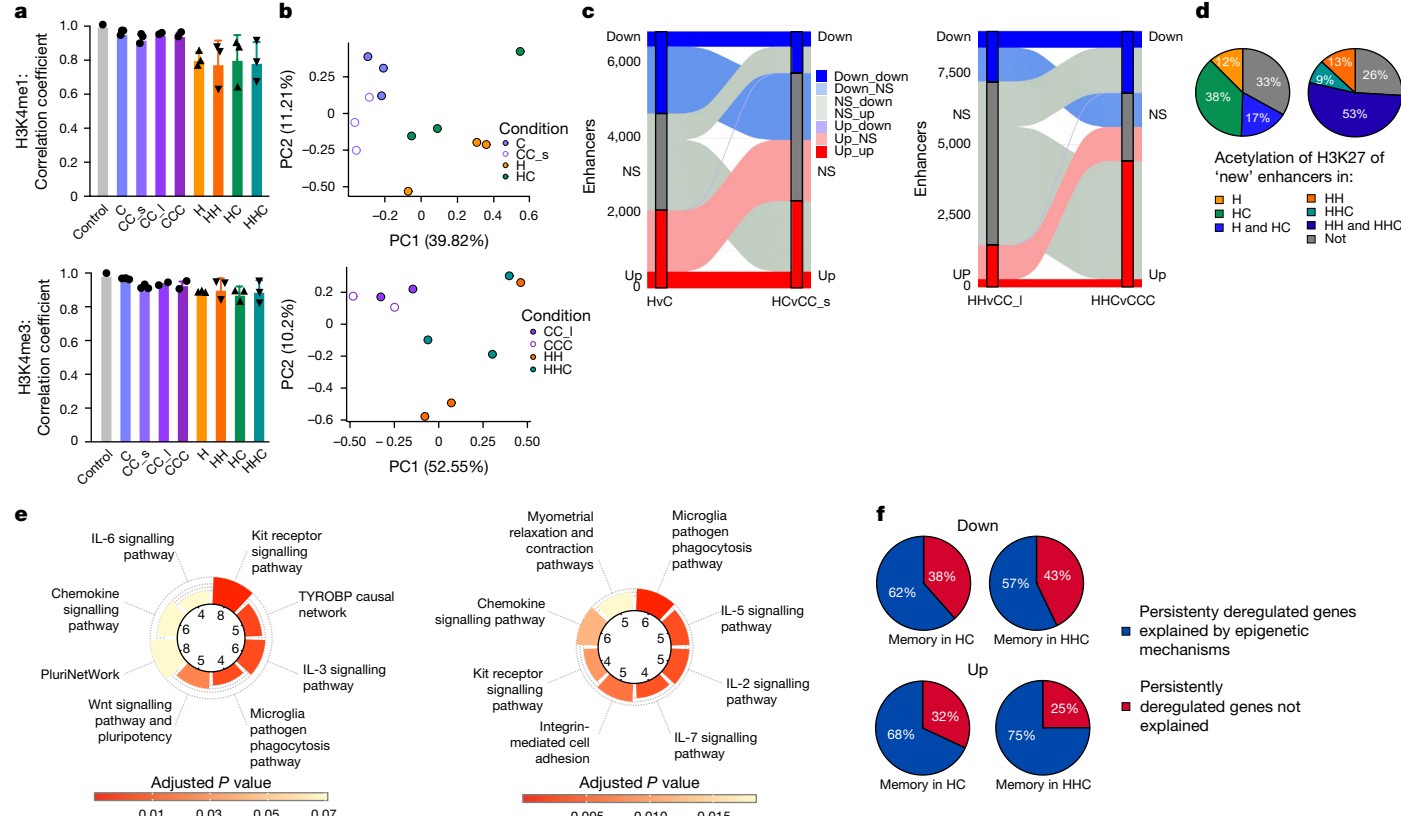

**Fig. 4 | Adipocyte enhancers retain an epigenetic memory. a**, Correlation coefficient $R$ (Pearson) of quantified peaks of H3K4me1 and H3K4me3 against a hypothetical healthy control ($n$ = 2–3 each) with s.d. Each dot represents an individual biological replicate. **b**, PCA plots of quantified adipocyte-specific enhancers as marked by H3K4me1. Each dot represents an individual biological replicate. **c**, Dynamics of differentially H3K4me1-marked enhancers ($y$ axis) from H to HC (left) and HH to HHC (right). **d**, H3K27ac status of genes linked to newly emerged enhancers marked by H3K4me1 (from **c**) in different conditions

identified by the presence of an H3K27ac peak associated to the gene. $n$ = 218 left and $n$ = 127 right. **e**, Top (significant) pathway terms for genes linked to newly emerged acetylated enhancers for H and HC (left) and HH and HHC (right) on the basis of WikiPathways database (Fisher's exact test, adjusted $P$ < 0.05 by the Benjamini–Hochberg method for correction). **f**, Proportion of down- and upregulated memory DEGs from TRAP–seq that can be explained by one or more epigenetic modality in HC ($n$ = 13; $n$ = 72) and HHC ($n$ = 7; $n$ = 36).

We performed snRNA-seq of epiAT from HCH and CCH mice and observed higher macrophage infiltration in both HCH and CCH epiAT compared with the WL time point, with a greater infiltration in HCH epiAT (Fig. 5i and Extended Data Fig. 10l). The proportion of LAMs was greater in HCH epiAT, similar to that of H and HC mice, whereas CCH epiAT showed a greater proportion of LAMs compared with CC epiAT, indicating LAM infiltration occurred early during HFD feeding (Extended Data Fig. 10m).

We assessed whether HCH and CCH adipocytes exhibited transcriptional differences. Neither the previous transcriptional status nor the transcriptional memory at the HC time point explained the transcriptional deregulation observed in adipocytes from HCH mice (Fig. 5j). Further analysis revealed that several DEGs in the HCH group were altered during obesity but recovered after WL in HC mice. Interestingly, these overlapped with promoters and enhancers carrying epigenetic memory (Figs. 3h and 5k,l). A more detailed analysis showed that epigenetic signatures could explain the 3–6 times more DEGs in the HCH group than the transcriptional memory or previous transcriptional status during HC (Fig. 5m). Specifically, the four hPTMs and ATAC–seq could predict or explain 31% of upregulated DEGs, which were related to inflammation, and 60% of downregulated DEGs, many of which were related to adipocyte function and identity, in the HCH group (Extended Data Fig. 10n,o).

Together, these findings suggest that a persistent epigenetic memory, including local changes of hPTM deposition, contributes to the altered transcriptional response in adipocytes in the 'yo-yo' model of

dieting and primes adipocytes for pathological responses to further HFD feeding, thus contributing to the pathophysiology of rebound obesity in mice. It is possible that other epigenetic modifications, such as other hPTMs, DNA methylation or non-coding RNAs, also contribute to the observed phenomena.

Although we performed well-controlled dietary intervention experiments in mice, the human AT samples were obtained from different BaS studies and AT depots, and reflect an overall heterogenous group of participants. Indeed, BaS is a successful but invasive method for achieving long-term WL[53], yet sleeve gastrectomy and Roux-en-Y gastric bypass (RYGB) also affect the gut microbiome, micronutrient absorption, bile acid metabolism and incretin signalling[54–57]. Nonetheless, we consistently observed retained transcriptional differences after significant WL in AT cells after sleeve gastrectomy (MTSS and LTSS studies), which induced significant WL, as well as after RYGB (NEFA study), which resulted in a complete return to a non-obese or lean state. The aforementioned alterations and the degree of WL achieved between individuals and studies are confounders that limit the direct comparability of our mouse and human data. The rapid WL achieved by BaS may even reduce or modify putative cellular memory in the human AT. Owing to the current lack of methods to isolate pure adipocyte nuclei from frozen human tissue, we could not perform the corresponding epigenetic analyses in human samples. Nonetheless, it stands to reason that obesity-induced transcriptional (and cellular) changes in humans are also mediated through epigenetic mechanisms that can persist after WL in the AT and contribute to human (patho)physiology.

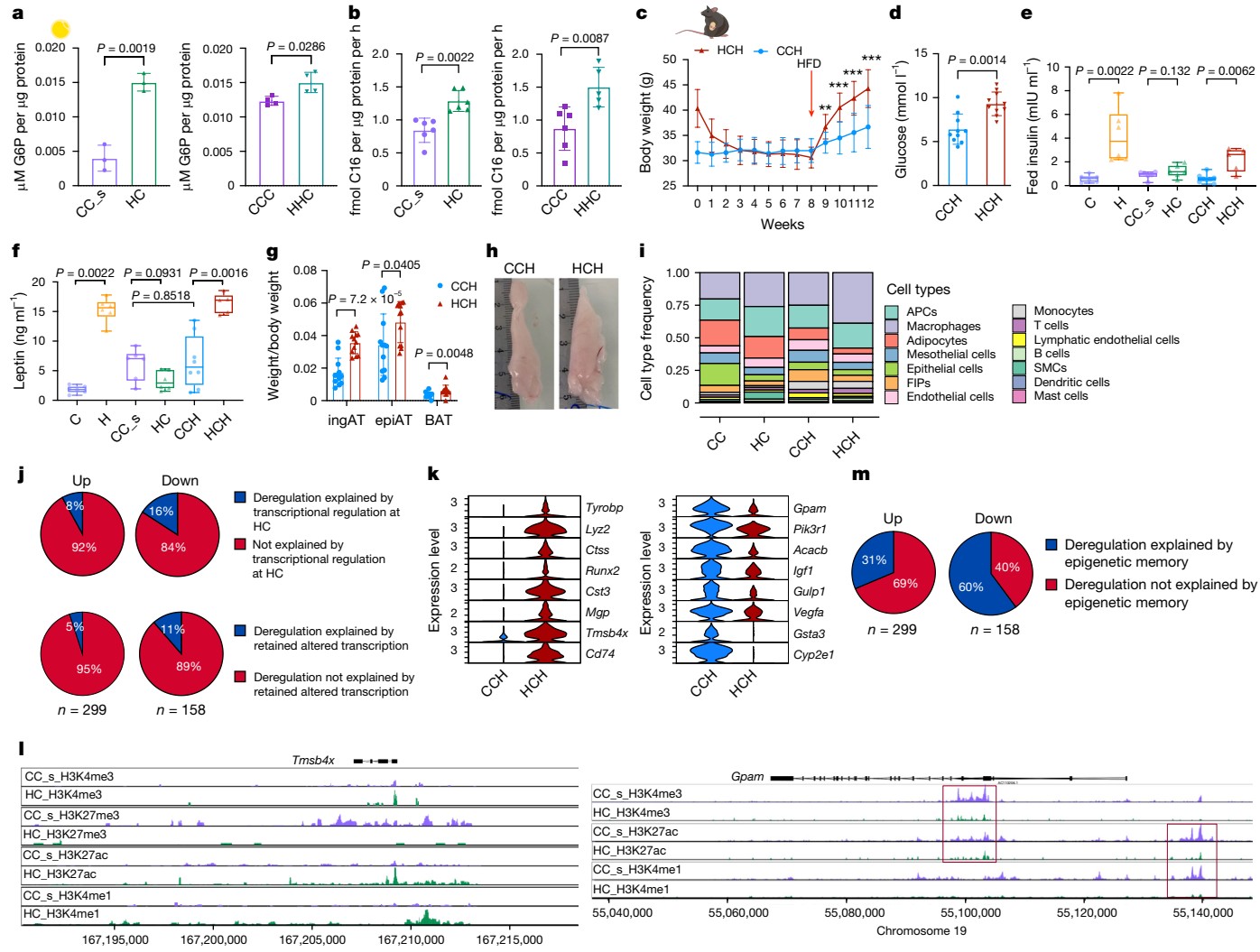

**Fig. 5 | Memory primes adipocytes and mice for an accelerated response to obesogenic stimuli. a**,**b**, Experiments with isolated, cultured primary epiAT adipocytes. Each dot represents an individual biological replicate of a pool of three mice. **a**, Glucose uptake. **b**, Palmitate uptake. **c**, HC and CC_s mice were put on HFD for 4 weeks. Body weight (n = 12 each). **d**–**g**, Experiments with HHC and CCH mice. Each dot indicates an individual biological replicate from two experiments. **d**, Fasting blood glucose (n = 10). **e**,**f**, Fed insulin (**e**) and leptin (**f**) (C, H, CC_s, HC: n = 6 each; CCH: n = 8; HCH: n = 5). **g**, Weights of ingAT, epiAT and BAT, normalized to body weight (n = 10). **h**, Representative images of epiAT. The ruler is in cm. **i**, Relative cell type abundance. **j**, Proportion of up- and downregulated DEGs in HCH adipocytes that can be explained by DEG status at HC time point or transcriptional memory. **k**, Normalized expression of selected DEGs in HCH adipocytes that were recovered in HC but were still differentially marked by one or more epigenetic modalities (Wilcoxon rank-sum test, adjusted P < 0.05 by the Bonferroni correction method; FC > ±0.5). **l**, Distribution of normalized reads of H3K4me3, H3K27me3, H3K27ac and H3K4me1 of HC and CC_s adipocytes at loci of *Tmsbx4* and *Gpam*. Scaling of reads was performed per hPTM. **m**, Proportion of up- and downregulated DEGs in HCH adipocytes that can be explained by an epigenetic memory. Significance was calculated between age-matched controls and experimental groups. Significance for **a**, **b**, **d**, **e**, **f** and **g** was calculated using two-tailed Mann–Whitney tests. Significance for **c** was calculated using unpaired, multiple t-tests with Benjamini, Krieger and Yekutieli post-hoc test for multiple comparisons. **FDR < 0.01, ***FDR < 0.001. Error bars represent s.d. Boxplots represent minimum, maximum and median.

Although our results do not provide final proof of a causal relationship between AT memory and the systemic yo-yo effect of accelerated weight gain, Hata et al. have shown that transplantation of WL epiAT into control mice enhances macular degeneration by impacting immune cells and angiogenesis and that epiAT macrophages retain an altered chromatin accessibility after WL[28]. Our epigenetic analysis of adipocytes of the same tissue could serve as an explanation of how these alterations in chromatin accessibility can be retained. Further investigations are required to determine whether—in addition to adipocytes and macrophages[28]—other post-mitotic or cycling cells, such as myofibers, neurons or APCs, also establish an epigenetic memory of obesity and contribute to the observed systemic weight regain effect.

Although our results are on the basis of BaS studies, the susceptibility to weight regain in human subjects undergoing WL using strict dietary regimens might be related to a transcriptional and/or epigenetic memory as well. At present, the use of incretin receptor agonists such as semaglutide or tirzepatide[6,58,59] has emerged as a promising non-invasive strategy for significant WL. However, the extent to which these agonists induce long-lasting WL and physiological changes in humans beyond withdrawal remains poorly studied. Studies on semaglutide and on tirzepatide have shown that substantial weight regain occurs after their withdrawal[6,59], indicating that at least these treatments do not induce stable, persistent changes. Whether this is also the case for other agonists remains to be investigated. Further studies are needed to elucidate whether these treatments could erase or diminish an obesogenic memory better than other non-surgery-based WL strategies.

The presence of a putative obesogenic epigenetic memory in adipocytes and potentially other cells suggests new potential therapeutic

avenues to improve WL maintenance in humans. Although our experiments focused on obesity, it is plausible that epigenetic memory could also play a role in many other contexts, including addictive diseases. Recent advancements in targeted epigenetic editing[60] and global remodelling of the epigenome[61,62] provide promising new approaches.

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

## Methods

### Data reporting

No statistical methods were used to predetermine sample size. The experiments were not randomized, and the investigators were not blinded to allocation during experiments and outcome assessment.

### Clinical sample acquisition

Human AT biopsies were obtained from three independent studies: MTSS, LTSS and NEFA.

**MTSS.** The MTSS samples comprised samples from omental visceral AT biopsies obtained in the context of a two-step BaS treatment, which included a sleeve gastrectomy as the first step (T0) and laparoscopic RYGB as the second step (T1)[16]. Individuals with syndromal, monogenic, early-onset obesity or individuals with other known concurrent diseases, including acute infections or malignant diseases, were not included in the study. Individuals were not required to adhere to any specific diet before or after surgery but received individual dietary recommendations during regular visits in the obesity management centre. Insulin resistance was determined using a hyperinsulinaemic–euglycaemic clamp technique or the homeostatic model assessment for insulin resistance (HOMA-IR). Only biopsies from individuals that (1) lost 25% or more of BMI between T0 and T1 (Extended Data Table 1), (2) had undergone surgery at the Municipal Hospital Karlsruhe or Municipal Hospital Dresden-Neustadt, (3) were not diagnosed with diabetes, and (4) did not receive any glucose-lowering medication were used for snRNA-seq in this study. AT samples were collected during elective laparoscopic abdominal surgery as previously described[63], snap-frozen in liquid nitrogen and stored at −80 °C. Body composition and metabolic parameters were measured as previously described[64]. Samples of healthy individuals who were not obese were collected during routine elective surgeries such as herniotomies, explorative laparoscopies and cholecystectomies at the same hospitals. The study was approved by the Ethics Committee of the University of Leipzig under approval number 159-12–21052012 and was performed in agreement with the Declaration of Helsinki.

**LTSS.** The human study samples comprised samples from omental visceral and subcutaneous abdominal AT, collected in the context of a two-step BaS treatment. Following an initial sleeve gastrectomy (T0), a laparoscopic RYGB was made in the second step (T1)[16]. Individuals with syndromal, early-onset obesity or individuals with other known concurrent diseases, including acute infections or malignant diseases, were not included in the study. Individuals did not adhere to any specific diet before or after surgery but received individual healthy diet recommendations during regular visits in the obesity management centre. Insulin resistance was determined using HOMA-IR. Only individuals that (1) lost 25% or more of BMI between T0 and T1 (Extended Data Table 1), (2) had undergone surgery at the Leipzig University Hospital, (3) were not diagnosed with diabetes and (4) did not receive any glucose-lowering medication were included. AT samples were collected during elective laparoscopic abdominal surgery as previously described[63], snap-frozen in liquid nitrogen and stored at −80 °C. Body composition and metabolic parameters were measured as previously described[64]. Samples from healthy donors that were not obese were collected during routine elective surgeries (herniotomies, explorative laparoscopies, cholecystectomies) at the same hospital. The study was approved by the Ethics Committee of the University of Leipzig under approval number 159-12–21052012 and performed in agreement with the Declaration of Helsinki.

**NEFA study.** The NEFA study (NCT01727245) comprises samples from subcutaneous abdominal AT from individuals before and after RYGB surgery, as well as healthy controls who had never been obese[8,65]. For this, biopsies were obtained under local anaesthesia before (T0) and 2 yr post-surgery (T1). Only samples from individuals that (1) lost more than 25% BMI between T0 and T1, (2) were not diagnosed with diabetes at T0 and T1 and (3) did not take glucose-lowering medication were included in the present study (Extended Data Table 1). Samples from control subjects were obtained from individuals that were BMI- and age-matched to RYGB patients at T1 as reported previously[8]. AT samples were handled as reported before[65], snap-frozen in liquid nitrogen and stored at −80 °C. The study was conducted in accordance with the Declaration of Helsinki and approved by the Ethics Committee of the Karolinska Institute, Stockholm (approval number 2011/1002-31/1).

### Mice

All mice were kept on a 12-h/12-h light/dark cycle at 20–60% (23 °C) humidity in individually ventilated cages, in groups of between two and five mice, in a pathogen-free animal facility in the SLA building at ETH Zurich. The health of mice was monitored closely, and any mouse exhibiting persistent clinical signs of ill health or distress was excluded from this study. The 16- and 29-week-old male C57BL/6J diet-induced obesity mice (catalogue no. 380050) and diet-induced obesity control mice (catalogue no. 380056) were obtained from The Jackson Laboratory and were kept on the respective diets for another 2 weeks until tissue harvest or diet switch. Different mice were used for insulin tolerance tests and glucose tolerance tests. AdipoERCre[66] and NuTRAP[67] mice were maintained on a C57BL/N background. Homozygous NuTRAP and AdipoERCre mice were bred to generate AdipoERCre x NuTRAP mice. AdipoERCre x NuTRAP mice were kept on HFD or chow diet for 12 or 25 weeks before tissue harvest or diet switch. The HFD used contained 60% (kcal%) fat (diet no. 2127, Provimi Kliba); the low-fat chow diet used contained 10% (kcal%) fat (diet no. 2125, Provimi Kliba). During the WL period both experimental groups received chow diet (diet no. 3437, Provimi Kliba). All animal experiments were approved by the Cantonal Veterinary Office, Zurich.

**Tamoxifen application.** The 4–5-week-old AdipoERCre x NuTRAP mice were gavaged two times with 1 mg of tamoxifen dissolved in corn oil. Tamoxifen was washed out for 2 weeks before starting HFD.

### Physiological measurements

**Glucose tolerance test.** Mice were fasted for 6 h during dark phase before administration of 1 g of glucose per kg body weight by intraperitoneal injection. Blood was collected from the tail vein at 0, 15, 30, 60, 90 and 120 min and blood glucose concentrations were measured using an Accu-Check Aviva glucometer.

**Insulin tolerance test.** Mice were fasted for 6 h during dark phase before administration of 1 U per kg body weight of human insulin (insulin Actrapid HM, Novo Nordisk) by intraperitoneal injection. Blood was collected from the tail vein at 0, 15, 30, 60, 90 and 120 min and blood glucose concentrations were measured using a Accu-Check Aviva glucometer.

**In vivo indirect calorimetry.** Measurements were obtained from one 8-cage and one 16-cage Promethion Core Behavioral System that were in the same room. Mice were habituated to the system for 36 h before measurements were started.

**Live body composition.** Mice were fasted for 6 h during dark phase. Live mouse body composition was measured with a magnetic resonance imaging technique (EchoMRI130, Echo Medical Systems). Fat and lean mass were analysed using EchoMRI 14 software.

**Fasting insulin.** EDTA plasma was isolated from fasted blood samples (fasting 6 h). Insulin was measured with Ultra Sensitive Mouse Insulin ELISA Kit (Crystal Chem, catalogue no. 90080).

**Postprandial insulin.** EDTA plasma (50 μl) was thawed on ice and used in a custom U-PLEX assay (Meso Scale Discovery) according to the

manufacturer's instructions. A Mesoscale SI 2400 was used to read the plate.

**Postprandial leptin.** EDTA plasma (50 μl) was thawed on ice and used in a custom U-PLEX assay (Meso Scale Discovery) according to the manufacturer's instructions. A Mesoscale SI 2400 was used to read the plate.

**Liver triglycerides.** First, 50 mg of frozen liver was homogenized in 1 ml of isopropanol, lysed for 1 h at 4 °C and centrifuged for 10 min at 2,000$g$ at 4 °C. The supernatant was transferred into a new tube and stored at −80 °C until use. Triglyceride levels were measured by mixing 200 μl of reagent R (Monlab, catalogue no. SR-41031) and 5 μl of sample or Cfas calibrator dilutions (Roche, catalogue no. 10759350; lot no. 41009301), then incubating for 10 min while shaking at room temperature and measuring optical density at 505 nm ($OD_{505}$) with a plate reader (BioTek Gen5 Microplate Reader).

### Cell culture experiments

**AT digestion.** AT was minced and digested at 37 °C while shaking in collagenase buffer (25 mM $NaHCO_3$, 12 mM $KH_2PO_4$, 1.3 mM $MgSO_4$, 4.8 mM KCl, 120 mM NaCl, 1.2 mM $CaCl_2$, 5 mM glucose, 2.5% BSA; pH 7.4) using 2 mg of collagenase type II (Sigma-Aldrich, catalogue no. C6885-1G) per 0.25 g of tissue. After 30 min tissues were resuspended, and for ingAT digestion continued for 15 min whereas epiAT was processed immediately. An equal volume of growth medium (DMEM (Gibco, catalogue no. 31966021), 10% FBS (Gibco, catalogue no. 10500-064, Lot no. 2378399H), 1% penicillin-streptomycin (Gibco, catalogue no. 15140-122)) was added and digested tissue was centrifuged for 4 min at 300$g$, and the floating fraction was transferred into a new Falcon tube and kept at 37 °C. The SVF was resuspended in 5 ml of erythrocyte lysis buffer (154 mM $NH_4Cl$, 10 mM $NaHCO_3$, 0.1 mM EDTA, 1% penicillin-streptomycin), incubated at room temperature for 5 min, filtered through a 40 μM mesh filter and centrifuged for 5 min, 300$g$. The SVF was resuspended in growth medium and counted.

**SVF differentiation.** A total of 10,000 cells were plated into one well of a collagen-coated (Sigma-Aldrich, catalogue no. C3867) 96-well plate and kept in culture until they reached confluency, with media change every 48 h. At 2 d post-confluence, medium was changed to induction medium (DMEM, 10% FBS, 1% penicillin-streptomycin, 10 nM insulin (Sigma-Aldrich, catalogue no. I9278), 0.5 mM 3-isobutyl-1-methylxanthin (Sigma-Aldrich, catalogue no. I7018-1G), 1 μM dexamethasone (Sigma-Aldrich, catalogue no. D4902), 1 μM rosiglitazone (Adipogen, catalogue no. AG-CR1-3570-M010)). After 48 h medium was changed to maintenance medium (DMEM, 10% FBS, 1% penicillin-streptomycin, 10 nM insulin). Medium was changed every 48 h for 8 d.

**AdipoRed assay.** The SVF was cultured as described and controls were either kept in growth medium or only maintenance medium without induction. On day 8 after induction, cells were washed twice in PBS, and AdipoRed (Lonza, catalogue no. LZ-PT-7009) reagent was used according to the manufacturer's instructions and read with a plate reader (BioTek Gen5 Microplate Reader).

**Primary adipocyte culture.** Primary floating adipocytes were cultured under membranes according to Harms et al.[68]. Packed adipocytes (30 μl) were seeded onto one membrane and kept in inverted culture for 48 h in maintenance medium (DMEM-F12 (Gibco, catalogue no. 31330095), 10% FBS, 1% penicillin-streptomycin, 10 nM insulin). After 48 h of maintenance, adipocytes were washed and serum and glucose starved overnight in KREBBS-Ringer buffer (120 mM NaCl, 4.7 mM KCl, 1.2 mM $KH_2PO_4$, 1.2 mM $MgSO_4$, 2.5 mM $CaCl_2$, 25 mM HEPES (Lonza, catalogue no. BEBP17-737E), pH 7.4) and 2.5% fat-free BSA (Sigma-Aldrich, catalogue no. A6003).

**Glucose uptake.** Glucose uptake from primary adipocytes was measured using the Glucose Uptake-Glo Assay Kit (Promega, catalogue no. J1341) according to the manufacturer's instructions. Adipocytes were preincubated with 5 nM insulin for 15 min before 2-deoxy-D-glucose was added at 1 mM final concentration. Protein concentration was measured using a Pierce 660 nm Protein Assay Kit (Thermo Fisher, catalogue no. 22662) and the Ionic Detergent Compatibility Reagent (Thermo Fisher, catalogue no. 22663). Both assays were read with a plate reader (BioTek Gen5 Microplate Reader).

**C16 uptake.** Starved adipocytes were incubated with 5 nM BODIPY-palmitate (Thermo Fisher, catalogue no. D3821) in the presence of 10 nM insulin for 1 h. Subsequently, adipocytes were washed twice and lysed in 200 μl of RIPA buffer. Then, 100 μl of lysate was used to measure BODIPY signal. Diluted lysate was used to measure protein concentration using a DC Protein Assay Kit II (Bio-Rad Laboratories, catalogue no. 5000112) for normalization. Both assays were read with a plate reader (BioTek Gen5 Microplate Reader).

**Histology.** Tissues were collected, fixed in 4% PBS-buffered formalin for 72 h at 4 °C and stored in PBS at 4 °C. Following paraffin embedding, tissues were sent to the pathology service centre at Instituto Murciano de Investigación Biosanitaria Virgen de la Arrixaca for sectioning, trichrome staining, haematoxylin and eosin staining, and imaging. Tissues from two independent experiments were sent for sectioning.

**Adipocyte size quantification.** Images of ingAT and epiAT were taken with 3DHISTECH Slide Viewer 2 and then analysed with Adiposoft[69] using Fiji ImageJ[70]. Five to ten images were taken of each section belonging to a biological replicate ($n = 4$).

### Sample processing and library preparation

**Isolation of nuclei from mouse tissue.** Nuclei were isolated from snap-frozen epiAT in ice-cold Nuclei Extraction Buffer (Miltenyi, catalogue no. 130-128-024) supplemented with 0.2 U μl$^{-1}$ recombinant RNase Inhibitor (Takara, catalogue no. 2313) and 1× cOmplete EDTA-free Protease Inhibitor (Roche, catalogue no. 5056489001) using the gentleMACS Octo Dissociator (Miltenyi, catalogue no. 130-096-427), using C-tubes (Miltenyi, catalogue no. 130-093-237). Nuclei were subsequently filtered through a 50 μm cell strainer (Sysmex, catalogue no. 04-0042-2317) and washed two times in PBS-BSA (1% w/v) containing 0.2 U μl$^{-1}$ RNase inhibitor. For snRNA-seq, five mice were pooled per condition.

**Isolation of nuclei from human tissue.** Nuclei were isolated from snap-frozen human AT (10–50 mg) in ice-cold Nuclei Extraction Buffer (Miltenyi, catalogue no. 130-128-024) supplemented with 1 U μl$^{-1}$ recombinant RNase Inhibitor (Takara, catalogue no. 2313), 1× cOmplete EDTA-free Protease Inhibitor (Roche, catalogue no. 5056489001) and 10 mM sodium butyrate using the gentleMACS Octo Dissociator (Miltenyi, catalogue no. 130-096-427), using C-tubes (Miltenyi, catalogue no. 130-093-237).

The nuclei suspension was filtered through a 50 μm strainer, supplemented with PBS-BSA (1% w/v) containing 1× protease inhibitor and RNase inhibitor and centrifuged at 4 °C, at 500$g$ for 10 min. The nuclei pellet was resuspended in 1 ml of PBS-BSA (1%, w/v) supplemented with RNase inhibitor (0.5 U μl$^{-1}$) and 1× protease inhibitor and was transferred into a new 1.5 ml tube.

**snRNA-seq of AT.** Nuclei were counted using a haemocytometer and Trypan blue, concentration was adjusted to approximately 1,000 nuclei per μl and they were loaded onto a G-chip (10x Genomics, catalogue no. PN-1000127). Single-cell gene expression libraries were prepared using the Chromium Next GEM Single Cell 3′ v3.1 kit (10x Genomics) according to the manufacturer's instructions. To accommodate for low

RNA content, two cycles were added to the complementary DNA amplification PCR. Libraries were pooled equimolecularly and sequenced in PE150 (paired-end 150) mode on a NovaSeq 6000 with about 40,000 reads per nucleus at Novogene or using a NovaSeqX at the Functional Genomics Center, Zurich.

**Paired TRAP–seq, CUT&Tag and ATAC–seq.** Paired TRAP–seq, CUT&Tag and ATAC–seq protocols were developed on the basis of published protocols[67,71–74].

**Ribosome and nuclei isolation.** Nuclei and ribosomes were isolated from snap-frozen epiAT from AdipoERCre x NuTRAP mice in ice-cold Nuclei Extraction Buffer (Miltenyi, catalogue no. 130-128-024) supplemented with 0.2 U μl$^{-1}$ recombinant RNase Inhibitor (Takara, catalogue no. 2313), 1× cOmplete EDTA-free Protease Inhibitor (Roche, catalogue no. 5056489001) and 10 mM sodium butyrate using the gentleMACS Octo Dissociator (Miltenyi, catalogue no. 130-096-427), using C-tubes (Miltenyi, catalogue no. 130-093-237). The nuclei suspension was filtered through a 50 μm strainer and centrifuged at 4 °C, 500g for 5 min. The supernatant was transferred into a new tube and supplemented with 2 mM dithiothreitol, 100 μg ml$^{-1}$ cycloheximide (Sigma-Aldrich, catalogue no. 01810) and 1 mg ml$^{-1}$ sodium heparin (Sigma-Aldrich, catalogue no. H3149-10KU) and kept on ice. The nuclei pellet was resuspended in 1 ml of PBS-BSA (1%, w/v) supplemented with 0.2 U μl$^{-1}$ RNase inhibitor, 1× cOmplete EDTA-free Protease Inhibitor and 10 mM sodium butyrate and transferred into a new 1.5 ml tube. Nuclei were centrifuged and subsequently bound to Dynabeads MyOne Streptavidin C1 beads (Thermo Fisher, catalogue no. 65002) for 30 min at 4 °C followed by three washes with PBS-BSA (1% w/v).

**TRAP–seq.** Per sample, 25 μl of GFP-Trap Magnetic Agarose Beads (ChromoTEK, catalogue no. gtma-20) were washed in 2 ml of polysome lysis buffer (50 mM TRIS-HCl pH 7.5, 100 mM NaCl, 12 mM MgCl$_2$, 1% Igepal CA-630 (Sigma-Aldrich, catalogue no. I8896), 1× protease inhibitor). The supernatant was mixed with the beads and incubated at 4 °C on a rotator for 1–2 h. Subsequently, tubes were put on a magnetic stand and the supernatant was removed. The beads were washed three times with polysome lysis buffer supplemented with 2 mM dithiothreitol (Sigma-Aldrich, catalogue no. D0632-10G), 100 μg ml$^{-1}$ cycloheximide (Sigma, catalogue no. D0632-10G) and 1 mg ml$^{-1}$ sodium heparin (VWR, catalogue no. ACRO411210010) and resuspended in 1 ml Trizol (Thermo Fisher, catalogue no. 15596). Trizol preserved samples were kept at −80 °C until RNA isolation. RNA was isolated by adding 200 μl of chloroform (Sigma-Aldrich, catalogue no. 288306) to samples, followed by shaking and centrifugation at 4 °C, 12,000g for 15 min. The aqueous phase was transferred into a new tube and RNA was isolated and DNase treated with the RNA Clean and Concentrator-5 kit (Zymo Research, catalogue no. R1016), following the manufacturer's instructions.

RNA libraries were prepared by performing reverse transcription and template switching using Maxima H Minus reverse transcriptase (Thermo Fisher, catalogue no. EP0753), a template switch oligo and an oligodT primer to generate full-length cDNA. cDNA was amplified using the KAPA Hotstart 2x ReadyMix (Roche Diagnostics, catalogue no. 7958935001). Then, 1–3 ng of cDNA was tagmented using 1.3 μg of Tn5 and amplified using KAPA HiFi plus dNTPs (Roche Diagnostics, catalogue no. 07958846001) and the following PCR settings: 72 °C 5 min, 98 °C 30 s, 10 cycles of 98 °C for 10 s, 63 °C for 30 s, 72 °C for 1 min, hold at 4 °C. Libraries were quantified using the KAPA library quantification kit (Roche Diagnostics, catalogue no. 079602), and sequenced in PE150 mode on a NovaSeq 6000 at Novogene.

**CUT&Tag.** CUT&Tag was performed as previously described with minor adjustments[74,75]. All buffers were supplemented with 1 x cOmplete EDTA-free Protease Inhibitor and 10 mM sodium butyrate.

Briefly, nuclei bound to beads were aliquoted into 96-well LoBind plates (Eppendorf, catalogue no. 0030129547) and incubated with primary antibodies—anti-H3K4me3 (abcam, catalogue no. ab8580), anti-H3K27me3 (Cell Signaling Technology, catalogue no. C36B11), anti-H3K27ac (abcam, catalogue no. ab4729), anti-H3K4me1 (abcam, catalogue no. ab8895)—overnight at 4 °C. With the plate on a magnet, the primary antibody solution was removed, and the beads were resuspended in secondary antibody solution (guinea pig anti-rabbit IgG (antibodies-online, catalogue no. ABIN101961)) and incubated at room temperature. pA-Tn5 was bound to antibodies, and transposition was performed at 37 °C and stopped using TAPS-Wash solution. Nuclei were lysed and pA-Tn5 decrosslinked using SDS-release solution. PCR was performed using KAPA HiFi plus dNTPs (Roche Diagnostics, catalogue no. 07958846001) with the following PCR settings: 72 °C 5 min, 98 °C 30 s, 15 cycles of 98 °C 10 s, 63 °C 30 s, and 72 °C final extension for 1 min, hold at 4 °C.

**ATAC–seq.** Beads with nuclei were resuspended in ATAC–seq solution (10 mM TAPS pH 8.5, 5 mM MgCl$_2$, 10% DMF (Sigma-Aldrich, catalogue no. D4551), 0.2 μg μl$^{-1}$ transposase (Tn5)) and incubated at 37 °C for 30 min. Thereafter, 100 μl of DNA binding buffer (Zymo Research, catalogue no. D4003-1) was added and samples were stored at −20 °C. Then, DNA was extracted using Zymo DNA Clean and Concentrator-5 (Zymo Research, catalogue no. D4004). Library amplification was performed using KAPA HiFi plus dNTPs (Roche Diagnostics, catalogue no. 07958846001) and the following PCR settings: 72 °C 5 min, 98 °C 30 s, 10 cycles of 98 °C 10 s, 63 °C 30 s, 72 °C 1 min, hold at 4 °C.

Both ATAC–seq and CUT&Tag libraries were cleaned using SPRI beads, eluted in nuclease-free water and pooled equimolecularly after library quantification using the KAPA library quantification kit (Roche Diagnostics, catalogue no. 079602). Libraries were sequenced in PE150 mode on a NovaSeq 6000 at Novogene.

## Sequencing data processing

**snRNA-seq data processing and analysis. Data integration and differential expression analysis for mouse snRNA-seq.** The 10x Genomics Cell Ranger v.6.1.2 pipeline was used for demultiplexing, read alignment to reference genome mm10-2020A (10x Genomics), barcode processing and unique molecular identifier (UMI) counting with Include introns argument set to 'True'. The R package Seurat v.4.1.0 (ref. 76) was used to process, integrate and analyse datasets. scDblFinder[77] was used to identify and remove doublets. Nuclei with unique feature counts less than 500 or greater than 3,000 and UMI counts greater than 40,000 were discarded during quality control (Extended Data Fig. 11a). Highly expressed genes such as mitochondrial genes, pseudogenes and *Malat1* were excluded from the count matrix before normalization. SoupX[78] was used to estimate potential ambient RNA contamination in all samples, but no sample required any correction. Samples were normalized using sctransform and integrated using the CCA (canonical correlation analysis) method built into Seurat. Filtered, normalized and integrated nuclei data were clustered by using the Louvain algorithm with a resolution of 0.4 using the first 30 principal components. Cluster markers were identified on the basis of differential gene expression analysis (Wilcoxon rank-sum test with |log$_2$FC| > 0.25 and adjusted $P$ < 0.05). Clusters were then annotated on the basis of known markers from literature[34,36,37,46,79,80]. Additionally, our manual cluster annotation was confirmed by reference mapping against a reference male mouse epiAT[34] dataset (Extended Data Fig. 11b,c). Differential expression analysis (Wilcoxon rank-sum test with |log$_2$FC| > 0.5 and adjusted $P$ < 0.01) per cell type between different conditions was done using the FindMarkers function from Seurat. Differential expression analysis hits were intersected with a list of epigenetic modifier genes (see the Source Data to Extended Data Fig. 8) to investigate their expression dynamics. For visualization of snRNA-seq data we used the R package SCpubr v.1 (ref. 81).

**Data integration and differential expression analysis for human snRNA-seq.** The 10x Genomics Cell Ranger v.7.2.0 pipeline was used for demultiplexing, read alignment to reference genome GRCh38-2020-A (10x Genomics), barcode processing and UMI counting, with force cells set to 10,000. The R package Seurat v.4.1.0 (ref. 76) was used to process, integrate and analyse datasets. scDblFinder[77] was used to identify and remove doublets. Nuclei with unique feature counts <300 or >4,000 (LTSS) / 6,000 (NEFA), UMI counts >15,000 (LTSS) / 25,000 (NEFA) and mitochondrial gene counts greater than 5% were discarded during quality control (Extended Data Fig. 12). SoupX[78] was used to estimate and correct for potential ambient RNA contamination in all samples. Samples were normalized using sctransform and integrated using the CCA method built into Seurat. Filtered, normalized and integrated nuclei data were clustered by using Louvain algorithm using the first 30 principal components. For each study, the cluster resolution was determined using the R package clustree[82]. Cluster markers were identified on the basis of differential gene expression analysis (Wilcoxon rank-sum test with $|\log_2 FC| > 0.25$ and adjusted $P < 0.01$). Clusters were then annotated on the basis of known markers from literature[34–37,83]. Additionally, our manual cluster annotation was confirmed by reference mapping against reference human white AT atlas[34] (Extended Data Figs. 2 and 3). For each AT depot, adipocytes from two studies were integrated together using the first 20 principal components following the steps as mentioned above. Differential expression analysis (Wilcoxon rank-sum test with $|\log_2 FC| > 0.5$ and adjusted $P < 0.01$) per cell type between different conditions was done using the FindMarkers function from Seurat. Differential expression analysis hits were validated using MAST and likelihood-ratio tests using the FindMarkers function from Seurat. For visualization of snRNA-seq data, we used the R package SCpubr v.1 (ref. 81).

**SNP-based demultiplexing of human snRNA-seq datasets.** To perform SNP calling and demultiplexing on the pooled samples, cellsnp-lite[84] was first used to call SNPs on a cell level using the 1000 Genomes-based reference variant call file for hg38 at a resolution of 7.4 million SNPs. SNPs with less than 20 counts and a minor allele frequency of less than 10% were filtered out, as per the developer recommendations. Finally, the tool vireo[85] was used to demultiplex the pooled data using the cellsnp-lite-derived genotype information.

For each donor, we analysed tissue composition and removed nuclei belonging to donors in the case in which no nuclei were assigned as adipocytes (one case in NEFA) or more than 50% or nuclei were assigned as B cells (one case in MTSS; lean donor) after correspondence with surgeons.

**Transcriptional retention.** DEGs from obese and WL cells from mouse and human were overlayed, respectively. A DEG was considered restored if it was no longer deregulated in WL cells when compared with controls. If not restored, we considered a DEG part of a transcriptional memory. Clusters identified as similar cell types (for example, three clusters of endothelial cells) were merged for DEG quantification but not differential expression analysis itself. For human snRNA-seq, only cell types for which we obtained at least 30 cells per donor were considered for the retention analysis. T cells were not included in differential expression analysis or transcriptional retention analysis. For integrated human adipocyte differential expression analysis quantification, noncoding transcripts were excluded.

**TRAP-seq.** Quality control of the raw reads was performed using FastQC v.0.11.9. Raw reads were trimmed using TrimGalore v.0.6.6 (https://github.com/FelixKrueger/TrimGalore). Filtered reads were aligned against the reference mouse genome assembly mm10 using HISAT2 v.2.2.1. Raw gene counts were quantified using the featureCounts[86] program of subread v.2.0.1. Differential expression analysis was performed using the R package EdgeR[87], with $|\log_2 FC| \geq 1$ and nominal $P < 0.01$ as cut-offs.

**CUT&Tag and ATAC-seq data processing and analysis.** Quality control of CUT&Tag and ATAC-seq data and generation of bedgraph files was performed as described previously[75]. Peaks were called from CUT&Tag sequencing and ATAC-seq libraries on individual bedgraph files using SEACR[88] v.1.3 in stringent mode with a peak calling threshold of 0.01. Peaks overlapping with mouse blacklist regions[89] were filtered out. Called peaks were annotated using the R package ChIPSeeker[90]. Peak fold enrichment against genomic features was calculated using the formula: Σ(base pair (bp) overlap) × genome_size/[Σ(bp hPTM peak) × Σ(bp genomic feature)]. Genomic features tracks were downloaded from ENCODE using the R package annotatr[91]. Visual quality control of bam files was performed with Seqmonk[92]. Called peaks were combined to generate a union peak list and quantified using the R package chromVAR[93] v.1.16, generating a raw peak count matrix.

**MOFA.** MOFA[50,94] was run to identify the driving variation source across all conditions using all data modalities. For each modality, the top 3,000 variable features (genes or peaks) between all samples were selected using the R package DESeq2 (ref. 95) and used as input to train the MOFA model. The trained MOFA model represented data variability in terms of five latent factors, which were further explored and visualized.

**Generation of enhancer tracks of adipocytes.** Adipocyte chromatin states were identified using ChromHMM v.1.22 (ref. 96) in concatenated mode with binned bam files (200-bp bins) from each condition combining all hPTMs and ATAC-seq. After final model selection[75] with eight chromatin states and emission parameter calculation of hPTMs and ATAC-seq, chromatin state fold enrichment was performed against genomic features and ENCODE candidate *cis*-regulatory elements. Enhancer states were selected on the basis of genomic localization and hPTM enrichment. Subsequently, an enhancer track was generated per condition and merged for differential analysis.

**Differential analysis of hPTMs and ATAC-seq. Promoters.** Promoters were defined using the getPromoters function from ChIPSeeker with *TxDb.Mmusculus.UCSC.mm10.knownGene* as input and setting the TSSRegion to c(-2000, 2000). Peaks overlapping with promoters were extracted using the annotatePeak function from ChIPseeker[90] by selecting peaks annotated as promoters. For differential analysis, our raw peak count matrix was filtered for these promoter regions and counts were aggregated at gene level. Differential analysis of the same hPTM between two conditions was performed using the R package EdgeR[87] with nominal $P < 0.01$ and $|\log_2 FC| > 1$ as cut-offs.

**Enhancers.** ChromHMM was used to identify regions in the genome that were marked by H3K4me1, H3K27ac and open (ATAC-seq) but not enriched for H3K4me3 and that were not promoters (Extended Data Fig. 9b–e). States 6 and 5 were selected as enhancer regions on the basis of their genomic locations (distal enhancer elements) (Extended Data Fig. 9b–e).

Our raw peak count matrix was filtered for enhancer regions defined by chromHMM, and peaks around the TSS (±2,000 bp) were discarded. Linkage of putative enhancers to genes was done using the R package ChIPSeeker by selecting the closest gene (TSS or gene body) within 20,000 bp distance. Putative enhancers farther away than 20,000 from a TSS or gene body were not linked to any gene and were discarded from downstream GSEA.

For each hPTM, the raw filtered peak matrices were log-normalized using the R package EdgeR and Pearson's correlation coefficient was computed using the cor function from the R package stats v.3.6.2.

Differential analysis of the same hPTM between two conditions was performed using the R package EdgeR with nominal FDR < 0.05 and $|\log_2 FC| > 1$ as cut-offs.

**PCA.** Raw gene and promoter/enhancer-specific peak count matrices were log-normalized using the R package EdgeR. PCA of the normalized count matrices was performed using the prcomp function of R package stats v.3.6.2.

**GSEA.** GSEA was performed using the R package enrichR[97–99]. For generation of heatmaps summarizing GSEA across cell types, significantly

enriched terms were selected using the adjusted *P* value (<0.01) and the combined.score (enrichment score) was scaled and visualized.

## Visualization
R v.4.2, GraphPad Prism v.9.5.1 and Seqmonk v.1.48.1 were used to generate plots and Affinity Designer and Publisher were used to adjust plots for clarity (for example, colour schemes).

## Statistical analysis of physiological parameters from mice
GraphPad Prism v.9.5.1 was used to analyse physiological data from mice. Each dataset of physiological parameters was tested for normality using the Shapiro–Wilk test. On the basis of the results, parametric or non-parametric tests were used to compare experimental with age-matched control groups. Tests are indicated in figure legends and the Source Data.

## Reporting summary
Further information on research design is available in the Nature Portfolio Reporting Summary linked to this article.

## Data availability
All mouse sequencing data that support the findings of this study have been deposited on GEO, with the accession code GSE236580. Human snRNA-seq data from the MTSS and LTSS cohorts are available upon request from F.v.M., C.W. and M.B. Human snRNA-seq data from the NEFA cohort are available upon request from F.v.M., N.K. and M.R. Analysis code for human and mouse data is available on GitHub and Zenodo[100]. An interactive snRNA-seq data browser link and links to interactive tables with results of differential gene expression and epigenetic analysis are available on GitHub (https://github.com/vonMeyennLab/AT_memory). Source data are provided with this paper.

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

**Acknowledgements** We thank all members of the von Meyenn group and the Wolfrum group for helpful discussions and support. We also thank M. Sütö, C. Sert, M. Klug, C. Leuzinger, C. Kellenberger, L. Maak and R. von Wartburg for help with animal experiments and handling; J. P. A. de Sousa for help with computational pipelines; E. Masschelein and T. Dahlby for help with the metabolic cage measurements; E. Seelig for help with acquisition of human samples and scientific support; J. Bohacek and R. Waag for NuTRAP mice; and the Protein Production and Structure Core Facility at EPFL for the production and purification of pA-Tn5 and Tn5, especially K. Lau, F. Pojer and M. Francois. This work was supported by ETH Zurich core funding (F.v.M.), a European Research Council Starting Grant (no. 803491, BRITE to F.v.M.), the Basel Research Centre for Child Health (Multi-Investigator Project 2020 to F.v.M.), the Deutsche Forschungsgemeinschaft (Project no. 209933838–SFB 1052 (project B1) to M.B.), the Margareta af Ugglas foundation (M.R.), the Swedish Research Council (M.R., N.M., including an establishing grant to L.M.), a European Research Council Synergy Grant (no. 856404, SPHERES to M.R.), the Novo Nordisk Foundation (including the MeRIAD consortium grant no. 0064142 to M.R., and no. NNF20OC0061149 to N.M.), the Knut and Alice Wallenberg's Foundation (Wallenberg Clinical Scholar to M.R.), the Center for Innovative Medicine (M.R.), the Swedish Diabetes Foundation (M.R.), the Stockholm County Council (M.R.), the Strategic Research Program in Diabetes at Karolinska Institutet (M.R.) and the European Foundation for the Study of Diabetes (Future Leaders award to N.M.). L.M. was funded by a postdoctoral grant from the Swedish Society for Medical Research.

**Author contributions** D.C.-C. and L.C.H. performed mouse experiments and made libraries from mouse tissues. L.C.H. performed experiments with human samples and cell culture experiments. L.C.H., A.G. and D.C.-C. analysed data. K.M. assisted with dissection, cell culture experiments and artwork. E.G. assisted with dissection. M.B., C.W. and M.R. supervised the clinical studies and M.B., A.H., L.M., M.R. and N.M. provided human AT samples and clinical data. F.N. SNP-demultiplexed snRNA-seq data. H.D. and W.S. collected additional metadata of human samples. M.B., C.W., M.R. and N.M. supervised the human studies and human AT sample collections. L.C.H., D.C.-C. and F.v.M. conceptualized the study, interpreted the data and wrote the manuscript. All authors contributed to editing and reviewing the manuscript and approved its publication.

**Funding** Open access funding provided by Swiss Federal Institute of Technology Zurich.

**Competing interests** M.B. received honoraria as a consultant and speaker from Amgen, AstraZeneca, Bayer, Boehringer-Ingelhiem, Lilly, Novo Nordisk and Sanofi. M.R. received honoraria as a consultant and speaker from AstraZeneca, Boehringer-Ingelheim, Lilly, Novo Nordisk and Sanofi. The other authors declare no competing interests.

**Additional information**
**Correspondence and requests for materials** should be addressed to Ferdinand von Meyenn.

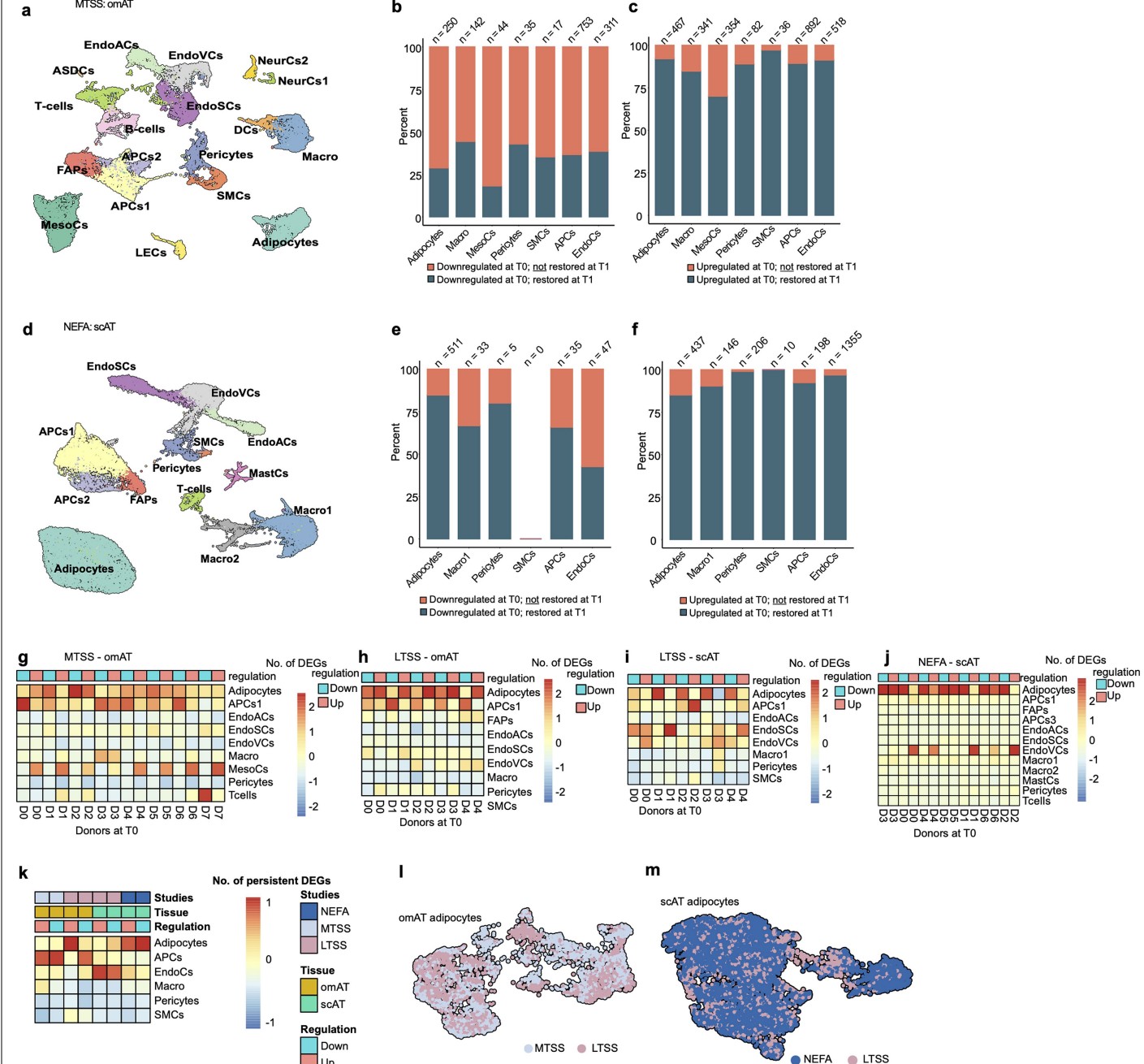

**Extended Data Fig. 1 | Human AT retains cellular transcriptional changes after bariatric surgery induced WL. a**, UMAP of 19,494 nuclei representing omAT pools from lean subjects ($n = 5$; 1 male, 4 females) and paired omAT from T0 and T1 ($n = 8$ each; 2 males, 6 females) from the MTSS study. **b,c** Proportion of retained transcriptional changes in highly abundant cell types of MTSS omAT. **d**, UMAP of 31,721 nuclei representing scAT pools from lean subjects ($n = 8$; 8 females) and paired scAT from T0 and T1 ($n = 7$ each; 7 females) from the NEFA study. **e,f** Proportion of retained transcriptional changes in highly abundant cell types of NEFA scAT. **g-j** Number of upregulated and downregulated DEGs per cell type obese donor scaled by column at T0 for omAT (left) and scAT (right) from MTSS, LTSS and NEFA studies. **k**, Number of persistently deregulated genes from T0 to T1 per cell type across AT pools from all studies.

**l**, UMAP of 4,958 nuclei representing adipocytes from MTSS omAT and LTSS omAT (total lean $n = 10$; total T0/T1 $n = 13$). **m**, UMAP of 13,231 nuclei representing adipocytes from NEFA scAT and LTSS scAT (total lean $n = 13$; total T0/T1 $n = 12$). Wilcoxon Rank Sum test, with adjusted p-value < 0.01 by the Bonferroni correction method and FC > ±0.5 was used for DEG identification in b, c, e-k. APCs, adipocyte progenitor cells; ASDCs, AXL+ dendritic cells; DCs, dendritic cells; EndoCs, endothelial cells; EndoACs, arteriolar EndoCs; EndoSCs, stalk EndoCs; EndoVCs, venular EndoCs; LECs, lymphatic endothelial cells; FAPs, fibro-adipogenic progenitors; Macro, macrophages; MastCs, mast cells; MesoCs, mesothelial cells; NeurCs, neuronal like cells; SMCs, (vascular) smooth muscle cells.

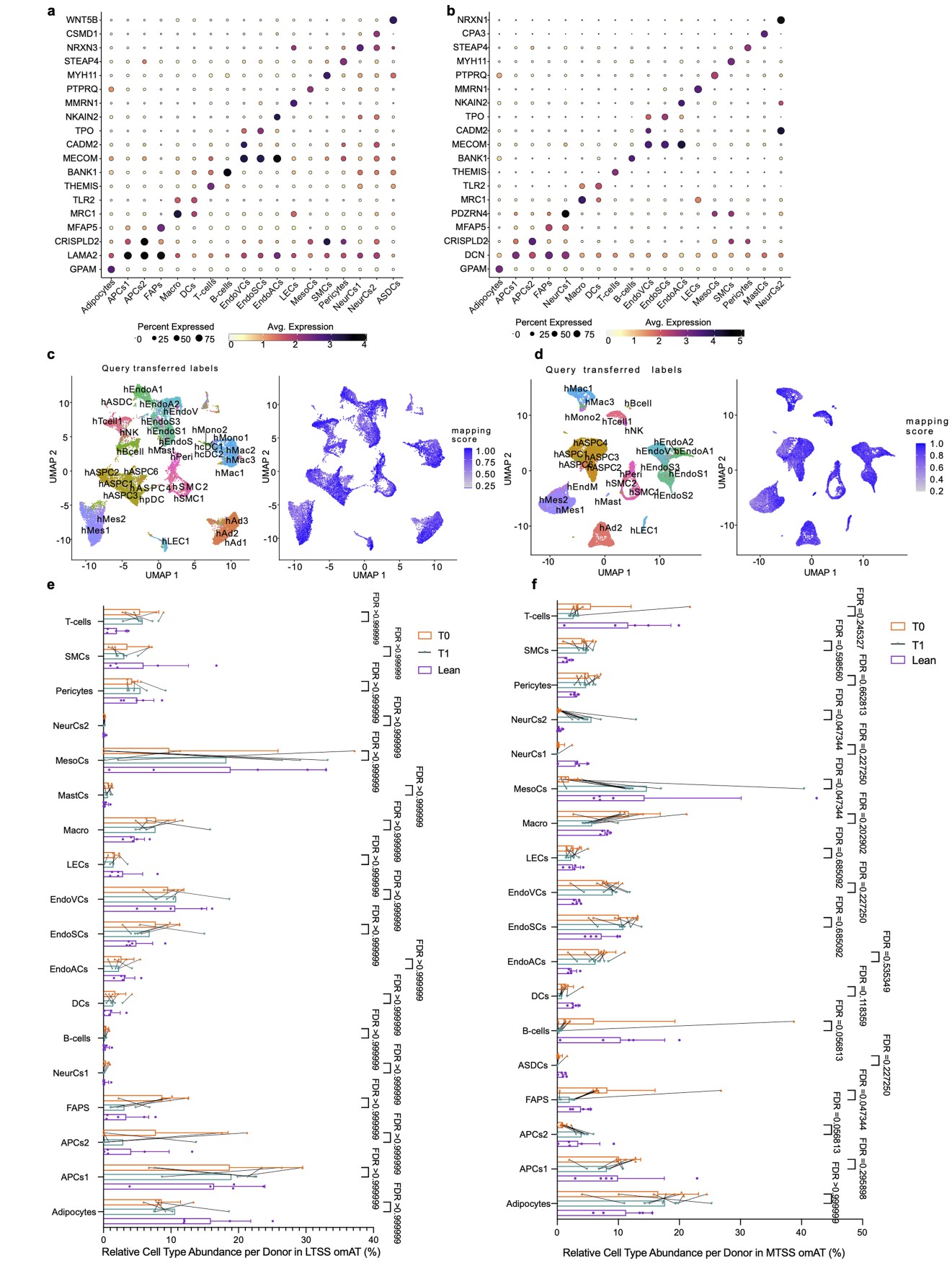

**Extended Data Fig. 2 | See next page for caption.**

**Extended Data Fig. 2 | Characterization of omAT composition. a**,**b**, Cluster markers used for annotating cell clusters in human omAT of the MTSS (left) and LTSS (right) study. **c**,**d**, UMAP visualization representing omAT pools from the MTSS study (c) and LTSS study (d) coloured by predicted cell subtypes from the Emont et al. visceral AT dataset from Caucasian individuals. Feature plots showing reference mapping scores illustrating how well omAT dataset maps to the Emont et al. dataset. **e**,**f**, Relative cell type abundance in omAT per condition and tissue donor of the LTSS (e) and MTSS (f) study. Lines connecting dots indicate paired samples. Significance between T0 and T1 for e-f was calculated using paired multiple Wilcoxon tests with Benjamini, Krieger and Yekutieli post hoc test for multiple comparisons. Error bars represent s.d.

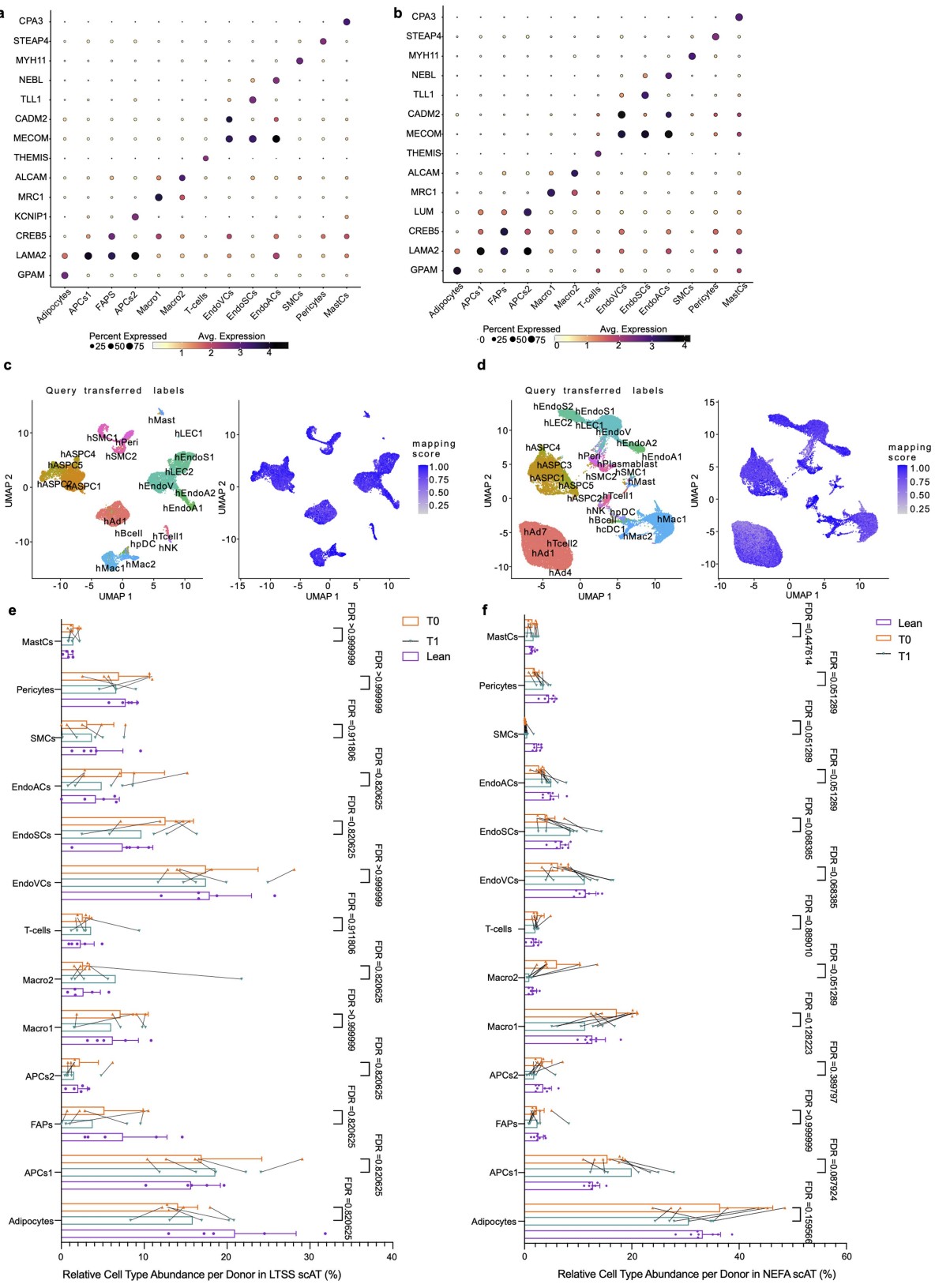

**Extended Data Fig. 3 | Characterization of scAT composition. a,b**, Cluster markers used for annotating cell clusters in human scAT of the LTSS (left) and NEFA (right) study. **c,d**, UMAP visualization representing scAT pools from the LTSS study (c) and NEFA study (d) coloured by predicted cell subtypes from the Emont et al. subcutaneous AT dataset from Caucasian individuals. Feature plots showing reference mapping scores illustrating how well scAT dataset maps to the Emont et al. dataset. **e,f**, Relative cell type abundance in scAT per condition and tissue donor of the LTSS (e) and NEFA (f) study. Lines connecting dots indicate paired samples. Significance between T0 and T1 for e-f was calculated using paired multiple Wilcoxon tests with Benjamini, Krieger and Yekutieli post hoc test for multiple comparisons. Error bars represent s.d.

**a** Persistently downregulated in omAT adipocytes MTSS study

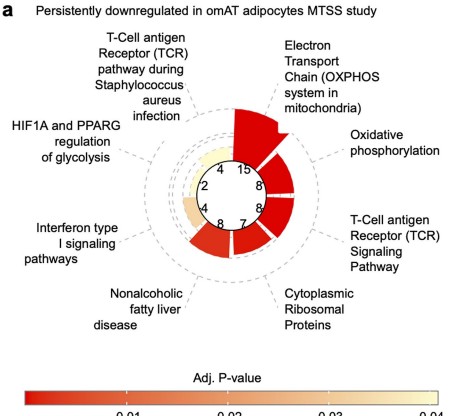

**b** Persistently downregulated in omAT adipocytes LTSS study

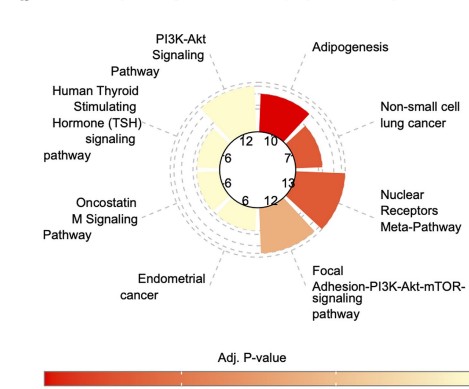

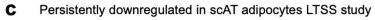

**c** Persistently downregulated in scAT adipocytes LTSS study

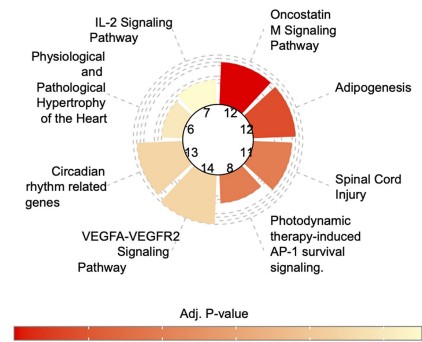

**d** Persistently downregulated in scAT adipocytes NEFA study

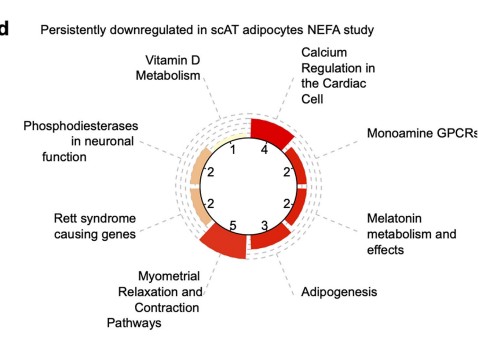

**e** Persistently upregulated in omAT adipocytes          MTSS study

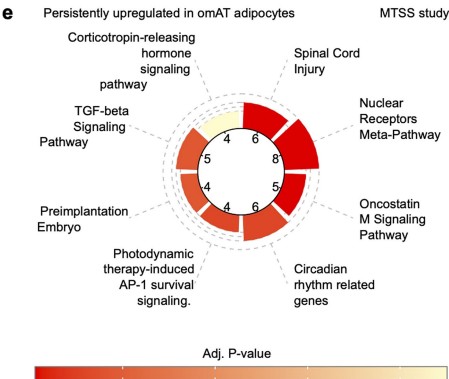

**f** Persistently upregulated in omAT adipocytes LTSS study

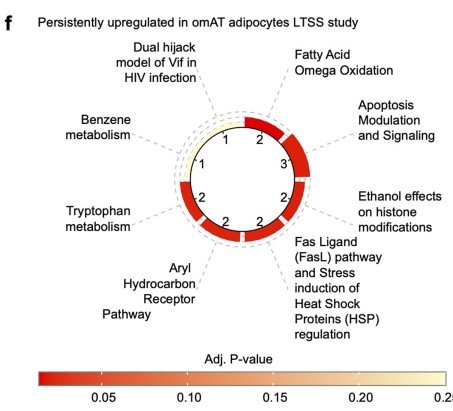

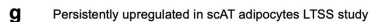

**g** Persistently upregulated in scAT adipocytes LTSS study

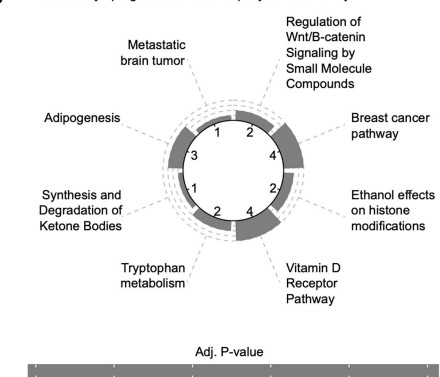

**h** Persistently upregulated in scAT adipocytes NEFA study

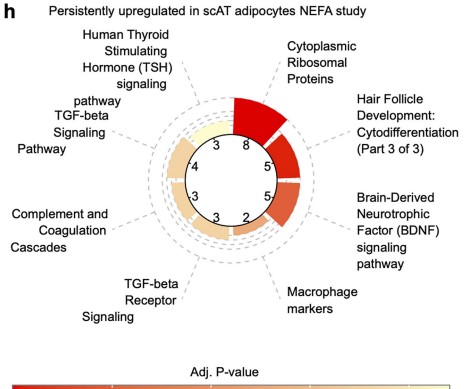

**Extended Data Fig. 4** | See next page for caption.

**Extended Data Fig. 4 | GSEA of retained DEGs in adipocytes. a,b**, Top (significant) persistently downregulated (memory) pathway terms in omental adipocytes of the MTSS (a) and LTSS (b) study based on Wikipathways database. **c,d**, Top (significant) persistently downregulated (memory) pathway terms in subcutaneous adipocytes of the LTSS (c) and NEFA (d) study based on Wikipathways database. **e,f**, Top (significant) persistently upregulated (memory) pathway terms in omental adipocytes of the MTSS (e) and LTSS (f) study based on Wikipathways database. **g,h**, Top (significant) persistently downregulated (memory) pathway terms in subcutaneous adipocytes of the LTSS (g) and NEFA (d) study based on Wikipathways database. In g enrichment is not significant. Significance was calculated using Fisher's exact test, with adjusted P-value < 0.05 by the Benjamini-Hochberg method for correction.

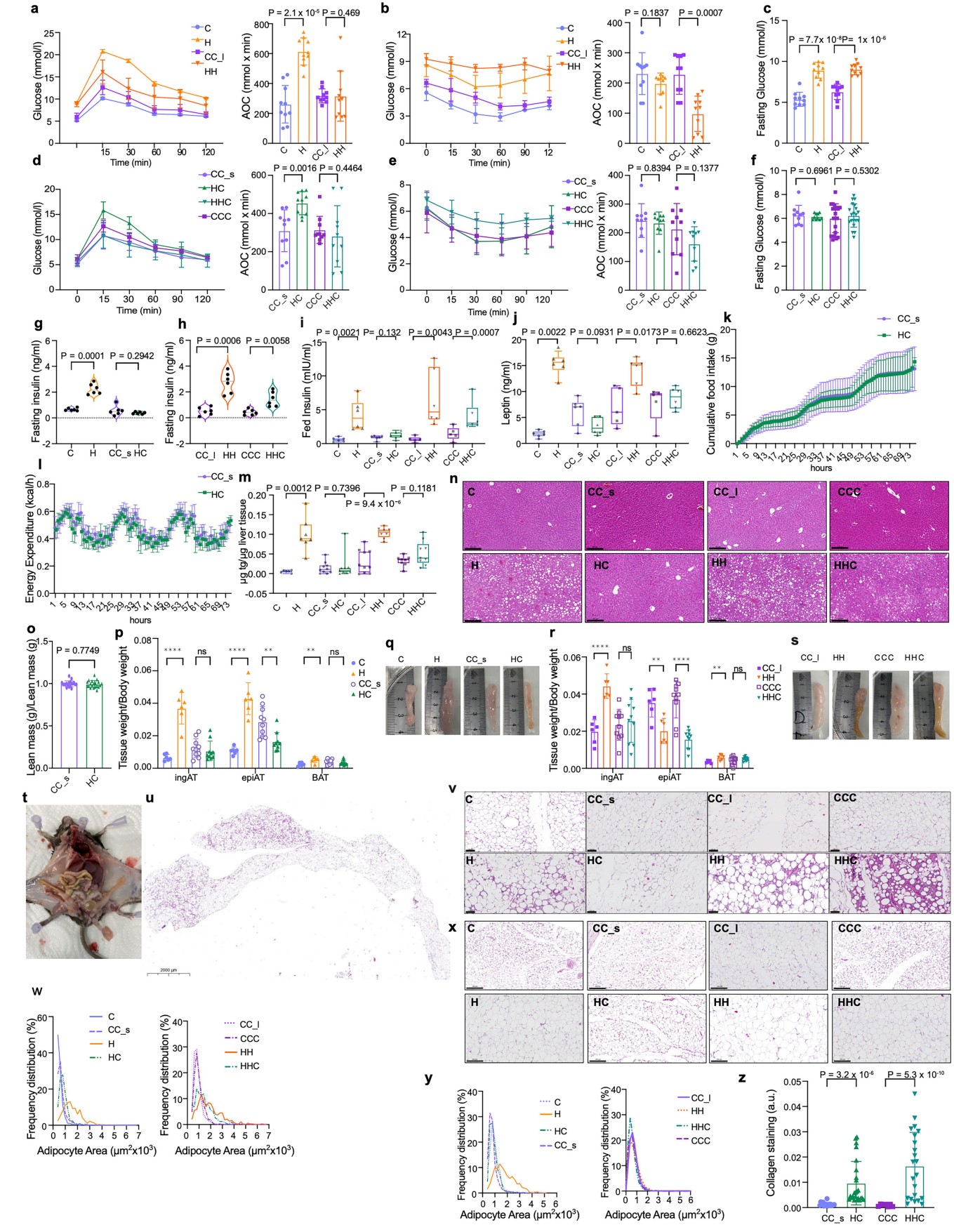

**Extended Data Fig. 5** | See next page for caption.

**Extended Data Fig. 5 | Weight loss largely resolves obesity induced physiological changes in mice.** Data from mouse experiments. For *n*<11, each dot represents an biological replicate. Data from 2-3 independent experiments. **a**, Glucose tolerance tests (GTTs) and area of the curve (AOC) for GTTs; (*n* = 10 each). **b**, Insulin tolerance tests (ITTs) and AOC for ITTs (*n* = 10 each). **c**, Fasting blood glucose (*n* = 10 each). **d**, GTTs and AOCs for GTTs; (*n* = 10 each from 2 independent experiments). **e**, ITTs and AOC for ITTs (*n* = 10 each from 2 independent experiments). **f**, Fasting blood glucose (CC_s&HC: *n* = 10 each, CCC&HHC: *n* = 20 each; from 2 independent experiments). **g,h**, Fasting insulin levels (*n* = 6). **i,j** Postprandial insulin and leptin levels. C, H, CC_s, HC, HH, HHC: *n* = 6 each; CC_l, CCC: *n* = 5 each. Boxplot represents minimum, maximum and median. **k**, Cumulative food intake from HC and CC_s mice in the last 3 days of WL chow diet feeding. (*n* = 10 mice each). **l**, Energy expenditure of HC and CC_s mice in the last 3 days of WL chow diet feeding. (*n* = 10 mice each). **m**, Liver triglycerides (tg) per μg liver tissue (C&H: *n* = 6, CC_s: *n* = 10, HC, CC_l, HHC: *n* = 9, HH&CCC: *n* = 8). Boxplot represents minimum, maximum and median. **n**, Haematoxylin and eosin (HE) staining liver sections, 20x magnification. Scale bar, 200 μm. **o**, Lean mass of HC and CC_s mice relative to lean mass measured at C and H timepoints of the same mice (right) (n = 19 each). **p**, weights of ingAT, epiAT and BAT, normalized to body weight (C&H: *n* = 6, HC&CC_s: *n* = 10, from 2 experiments). **q**, Representative photos of epiAT depots. Ruler is in cm. **r**, Weights of ingAT, epiAT and BAT, normalized to body weight (HH&CC_l: *n* = 6, CCC&HHC: *n* = 10). **s**, Representative photos of epiAT depots. Ruler is in cm. **t**, Representative photo of a HHC mouse. **u**, Representative image of a histological and HE stained section of a whole epiAT depot from a HHC mouse. Scale bar 2000 μm. **v**, Haematoxylin and eosin (HE) staining of epiAT, 20x magnification. Scale bar, 100 μm. Representative pictures. **w**, ingAT adipocyte area across conditions. (*n* = 4 mice each, 5-8 pictures each). **x**, HE staining of scAT, 20x magnification. Scale bar, 200 μm. **y**, epiAT adipocyte area across conditions. (*n* = 4 mice each, 5-8 pictures each). **z**, Quantification of collagen content from Maison's Trichome staining. (*n* = 4 mice each, 20 pictures each). Significance was calculated between age matched controls and experimental groups. Significance a, b, d, e, i, j, z was calculated using two-tailed Mann-Whitney tests. Significance for c, f-h, m was calculated using unpaired, two-tailed t-tests with Welch's correction. Error bars represent s.d. Significance for p and r was calculated using unpaired, multiple t-tests with Benjamini, Krieger and Yekutieli post hoc test for multiple comparisons. ns = *FDR* > 0.01, **FDR* < 0.01, ***FDR* < 0.001, ****FDR* < 0.0001.

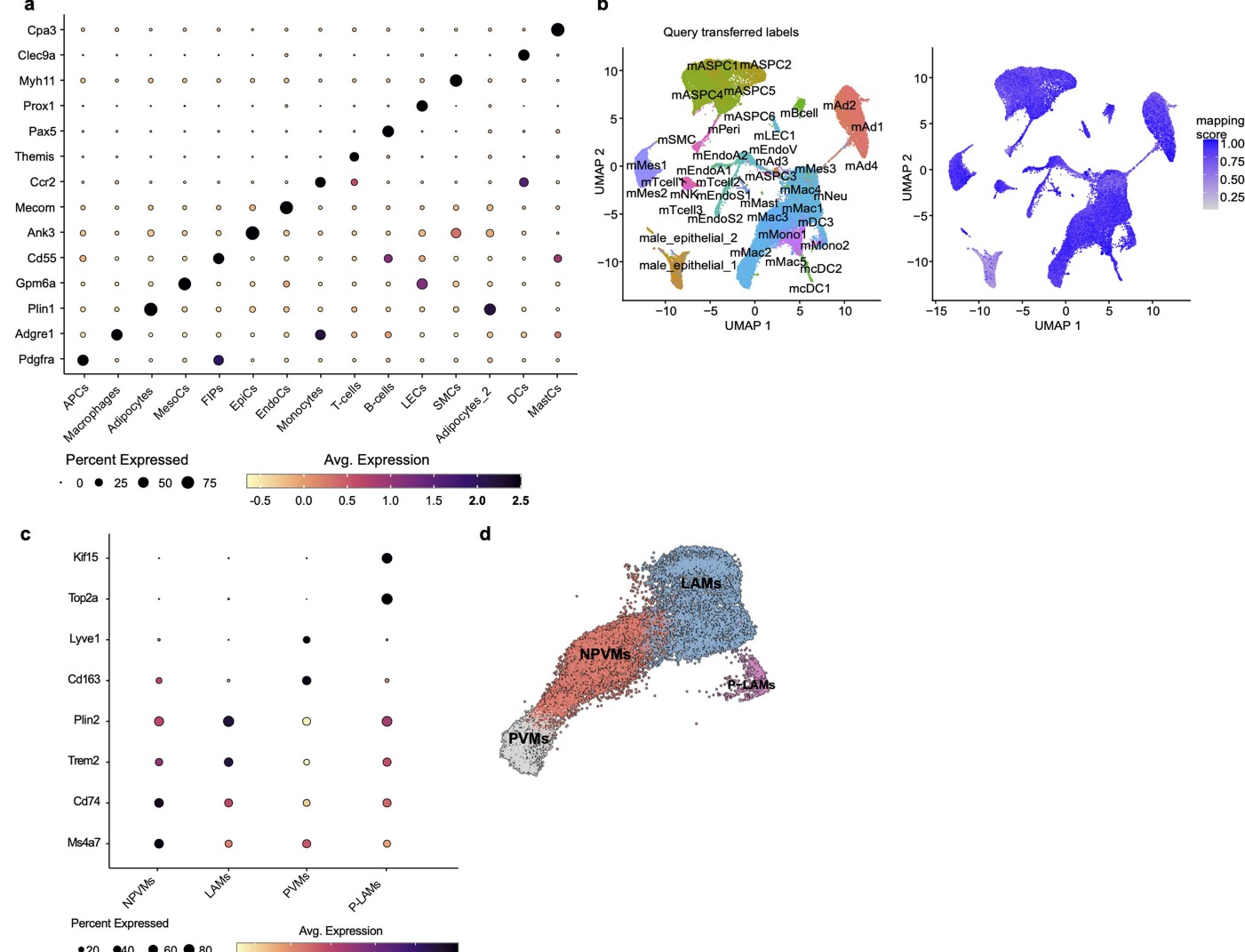

**Extended Data Fig. 6 | Annotation of mouse epiAT. a**, Cluster markers used to annotate cell clusters of mouse epiAT. **b**, UMAP visualization representing epiAT samples coloured by predicted cell subtypes from the Emont et al. mouse epididymal AT dataset. Feature plots showing reference mapping scores illustrating how well this dataset maps to the Emont et al. dataset. **c**, Macrophage subcluster markers. **d**, UMAP of 16,567 nuclei representing macrophage subclusters. APCs, adipocyte progenitor cells; DCs, dendritic cells; EpiCs, epithelial cells; EndoCs, endothelial cells; FIPs, fibro-inflammatory progenitors; LECs, lymphatic endothelial cells; MastCs, mast cells; MesoCs, mesothelial cells; SMCs, (vascular) smooth muscle cells; NPVMs, non-perivascular macrophages; LAMs, lipid-associated macrophages; PVMs, perivascular macrophages; P-LAMs, proliferating LAMs.

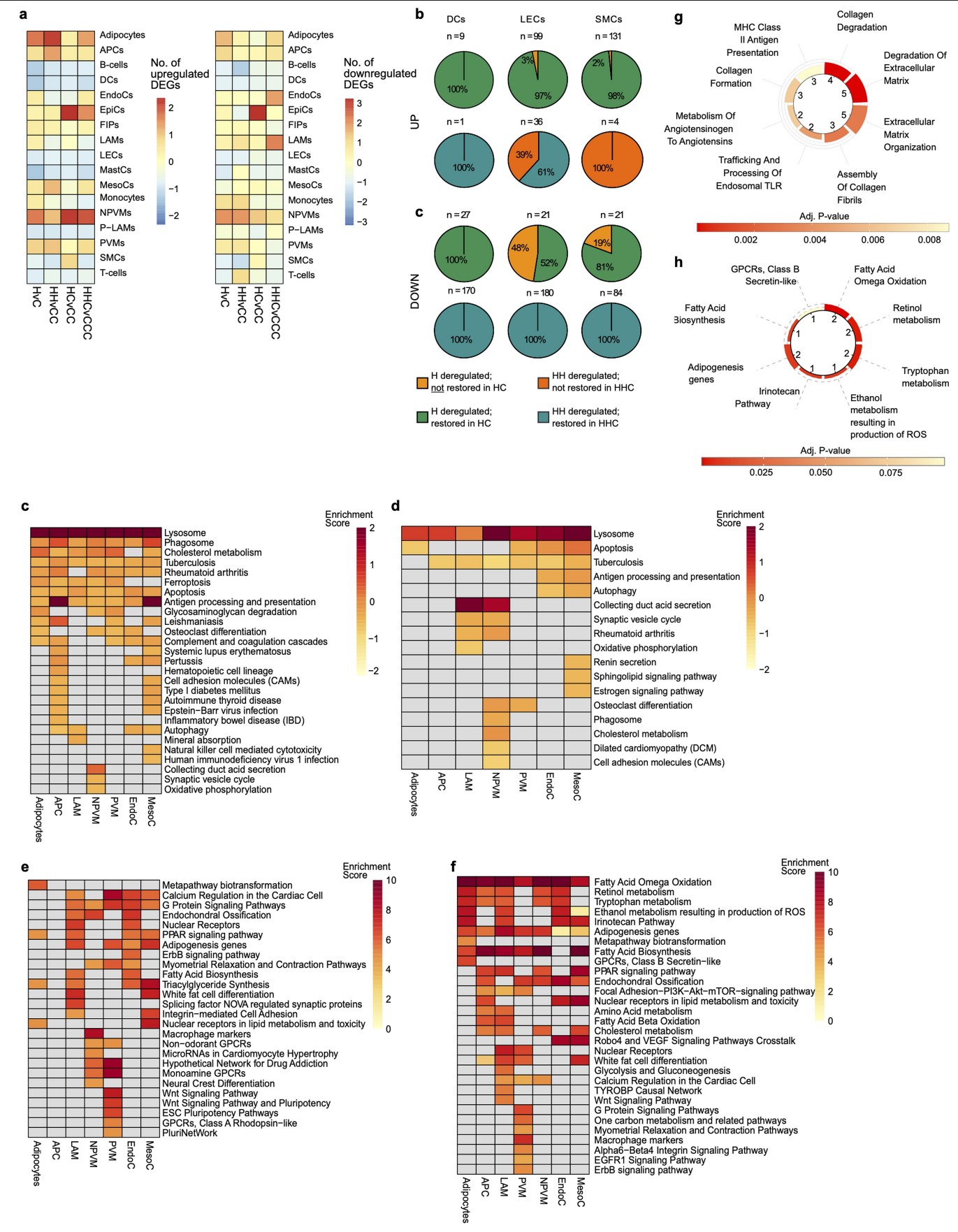

**Extended Data Fig. 7 |** See next page for caption.

**Extended Data Fig. 7 | Transcriptional changes persist weight loss in epiAT.**
**a**, Number of upregulated (left) and downregulated (right) DEGs per cell type per comparison (H vs C, HC vs CC, HH vs CC, HHC vs CCC) scaled by column. **b**, Proportion of retained transcriptional changes in different cell types. (Wilcoxon Rank Sum test, adjusted p-value < 0.05 by the Bonferroni correction method; FC > ±0.5). **c,d**, Top (significant) persistently upregulated (memory) (c) and downregulated (d). pathway terms in HC adipocytes based on Wikipathways database. **e,f**, Significant Wikipathways term enrichment scores related to persistently upregulated genes in HHC (f) and HC (g) per cell type.

**g,h**, Significant Wikipathways term enrichment scores related to persistently downregulated genes in HHC (f) and HC (g) per cell type. Significance for c-h was calculated using Fisher's exact test, with adjusted P-value < 0.05 by the Benjamini-Hochberg method for correction. APCs, adipocyte progenitor cells; DCs, dendritic cells; EpiCs, epithelial cells; EndoCs, endothelial cells; FIPs, fibro-inflammatory progenitors; LECs, lymphatic endothelial cells; MastCs, mast cells; MesoCs, mesothelial cells; SMCs, (vascular) smooth muscle cells; NPVMs, non-perivascular macrophages; LAMs, lipid-associated macrophages; PVMs, perivascular macrophages; P-LAMs, proliferating LAMs.

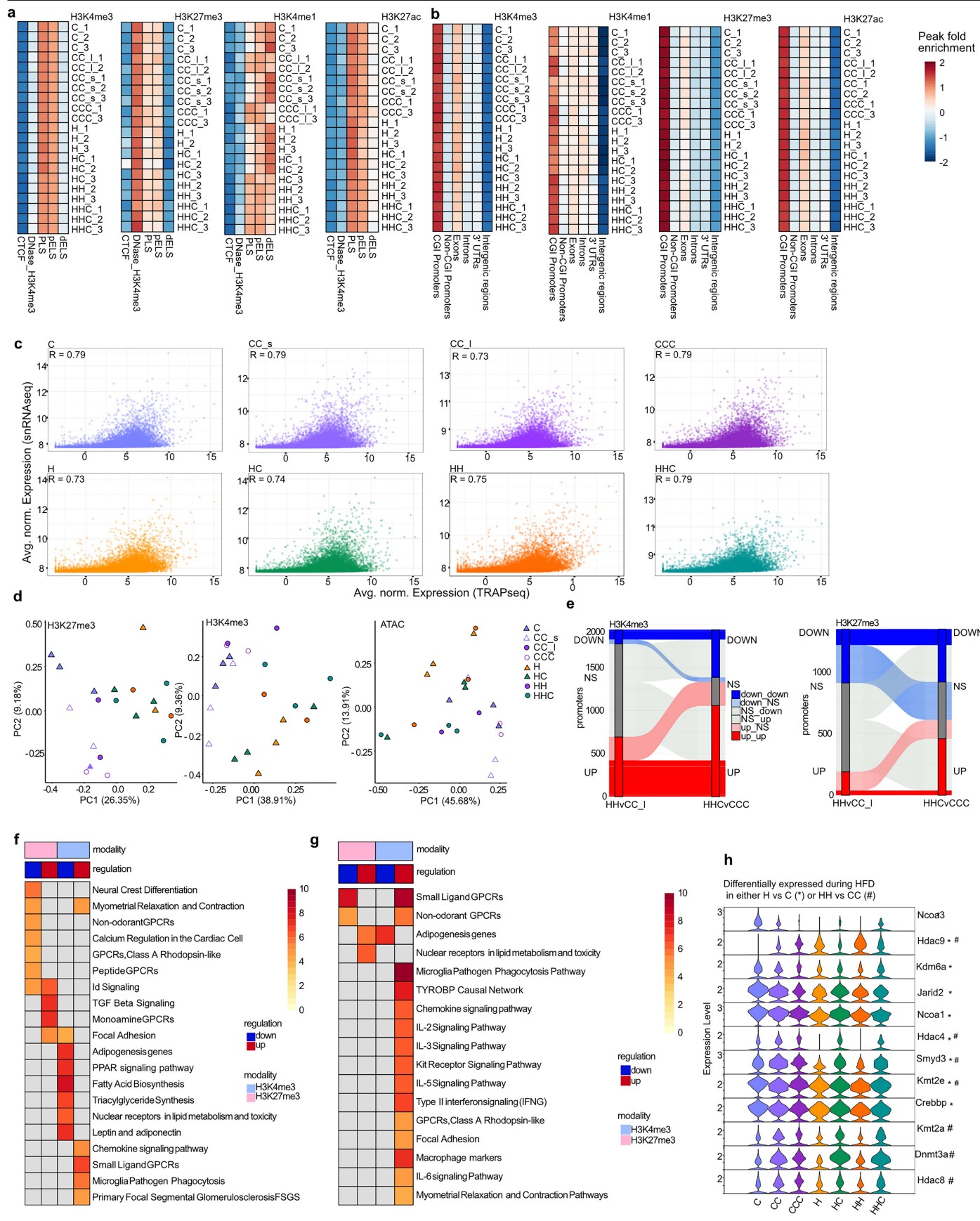

**Extended Data Fig. 8** | See next page for caption.

**Extended Data Fig. 8 | Epigenetic memory persists after weight loss.**
**a**, Peak fold enrichment of called peaks from each CUT&Tag library for genomic features, scaled from −2 to 2. **b**, Peak fold enrichment of called peaks from each CUT&Tag library for ENCODE cCREs, scaled from −2 to 2.
**c**, Scatterplots of pairwise correlation of average normalized expression of pseudo bulk adipocytes from snRNA-seq (log2 cpm) and average normalized expression of translating RNA (TRAPseq) from labelled adipocytes (log2 cpm) per condition. Spearman's correlation coefficient R is indicated. **d**, PCA plots of H3K4me3, H3K27me3 and ATAC-seq across all conditions quantified over peaks overlapping promoters with reads summed up at gene level; each dot represents one biological replicate. **e**, Dynamics of differentially H3K4me3-marked (left) and H3K27me3-marked promoters (y-axis) from HH to HHC.
**f**, Significant Wikipathways term enrichment scores related to genes associated with persistently differentially marked promoters by H3K27me3 (from Fig. 4f) or H3K4me3 (from Fig. 4g) in HC adipocytes. **g**, Significant Wikipathways term enrichment scores related to genes associated with persistently differentially marked promoters by H3K27me3 (from e) or H3K4me3 (from e) in HHC adipocytes. **h**, Expression of genes encoding for epigenetic modifiers significantly deregulated either in H (*) or HH (#) adipocytes. (Wilcoxon Rank Sum test, adjusted p-value < 0.05 by the Bonferroni correction method; fold change (FC) > ±0.5). None of the epigenetic modifiers are deregulated in HC or HHC adipocytes. Significance for f-g was calculated using Fisher's exact test, with adjusted P-value < 0.05 by the Benjamini-Hochberg method for correction. cCREs, candidate cis-regulatory elements as defined by ENCODE. CTCF, not TSS-overlapping and with high DNase and CTCF signals only; DNase–H3K4me3, not TSS-overlapping and with high DNase and H3K4me3 signals only; dELS, TSS-distal with enhancer-like signatures; PLS, TSS-overlapping with promoter-like signatures; pELS, TSS-proximal with enhancer-like signatures.

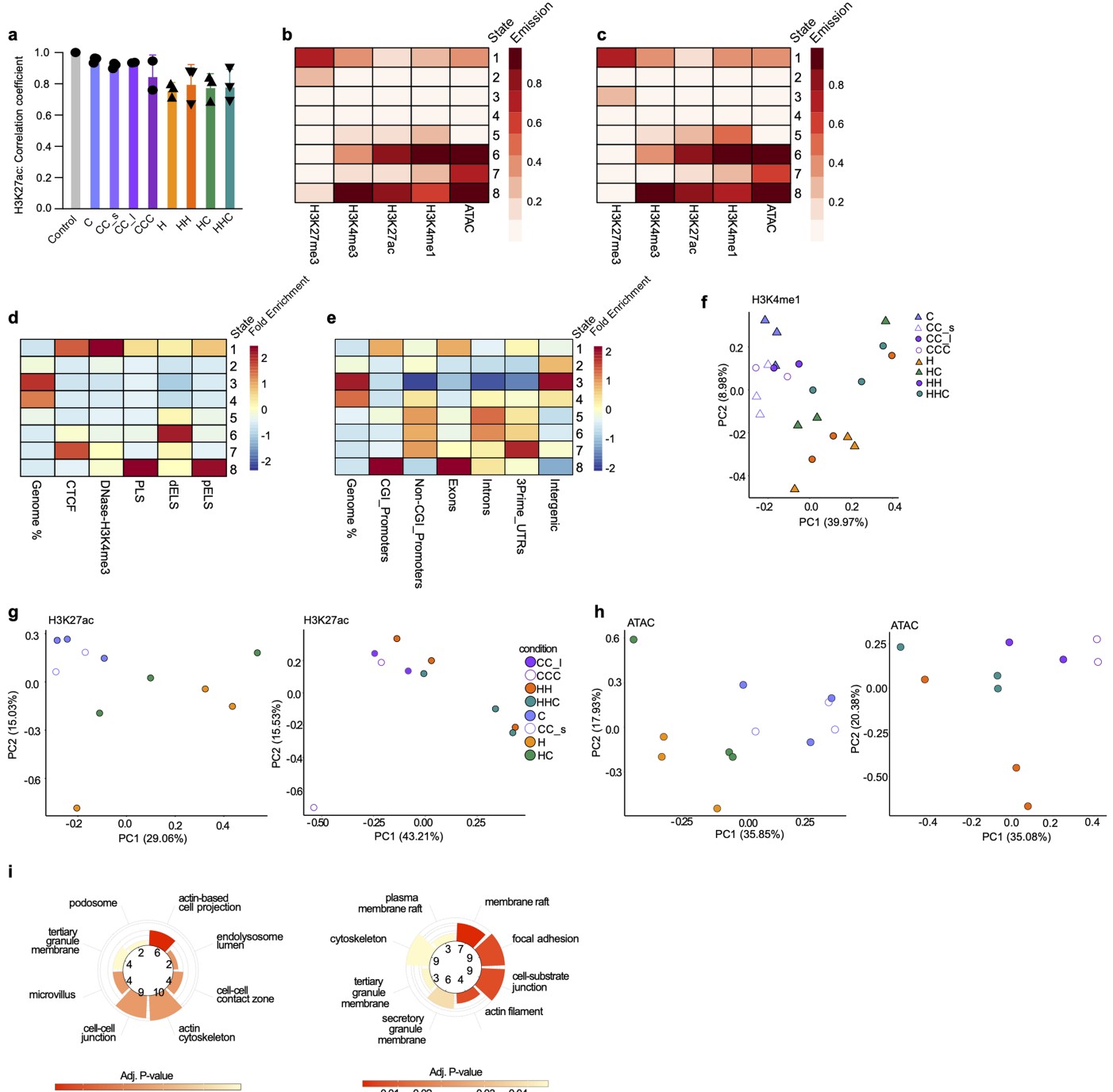

**Extended Data Fig. 9 | Adipocyte specific enhancers retain an epigenetic memory. a**, Correlation coefficient R (Pearson) of quantified peaks of H3K27ac against a hypothetical healthy control (*n* = 2-3 each) with s.d. Each dot represents an individual biological replicate. **b,c**, ChromHMM analysis of the adipocyte hPTM profiles for conditions C, CC_s, H and HC (b) and CC_l, CCC, HH and HHC (c). The colour scale corresponds to the emission parameter of each hPTM for each state. **d,e**, Fold enrichment of ChromHMM states from b and c for total genomic fraction coverage, ENCODE cCREs, and genomic features scaled from −2 to 2. State 5, 6 and 7 are identified as enhancers. **f**, PCA plot of quantified adipocyte specific enhancers from all conditions as marked by H3K4me1. Each dot represents an individual biological replicate. **g**, PCA plot of quantified adipocyte specific enhancers as marked by H3K27ac. Each dot

represents an individual biological replicate. **h**, PCA plot of quantified adipocyte specific enhancers as marked by H3K27ac. Each dot represents an individual biological replicate. **i**, Top (significant) GO Cellular Component terms for genes linked to newly emerged and acetylated enhancers for H and HC (left) and HH and HHC (right). (Fisher's exact test, adjusted p-value < 0.05 by the Benjamini-Hochberg method for correction). cCREs, candidate cis-regulatory elements as defined by ENCODE. CTCF, not TSS-overlapping and with high DNase and CTCF signals only; DNase−H3K4me3, not TSS-overlapping and with high DNase and H3K4me3 signals only; dELS, TSS-distal with enhancer-like signatures; PLS, TSS-overlapping with promoter-like signatures; pELS, TSS-proximal with enhancer-like signatures.

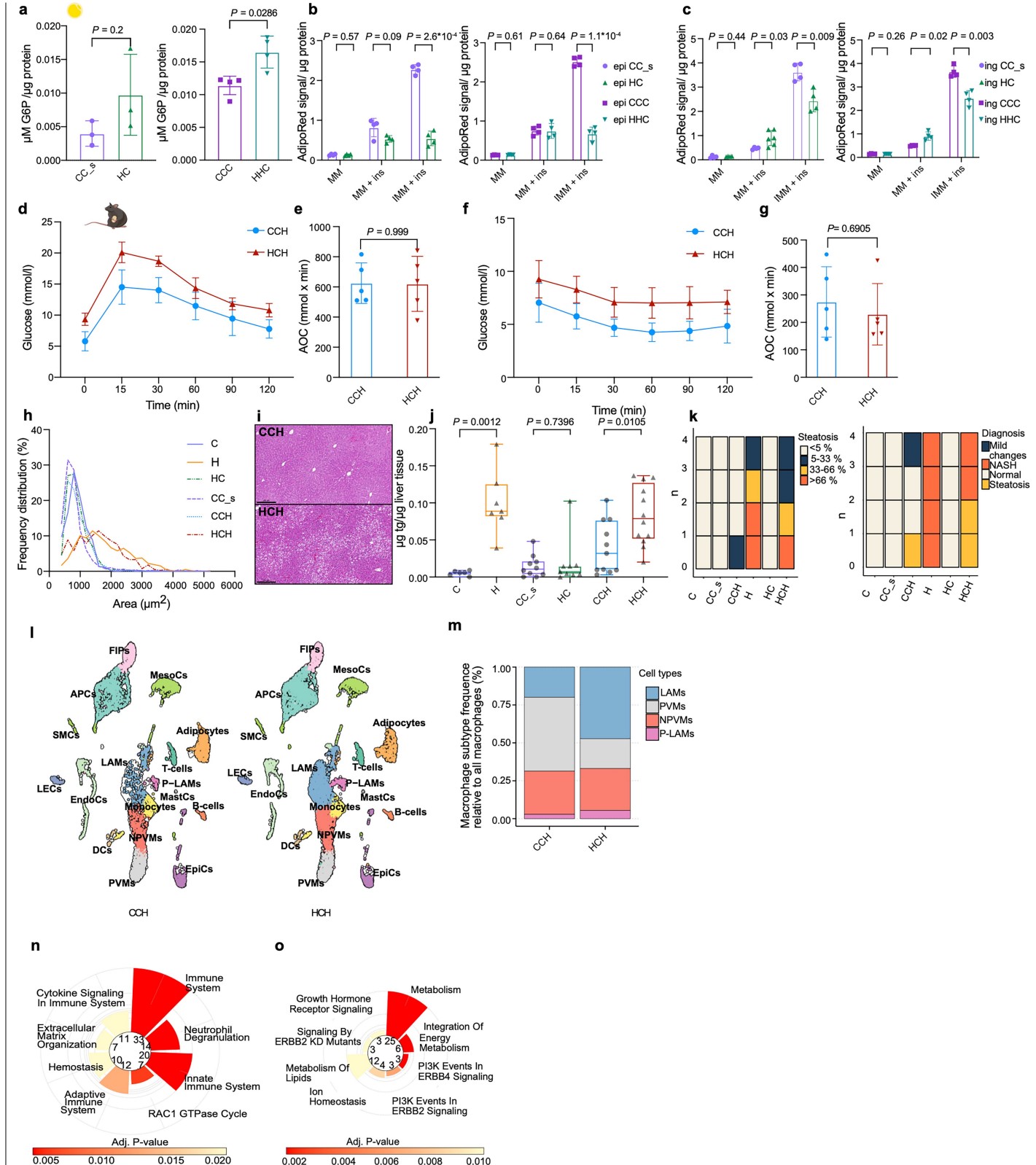

**Extended Data Fig. 10** | See next page for caption.

**Extended Data Fig. 10 | Other responses of primed mice and cells to obesogenic stimuli. a**, Glucose uptake of isolated, cultured primary adipocytes from ingAT from CC_s and HC (left) and CCC and HHC (right) mice. Each dot represents an individual biological replicate of a pool of 3 mice. **b,c**, AdipoRed signal of dividing SVF (MM), SVF stimulated with 10 nm insulin only (MM + Ins) and induced SVF with 10 nm insulin (IMM + Ins) 10 days after induction/no induction of differentiation from epiAT (b) and ingAT (c) SVF from CC_s, HC, CCC and HHC mice. Each dot represents an individual biological replicate of a pool of 3 mice. Every SVF pool was tested in all three conditions. **d**, GTT of HCH and CCH mice; blood glucose levels ($n = 5$ each). **e**, AOC of GTTs from d. **f**, ITT of CCH and HCH mice; blood glucose levels ($n = 5$ each). **g**, AOC from ITTs from e. **h**, Distribution of epiAT adipocyte area. ($n = 4$ mice each, 10 pictures each). **i**, Representative images of liver HE stained sections from CCH and HCH, 20x magnification, scale bar 200 µm. **j**, Liver tg per µg liver tissue (C&H:$n = 6$ each, CC_s: $n = 10$, HC: $n = 9$, CCH: $n = 11$, HCH: $n = 12$, from 2-3 experiments). Boxplot represents minimum, maximum and median. **k**, Pathological scoring of liver sections per group ($n = 4$ each). **l**, UMAP of 15,665 nuclei representing epiAT pools ($n = 5$ pooled mice each) from CCH and HCH split by condition. **m**, Relative abundance of macrophage subclusters. **n,o**, Top significant pathway terms from upregulated (n) and downregulated (o) HCH DEGs that are explained by the epigenetic state in HC adipocytes based on Reactome database. (Fisher's exact test, adjusted p-value < 0.05 by the Benjamini-Hochberg method for correction). Significance was calculated between age matched controls and experimental groups. Significance for a, e, g, was calculated using two-tailed Mann-Whitney tests. Significance for b, c was calculated using unpaired, two-tailed Student's $t$-tests with Welch's correction and Benjamini, Krieger, and Yekutieli correction for multiple testing. Significance for j was calculated using unpaired two-tailed Student's $t$-tests with Welch's correction. Error bars represent s.d. APCs, adipocyte progenitor cells; DCs, dendritic cells; EpiCs, epithelial cells; EndoCs, endothelial cells; FIPs, fibro-inflammatory progenitors; LECs, lymphatic endothelial cells; MastCs, mast cells; MesoCs, mesothelial cells; SMCs, (vascular) smooth muscle cells; NPVMs, non-perivascular macrophages; LAMs, lipid-associated macrophages; PVMs, perivascular macrophages; P-LAMs, proliferating LAMs.

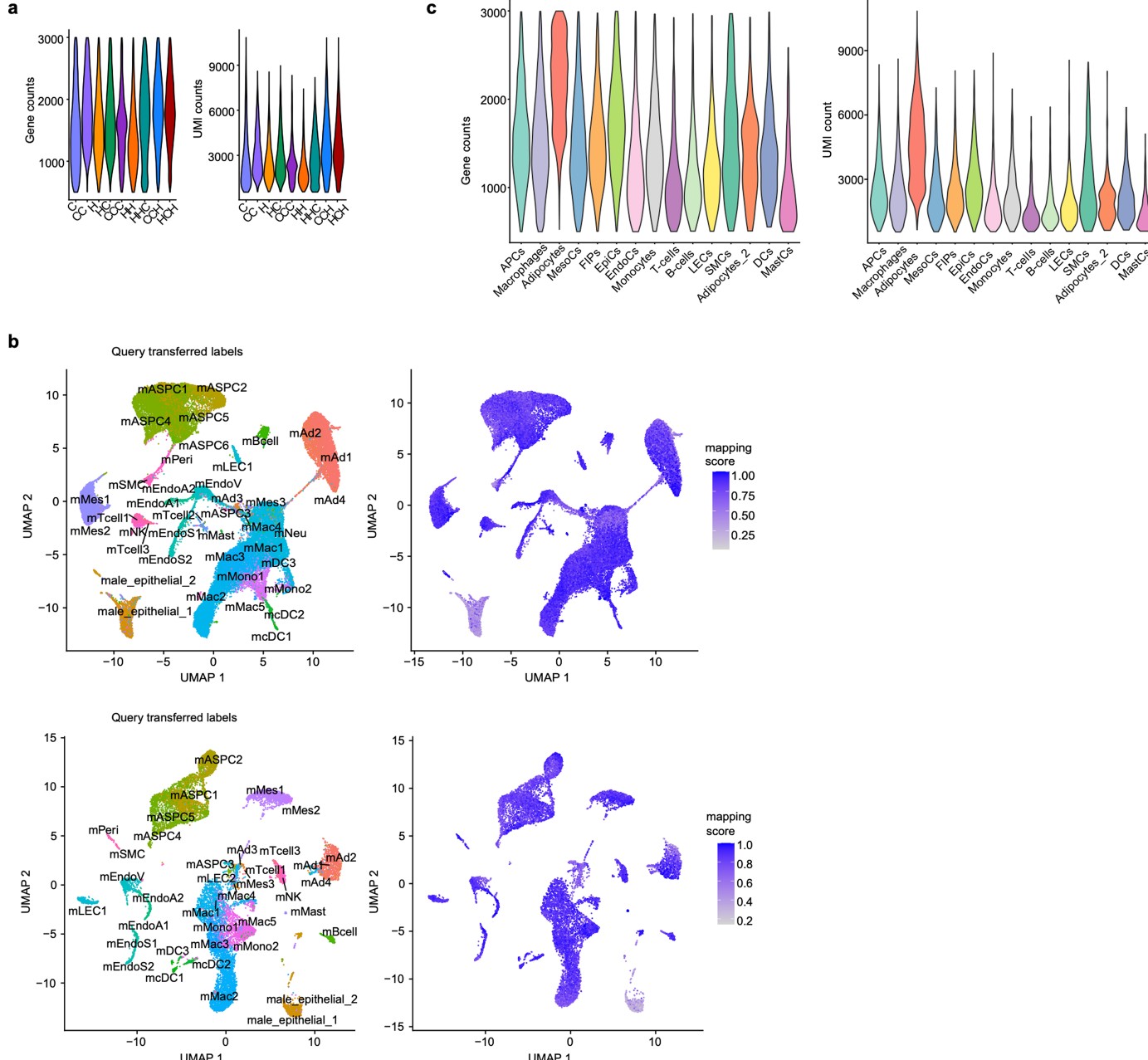

**Extended Data Fig. 11 | Quality metrics of mouse snRNAseq data. a**, Gene counts and the number of unique molecular identifiers (UMIs) per condition of mouse epiAT samples. **b**, UMAP visualization representing integrated epiAT samples from the weight loss study (C, CC, CCC, H, HH, HC, HHC) and from the "yoyo" study (CCH, HCH) coloured by predicted cell subtypes from the Emont et al. mouse epididymal AT dataset. Feature plots showing reference mapping scores illustrating how well these datasets maps to the Emont et al. dataset. **c**, gene counts and the number of UMIs per cell type from mouse epiAT samples.

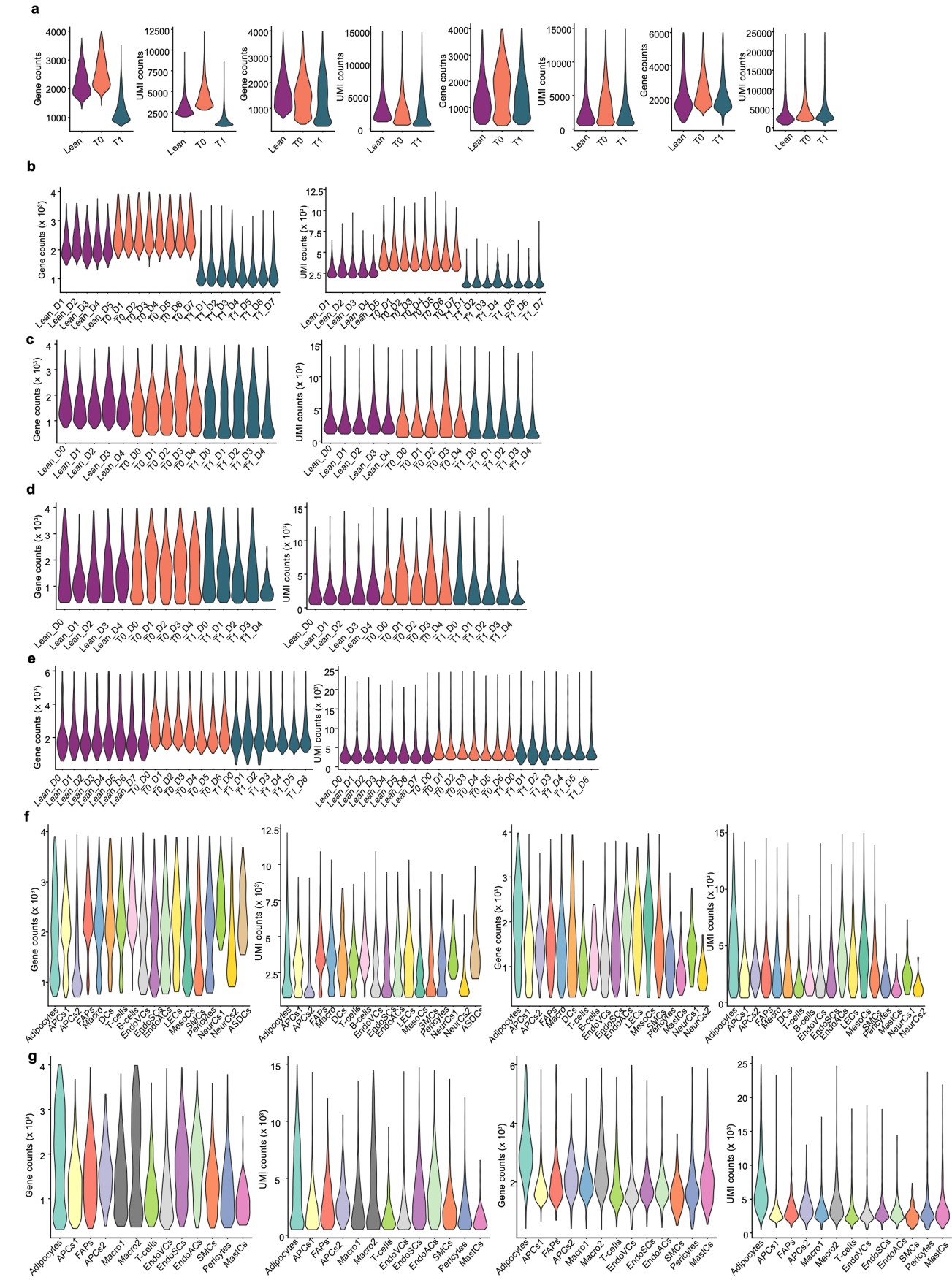

**Extended Data Fig. 12** | See next page for caption.

**Extended Data Fig. 12 | Quality metrics of human snRNAseq data. a**, Gene counts and the number of UMIs per condition in the omAT samples from the MTSS (left), LTSS (second left) and in scAT samples from the LTSS (second from right) and NEFA (right) study. **b,c**, Gene counts and the number of UMIs per donor in the omAT samples from the MTSS (b) and LTSS (c) study. **d,e**, Gene counts and the number of UMIs per donor in scAT samples from the LTSS (d) and NEFA (e) study. **f**, Gene counts and the number of UMIs per assigned cell type in the omAT samples from the MTSS (left) and LTSS (right) study. **g**, Gene counts and the number of UMIs per assigned cell type in the scAT samples from the LTSS (left) and NEFA (right) study.

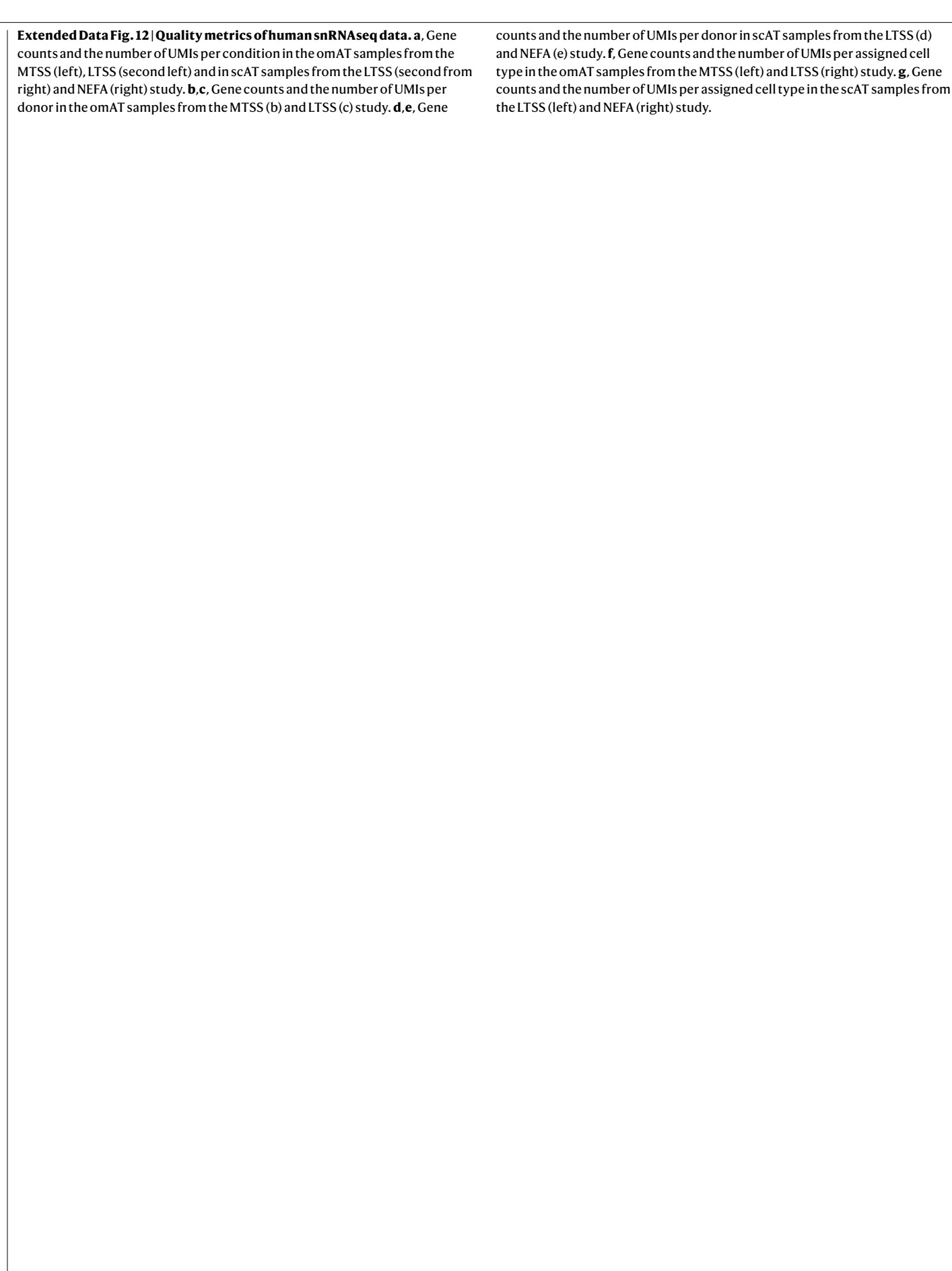

**Extended Data Table 1 | Clinical parameters**

| Parameter | Lean MTSS | Responder MTSS | Lean LTSS | Responder LTSS | Lean NEFA | Responder NEFA |
|---|---|---|---|---|---|---|
| BMI Loss (%) | NA | 31.26±4.77 | NA | 26.21±0.92 | NA | 34.22±7.61 |
| Sex (m/f) | 1/4 | 2/6 | 2/3 | 2/3 | 0/8 | 0/7 |
| Lipid lowering medication at T0 (yes/no) | 0/5 | 0/8 | 0/5 | 0/5 | 0/8 | 1/7 |
| Anti-diabetic medication at T0 (yes/no) | 0/5 | 0/8 | 0/5 | 0/5 | 0/8 | 0/7 |
| Age T0 (years) | 37.05±8.29 | 39.5±5.53 | 45.34±3.01 | 48.92±4.2 | 49.88±13.9 | 38.62±12.64 |
| BMI T0 (kg/m$^2$) | 24.97±2.86 | 48.19±7.01 | 25.04±1.22 | 46.96±6.38 | 26.11±2.9 | 40.35±2.15 |
| Fasting plasma glucose T0 (mmol/l) | NA | NA | 4.8±0.35 | 5.82±0.44 | 5.15±0.46 | 5.19±0.24 |
| HbA1cT0 (%) | 5.14±0.07 | 5.75±0.23 | 5.4±0.77 | 5.87±0.19 | 5.36±0.32 | 5.23±0.27 |
| HOMA IR T0 | NA | NA | 1.95±0.86 | 5.14±0.9 | 1.3±0.53 | 3.18±1.66 |
| Fasting plasma insulin T0 (pmol/l) | NA | NA | 64.14±29.32 | 137.68±24.8 | 35.17±14.37 | 86.59±46.19 |
| Cholesterol T0 (mmol/l) | 4.9±1.18 | 4.86±1.07 | 4.3±0.75 | 4.71±1.59 | 4.44±0.9 | 4.45±0.91 |
| TG T0 (mmol/l) | 1.23±0.37 | 1.98±0.55 | 0.82±0.14 | 2.12±0.92 | 0.76±0.26 | 1.18±0.43 |
| HDL-cholesterol T0 (mmol/l) | 2.24±0.72 | 1.09±0.19 | 1.82±0.34 | 1.13±0.08 | 1.47±0.4 | 1.14±0.13 |
| LDL-cholesterol T0 (mmol/l) | 2.04±1.54 | 3.39±1.04 | 2.31±0.8 | 2.97±1.17 | NA | NA |
| Age T1 (years) | NA | 41.5±5.53 | NA | 50.92±4.14 | NA | 40.62±12.64 |
| BMI T1 (kg/m$^2$) | NA | 33.26±6.25 | NA | 34.66±4.86 | NA | 26.51±2.04 |
| Fasting plasma glucose T1 (mmol/l) | NA | NA | NA | 4.93±0.43 | NA | 5.05±0.14 |
| HbA1c T1 (%) | NA | 5.41±0.54 | NA | 5.12±0.45 | NA | 5.16±0.2 |
| HOMA IR T1 | NA | NA | NA | 1.67±0.25 | NA | 0.87±0.18 |
| Fasting plasma insulin T1 (pmol/l) | NA | NA | NA | 52.84±5.78 | NA | 23.28±4.61 |
| Cholesterol T1 (mmol/l) | NA | 6.37±2.21 | NA | 4.07±0.76 | NA | 3.72±0.7 |
| TG T1 (mmol/l) | NA | 3.79±4.05 | NA | 0.93±0.17 | NA | 0.64±0.14 |
| HDL-cholesterol T1 (mmol/l) | NA | 1.16±0.08 | NA | 1.65±0.57 | NA | 1.54±0.27 |
| LDL-cholesterol T1 (mmol/l) | NA | 4.05±1.64 | NA | 2.42±1.01 | NA | NA |

Clinical parameters from the MTSS, LTSS and NEFA studies from T0, T1 and lean donors provided as mean ± s.d.

BMI, body mass index; HbA1c, glycated heamoglobin; HDL, high density lipoprotein; LDL, low density lipoprotein; HOMA-IR, homeostatic model assessment index of insulin resistance; TG, triglycerides.

# Reporting Summary

## Statistics

For all statistical analyses, confirm that the following items are present in the figure legend, table legend, main text, or Methods section.

| n/a | Confirmed | |
|---|---|---|
| ☐ | ☒ | The exact sample size (*n*) for each experimental group/condition, given as a discrete number and unit of measurement |
| ☐ | ☒ | A statement on whether measurements were taken from distinct samples or whether the same sample was measured repeatedly |
| ☐ | ☒ | The statistical test(s) used AND whether they are one- or two-sided<br>*Only common tests should be described solely by name; describe more complex techniques in the Methods section.* |
| ☒ | ☐ | A description of all covariates tested |
| ☐ | ☒ | A description of any assumptions or corrections, such as tests of normality and adjustment for multiple comparisons |
| ☐ | ☒ | A full description of the statistical parameters including central tendency (e.g. means) or other basic estimates (e.g. regression coefficient) AND variation (e.g. standard deviation) or associated estimates of uncertainty (e.g. confidence intervals) |
| ☐ | ☒ | For null hypothesis testing, the test statistic (e.g. *F*, *t*, *r*) with confidence intervals, effect sizes, degrees of freedom and *P* value noted<br>*Give P values as exact values whenever suitable.* |
| ☒ | ☐ | For Bayesian analysis, information on the choice of priors and Markov chain Monte Carlo settings |
| ☒ | ☐ | For hierarchical and complex designs, identification of the appropriate level for tests and full reporting of outcomes |
| ☒ | ☐ | Estimates of effect sizes (e.g. Cohen's *d*, Pearson's *r*), indicating how they were calculated |

*Our web collection on statistics for biologists contains articles on many of the points above.*

## Software and code

Policy information about availability of computer code

| Data collection | Sable Systems Promethion ExpeData 1.9.27b Software; BioTek Gen5 Software; 3DHISTECH Slide Viewer 2; Adiposoft as plugin in Fiji Image J; a NovaSeq 6000 and a NovaSeqX were used for sequencing |
|---|---|
| Data analysis | Analysis of non-sequencing data: GraphPad Prism v9.5.1<br>Analysis of sequencing data:<br><br>Single nucleus RNA seq for mouse:<br>The 10x Genomics Cell Ranger v.6.1.2 pipeline was used for demultiplexing, read alignment to reference genome The R packages Seurat v4.1.0; scDblFinder and Scpubr were used for analysis.<br><br>Single nucleus RNA seq (human):<br>The 10x Genomics Cell Ranger v.7.2.0 pipeline was used for demultiplexing, read alignment to reference genome The R packages Seurat v4.1.0; scDblFinder and Scpubr were used for analysis. SNP-calling and demultiplexing was performed using cellsnp-lite and vireo.<br><br>mouse TRAP-seq:<br>Quality control of the raw reads was performed using FastQC v0.11.9. Raw reads were trimmed using TrimGalore v0.6.6 (https://github.com/FelixKrueger/TrimGalore). Filtered reads were aligned against the reference mouse genome assembly mm10 using HISAT2 v2.2.1. Raw gene counts were quantified using the featureCounts program of subread v2.0.1. Differential expression analysis was performed using the R package EdgeR.<br><br>CUT&Tag and ATAC-seq: |

Quality control of the raw sequencing reads was performed using FastQC v0.11.9. Raw reads were trimmed off low-quality bases and adapter sequences using TrimGalore v0.6.6 (https://github.com/FelixKrueger/TrimGalore). Aligned bam files were sorted based on chromosomal coordinates using the sort function of samtools v1.13. Sorted bam files were summarized into bedgraph files using the genomecov function of bedtools v2.30. Peaks were called from CUT&Tag-seq and ATAC-seq libraries on individual bedgraph files using SEACR v1.3. Called peaks were combined to generate a union peak list and quantified using the R package chromVAR v1.16 generating a raw peak count matrix. MOFA2 v1.4.0 was used to identify variation and ChromHMM v1.22 for identifiying chromatin states.
Peaks were annotated with ChIPSeeker and differential analysis was performed using EdgeR.

Gene set enrichment:
The package enrichR was used.

For manuscripts utilizing custom algorithms or software that are central to the research but not yet described in published literature, software must be made available to editors and reviewers. We strongly encourage code deposition in a community repository (e.g. GitHub). See the Nature Portfolio guidelines for submitting code & software for further information.

# Data

Policy information about availability of data

All manuscripts must include a data availability statement. This statement should provide the following information, where applicable:
- Accession codes, unique identifiers, or web links for publicly available datasets
- A description of any restrictions on data availability
- For clinical datasets or third party data, please ensure that the statement adheres to our policy

All mouse sequencing data that support the findings of this study have been deposited on GEO, with the accession code GSE236580 (token for reviewers: qtcrauuovrghlcr). Human snRNAseq data from MTSS and LTSS studies are available upon request from CW and MB. Human snRNAseq data from the NEFA study is available upon request from FvM, MR and NM.

# Research involving human participants, their data, or biological material

Policy information about studies with human participants or human data. See also policy information about sex, gender (identity/presentation), and sexual orientation and race, ethnicity and racism.

| | |
|---|---|
| Reporting on sex and gender | The human studies MTSS (3 male, 10 female) and LTSS (4 male, 6 female) are part of the Leipzig Obesity Biobank (LOBB) maintained by Dr. Matthias Blüher. The human study NEFA (15 female) was run by Mikalel Rydén and Niklas Mejhert at the Karolinska Institute. |
| Reporting on race, ethnicity, or other socially relevant groupings | N/A |
| Population characteristics | MTSS: Only individuals that lost 25% of BMI (31.26 ± 4.77 % reduction) and lean, healthy (BMI < 27) controls were selected from the MTSS study for this study. LTSS study: Only individuals that lost 25% of BMI (26.21 ± 0.92 % reduction) and lean, healthy (BMI < 27) controls were selected from the LTSS study for this study. NEFA study: Only individuals that lost 25% of BMI (34.22 ± 7.61 % reduction) and lean, healthy (BMI < 27) controls were selected from the NEFA study for this study. |
| Recruitment | MTSS and LTSS studies are part of the Leipzig Obesity Biobank (LOBB). The NEFA study is registered with clinical trial number: NCT01727245 |
| Ethics oversight | The human studies MTSS and LTSS was conducted in accordance with the Declaration of Helsinki and approved by the Ethics Committee of the University of Leipzig (approval number: 159-12-21052012). Written informed consent was obtained from all subjects involved in the study prior to surgery. The human study NEFA was conducted in accordance with the with the Declaration of Helsinki and approved by the Ethics Committee of the Karolinska Institute, Stockholm (approval number: 2011/1002-31/1). |

Note that full information on the approval of the study protocol must also be provided in the manuscript.

# Field-specific reporting

Please select the one below that is the best fit for your research. If you are not sure, read the appropriate sections before making your selection.

☒ Life sciences          ☐ Behavioural & social sciences          ☐ Ecological, evolutionary & environmental sciences

For a reference copy of the document with all sections, see nature.com/documents/nr-reporting-summary-flat.pdf

# Life sciences study design

All studies must disclose on these points even when the disclosure is negative.

| | |
|---|---|
| Sample size | No sample size calculations were performed. Samples sizes were chosen to allow sufficient statistical analysis to be performed. For the human cohort sample size was dependent on the availability of samples of subjects meeting the selection criteria (e.g. weight loss, healthy). |
| Data exclusions | snRNAseq data from two donors were exlcuded after consultation with surgeons because they either contained more thant 50% B-cells or no adipocytes. This is explained in the Methods section. |
| Replication | Experimental findings were verified by biological replicates. Each experiment was performed multiple times. Replication attempts were successful and well correlated. At least two independent biological replicates were included in the RNA-seq, ATAC-seq and CUT&Tag (like ChIPseq) experiments. For snRNAseq 5 of mice individual biological replicates were pooled per condition. For human snRNAseq samples belonging to one group and time point were pooled and later SNP demultiplexed. |
| Randomization | Samples were not randomised and were allocated into experimental groups by condition. |
| Blinding | The investigators were not blinded to the mice as they themselves were treating and sacrificing the mice. However, investigators were blinded during mouse sample processing. Investigators were not blinded to the sample identity for sequencing data (human and mouse) as sequencing data was produced by objective quantitative methods. For histological image quantification the investigators analysing adipocytes and imaging sections were blinded. |

# Reporting for specific materials, systems and methods

We require information from authors about some types of materials, experimental systems and methods used in many studies. Here, indicate whether each material, system or method listed is relevant to your study. If you are not sure if a list item applies to your research, read the appropriate section before selecting a response.

## Materials & experimental systems

| n/a | Involved in the study |
|---|---|
| ☐ | ☒ Antibodies |
| ☒ | ☐ Eukaryotic cell lines |
| ☒ | ☐ Palaeontology and archaeology |
| ☐ | ☒ Animals and other organisms |
| ☒ | ☐ Clinical data |
| ☒ | ☐ Dual use research of concern |
| ☒ | ☐ Plants |

## Methods

| n/a | Involved in the study |
|---|---|
| ☐ | ☒ ChIP-seq |
| ☒ | ☐ Flow cytometry |
| ☒ | ☐ MRI-based neuroimaging |

## Antibodies

| | |
|---|---|
| Antibodies used | anti-H3K4me3 (abcam, #ab8580), anti-H3K27me3 (Cell Signaling Technology, #C36B11), anti-H3K27ac (abcam, #ab4729), anti-H3K4me1 (abcam, #ab8895) |
| Validation | anti-H3K4me3 (abcam, #ab8580): has been validated by abcam for human and cow and is predicted for mouse for ChIP; has been validated in various publications for mouse and other species for example in: https://doi.org/10.1016/j.molcel.2022.03.009; <br> anti-H3K27me3 (Cell Signaling Technology, #C36B11): has been validate for human, mouse, rat and monkey by Cell Signaling Technology for ChIP; <br> anti-H3K27ac (abcam, #ab4729): has been validate for human, mouse, rat, and cow by abcam for ChIP <br> anti-H3K4me1 (abcam, #ab8895): has been validate for human, mouse, rat, and cow by abcam for ChIP |

## Animals and other research organisms

Policy information about studies involving animals; ARRIVE guidelines recommended for reporting animal research, and Sex and Gender in Research

| | |
|---|---|
| Laboratory animals | C57NL/6-Tg(Adipoq CreER)426Biat/N x B6;129S6-Gt(ROSA)26Sortm2(CAG-NuTRAP)Evdr/J were used. C57BL/6J DIO (#380050) and DIO control (#380056) male mice were obtained from Jackson Laboratory (USA). <br> All mice were kept on a 12-h/12-h light/dark cycle and 20-60% (23℃) humidity in individually ventilated cages in groups of between two and five mice in a pathogen-free animal facility of SLA ETH Zurich. |
| Wild animals | No wild animals were used in this study. |
| Reporting on sex | Only male mice were used in this study. |

| Field-collected samples | No field collected samples were used in this study. |
|---|---|
| Ethics oversight | All animal experiments were approved by the cantonal veterinary office Zurich. |

Note that full information on the approval of the study protocol must also be provided in the manuscript.

## Plants

| Seed stocks | *Report on the source of all seed stocks or other plant material used. If applicable, state the seed stock centre and catalogue number. If plant specimens were collected from the field, describe the collection location, date and sampling procedures.* |
|---|---|
| Novel plant genotypes | *Describe the methods by which all novel plant genotypes were produced. This includes those generated by transgenic approaches, gene editing, chemical/radiation-based mutagenesis and hybridization. For transgenic lines, describe the transformation method, the number of independent lines analyzed and the generation upon which experiments were performed. For gene-edited lines, describe the editor used, the endogenous sequence targeted for editing, the targeting guide RNA sequence (if applicable) and how the editor was applied.* |
| Authentication | *Describe any authentication procedures for each seed stock used or novel genotype generated. Describe any experiments used to assess the effect of a mutation and, where applicable, how potential secondary effects (e.g. second site T-DNA insertions, mosiacism, off-target gene editing) were examined.* |

## ChIP-seq

### Data deposition

☒ Confirm that both raw and final processed data have been deposited in a public database such as GEO.

☒ Confirm that you have deposited or provided access to graph files (e.g. BED files) for the called peaks.

| Data access links<br>*May remain private before publication.* | All mouse sequencing data that support the findings of this study have been deposited on GEO, with the accession code GSE236580 (token for reviewers: qtcrauuovrghlcr). |
|---|---|
| Files in database submission | Mouse data: BED and fastq (in tar format) files for all indvidual CUT&Tag and ATAC-seq;<br><br>ATAC-seq:<br>GSM7558081 C_short_ATAC_1<br>GSM7558082 H_short_ATAC_1<br>GSM7558083 CC_short_ATAC_1<br>GSM7558085 HC_short_ATAC_1<br>GSM7558086 C_short_ATAC_2<br>GSM7558087 H_short_ATAC_2<br>GSM7558088 CC_short_ATAC_2<br>GSM7558090 HC_short_ATAC_2<br>GSM7558091 C_short_ATAC_3<br>GSM7558092 H_short_ATAC_3<br>GSM7558093 CC_short_ATAC_3<br>GSM7558094 HC_short_ATAC_3<br>GSM7558096 CC_long_ATAC_1<br>GSM7558097 HH_long_ATAC_1<br>GSM7558098 CCC_long_ATAC_1<br>GSM7558099 HHC_long_ATAC_1<br>GSM7558101 CC_long_ATAC_2<br>GSM7558102 HH_long_ATAC_2<br>GSM7558103 CCC_long_ATAC_2<br>GSM7558104 HHC_long_ATAC_2<br>GSM7558105 HH_long_ATAC_3<br>GSM7558107 HHC_long_ATAC_3<br><br>CUT&Tag:<br>GSM7558140 CC_long_H3K4me1_1_AdipoERCre<br>GSM7558142 HH_long_H3K4me1_1_AdipoERCre<br>GSM7558143 CCC_long_H3K4me1_1_AdipoERCre<br>GSM7558144 HHC_long_H3K4me1_1_AdipoERCre<br>GSM7558146 CC_long_H3K4me1_2_AdipoERCre<br>GSM7558147 HH_long_H3K4me1_2_AdipoERCre<br>GSM7558148 HHC_long_H3K4me1_2_AdipoERCre<br>GSM7558150 HH_long_H3K4me1_3_AdipoERCre<br>GSM7558151 CCC_long_H3K4me1_3_AdipoERCre<br>GSM7558152 HHC_long_H3K4me1_3_AdipoERCre<br>GSM7558153 CC_long_H3K4me3_1_AdipoERCre<br>GSM7558155 CCC_long_H3K4me3_1_AdipoERCre<br>GSM7558156 HHC_long_H3K4me3_1_AdipoERCre<br>GSM7558157 CC_long_H3K4me3_2_AdipoERCre |

```
GSM7558158 HH_long_H3K4me3_2_AdipoERCre
GSM7558159 HHC_long_H3K4me3_2_AdipoERCre
GSM7558161 HH_long_H3K4me3_3_AdipoERCre
GSM7558162 CCC_long_H3K4me3_3_AdipoERCre
GSM7558163 HHC_long_H3K4me3_3_AdipoERCre
GSM7558164 CC_long_H3K27ac_1_AdipoERCre
GSM7558165 HH_long_H3K27ac_1_AdipoERCre
GSM7558167 CCC_long_H3K27ac_1_AdipoERCre
GSM7558168 HHC_long_H3K27ac_1_AdipoERCre
GSM7558169 CC_long_H3K27ac_2_AdipoERCre
GSM7558170 HH_long_H3K27ac_2_AdipoERCre
GSM7558172 HHC_long_H3K27ac_2_AdipoERCre
GSM7558173 HH_long_H3K27ac_3_AdipoERCre
GSM7558174 CCC_long_H3K27ac_3_AdipoERCre
GSM7558175 HHC_long_H3K27ac_3_AdipoERCre
GSM7558177 CC_long_H3K27me3_1_AdipoERCre
GSM7558178 CCC_long_H3K27me3_1_AdipoERCre
GSM7558179 HHC_long_H3K27me3_1_AdipoERCre
GSM7558181 CC_long_H3K27me3_2_AdipoERCre
GSM7558182 HH_long_H3K27me3_2_AdipoERCre
GSM7558183 HHC_long_H3K27me3_2_AdipoERCre
GSM7558184 HH_long_H3K27me3_3_AdipoERCre
GSM7558186 CCC_long_H3K27me3_3_AdipoERCre
GSM7558187 HHC_long_H3K27me3_3_AdipoERCre
GSM7558188 C_short_H3K4me1_1_AdipoERCre
GSM7558189 H_short_H3K4me1_1_AdipoERCre
GSM7558190 CC_short_H3K4me1_1_AdipoERCre
GSM7558192 HC_short_H3K4me1_1_AdipoERCre
GSM7558193 C_short_H3K4me1_2_AdipoERCre
GSM7558194 H_short_H3K4me1_2_AdipoERCre
GSM7558195 CC_short_H3K4me1_2_AdipoERCre
GSM7558196 HC_short_H3K4me1_2_AdipoERCre
GSM7558198 C_short_H3K4me1_3_AdipoERCre
GSM7558199 H_short_H3K4me1_3_AdipoERCre
GSM7558200 CC_short_H3K4me1_3_AdipoERCre
GSM7558202 HC_short_H3K4me1_3_AdipoERCre
GSM7558203 C_short_H3K4me3_1_AdipoERCre
GSM7558204 H_short_H3K4me3_1_AdipoERCre
GSM7558206 CC_short_H3K4me3_1_AdipoERCre
GSM7558207 HC_short_H3K4me3_1_AdipoERCre
GSM7558208 C_short_H3K4me3_2_AdipoERCre
GSM7558209 H_short_H3K4me3_2_AdipoERCre
GSM7558211 CC_short_H3K4me3_2_AdipoERCre
GSM7558212 HC_short_H3K4me3_2_AdipoERCre
GSM7558213 C_short_H3K4me3_3_AdipoERCre
GSM7558214 H_short_H3K4me3_3_AdipoERCre
GSM7558215 CC_short_H3K4me3_3_AdipoERCre
GSM7558217 HC_short_H3K4me3_3_AdipoERCre
GSM7558218 C_short_H3K27ac_1_AdipoERCre
GSM7558219 H_short_H3K27ac_1_AdipoERCre
GSM7558220 CC_short_H3K27ac_1_AdipoERCre
GSM7558222 HC_short_H3K27ac_1_AdipoERCre
GSM7558223 C_short_H3K27ac_2_AdipoERCre
GSM7558224 H_short_H3K27ac_2_AdipoERCre
GSM7558226 CC_short_H3K27ac_2_AdipoERCre
GSM7558227 HC_short_H3K27ac_2_AdipoERCre
GSM7558228 C_short_H3K27ac_3_AdipoERCre
GSM7558229 H_short_H3K27ac_3_AdipoERCre
GSM7558231 HC_short_H3K27ac_3_AdipoERCre
GSM7558232 C_short_H3K27me3_1_AdipoERCre
GSM7558233 H_short_H3K27me3_1_AdipoERCre
GSM7558234 CC_short_H3K27me3_1_AdipoERCre
GSM7558235 HC_short_H3K27me3_1_AdipoERCre
GSM7558236 C_short_H3K27me3_2_AdipoERCre
GSM7558238 H_short_H3K27me3_2_AdipoERCre
GSM7558239 CC_short_H3K27me3_2_AdipoERCre
GSM7558240 HC_short_H3K27me3_2_AdipoERCre
GSM7558241 C_short_H3K27me3_3_AdipoERCre
GSM7558242 CC_short_H3K27me3_3_AdipoERCre
GSM7558244 HC_short_H3K27me3_3_AdipoERCre
```

Genome browser session
(e.g. UCSC)

No genome browser session was created.

## Methodology

| | |
|---|---|
| Replicates | CUT and TAG : 3 biological replicates for C, CC_s, H, HC, HH, HHC for each hPTM, 2 biological replicates for CC_l and CCC for each hPTM. |
| Sequencing depth | Minimum of of 6 million reads per sample. Paired end sequencing, with each read 150 bp length. |
| Antibodies | anti-H3K4me3 (abcam, #ab8580), anti-H3K27me3 (Cell Signaling Technology, #C36B11), anti-H3K27ac (abcam, #ab4729), anti-H3K4me1 (abcam, #ab8895) |
| Peak calling parameters | Peaks were called from CUT&Tag-seq and ATAC-seq libraries on individual bedgraph files using SEACR v1.3 in stringent mode with a peak calling threshold of 0.01. |
| Data quality | Analysis code for mouse data is available on GitHub (https://github.com/vonMeyennLab/AT_memory) |
| Software | Analysis code for mouse data is available on GitHub (https://github.com/vonMeyennLab/AT_memory) |

