## [Peer Review file · Nature]

Manuscript Title: Adipose tissue retains an epigenetic memory of obesity after weight loss

Reviewer Comments & Author Rebuttals

Reviewer Reports on the Initial Version:

Referees' comments:

Referee #1 (Remarks to the Author):

In this article the authors examine the concept of metabolic memory in the setting of obesity and weight loss. This is a very nicely done study where they examine the role of diet and WL on epigenetic memory and gene programming in adipocytes – both using human samples and murine models. Overall I think the study is well done, I have the following specific comments:

1. For Figure 1, it would be useful to also compare subjects that lost weight through diet and exercise to confirm that some of these findings are not related to the bariatric surgery itself. It is possible that the surgery itself alters the cell phenotypes as opposed to the weight loss.
2. The conclusion from Figure 1 can be misleading – where they state that the more weight you lost the less metabolic memory you have – this could be due to having less fat tissue (hence less inflammatory cytokines in your adipose tissue milieu) and this could be contributing to the poised cells seen in the snRNAseq analysis. It would be useful to understand the relative amount of inflammatory cytokines present in the adipose tissue (as this acts often as the upstream signal for inflammatory macrophages and other cells involved in inflammation).
3. Expression of the key genes related to inflammation or ECM in Fig 4 – that were identified through ATAC and other sequencing methods should be validated – These cells should be sorted and examined to confirm that they are expressing increased levels of some of these key genes identified and called out on the sequencing. The same should be done related to ATAC seq where ChIP PCR could confirm the H3K4me3/H3K27me3 at the promoters of these key genes in cells isolated to confirm the sequencing findings.
4. As part of Figure 6, would be important to also phenotypically characterize these cells by flow cytometry and/or other methods. It is unclear if the phenotype of these cells is different.

Referee #2 (Remarks to the Author):

This is a very well written and interesting article by von Meyenn and colleagues describing mouse and human data on epigenetic memory of adipose tissue cells after weight loss. A strength of the paper is snRNAseq in both humans and mice, allowing for inclusion of data from adipocytes, precursors, and immune cells. The extensive epigenetic analysis was truly a tour de force. The results will be interesting to a broad audience.

Major Comments

1. Do the authors have the weight history of the bariatric surgery patients. They might be able to determine whether the adipocyte memory phenotype correlates with the number of weight cycles.
2. The T cell data is interesting and could be followed up on. It would be valuable to subset the T cells so that changes in CD8 and Tregs could be determined.
3. Trem2 data should be added to extended figure 1c.
4. The entire weight trajectory, both high fat and low fat phases, should be shown in figure 2b and c.
5. Subsetting of cell populations for mouse transcriptomic data would greatly enhance the conclusions that can be made from this dataset. For example, CD8 T cells and Tregs have drastically different roles in adipose tissue and really can't be assessed as one population.
6. Extended figure 7d – a 2-way ANOVA should be performed. While true the increment above the curve is not different, the baseline glucose is quite different. The same is true for the ITT data in ext 7f-g. This will change the interpretation of the data and actually makes more sense given that the CCH mice weigh less than the HCH mice.
7. There is quite a bit of variability in the body weight of the mice. How were mice selected for the transcriptomic and epigenetic analyses?

Minor comments

1. The statistical symbols in Figure 1h seem to not always be aligned properly.
2. The naming of the various mouse groups was not intuitive to me and also were not used consistently throughout the manuscript.
3. Please be clear as to whether the 10% diet is a true chow diet or is a formulated diet to match the micronutrients of the 60% high fat diet.
4. Authors should discuss why the HH mice did not have worse glucose tolerance than the H mice. Was glucose dosed to body weight or lean body mass? Others have shown that mice can start to lose epididymal fat mass after such a long time on high fat diet. The authors should reference the work of Strissel and Obin (doi: 10.2337/db07-0767) showing that this is due to adipocyte apoptosis.
5. I think the label for Figure 2n should be CC_s instead of CC. Also, it would be helpful to have the y-axis for Fig 2n and 2o on the same scale.
6. Fig 2p y-axis should be labeled as Triglycerides before the units.
7. Subclasses of macrophages in Fig 3c are not provided – i.e. the panel needs a key.
8. Authors should be careful of their conclusions for difference in HC and HHC adipocytes in extended fig 7a as they only have 3-4 samples per group and are likely underpowered to detect differences.

Alyssa Hasty

Referee #3 (Remarks to the Author):

Epigenetic memory represents an important area of research in metabolic disease. In this report Hinte et al. have explored whether adipose tissue retains an epigenetic memory of prior obesity. Data are derived from human adipose tissues and mouse interventions.

Primary strengths of this work include the significance of the topic, use of single-cell transcriptome and epigenome inquiry and the use of “Yo-Yo” diet murine models to support human clinical observations. Overall, the manuscript is well written, the study comprehensive, and further benefits from the use of state-of-the-art single cell transcriptome and epigenome analysis techniques. Concepts and approaches should be useful for the study of epigenetic memory in other tissue types and disease models, which the author may consider highlighting in the discussion section of this work. There are however several aspects (described below) that need to be addressed and can help enhance the impact of the study. are described below.

Major comments:

- 1) The use of human visceral adipose tissue samples for single-cell analysis is insightful into the effects of bariatric surgery and the retention of obesogenic signatures even after significant weight loss.
 - a. Authors suggest that weight loss “diminished the strength and numbers of interactions between cell types, suggesting a possible return to a healthier state.” It will be helpful to present such data for lean individuals in figure 1c, to see the relative resemblance to lean condition.
 - b. In this section, authors state that the change in cell abundances between obese and weight loss conditions might “also be affected by sampling effects.” It is unclear which sampling effects are of potential concern (batch, biopsy differences between lean/obese individuals, etc). This leads one to consider how significant sampling errors may be between visAT biopsies. An estimate of sampling variability may be useful for supplemental information, and additional clarity may be needed here.
 - c. In figure 1f, it is unclear why Reactome 2022 pathway analysis was utilized for upregulated transcripts, whereas downregulated transcripts were assessed by WikiPathways 2019. To avoid results bias, both up and downregulated sets should be assessed by the same databases, both being fine, and additional results provided in supplemental material. The rationale for choosing these analytical tools should be provided
 - d. Data sets analyzed include samples from 32 females and 12 males. There is significant variability in gene expression of select genes correlated with BMI loss in figure 1i. Are their sex-specific differences? One point of additional clarity. This data indicates that 43 individuals were assessed, but in figure 1a n=22 total samples are analyzed. Was gene expression analyzed for the 43 individuals using a different approach? Methods and figure legend descriptions should be revised for clarity.
 - e. Line 107, page 4. Authors suggest “individuals with weaker transcriptional memory of obesity experience greater weight loss (WL).” It is recommended that this topic be revisited in the discussion as potential mechanisms that some individuals may erase transcriptional memories more effectively (diet vs exercise vs longevity of initial obesity).
- 2) As related to the experimental design and results in figure 6c, previously obese mice rebound more rapidly to a state of obesity, and the effect is greater for those mice previously fed high fat

diets longest. Though perhaps impractical to repeat most assays, weight loss tracking may be reasonable to establish how long, if possible, reversal of obesogenic memory may take. That is, in both HC and HHC conditions weight normalization is achieved by as early as 4 weeks. Can additional time on a normal chow diet erase the weight regain tendency? What may be the effects of a low-calorie intervention at this point on both populations on such memories?

3) Line 334, page 10. Authors conclude that “4 hPTMs and ATAC-seq could predict or explain 31% of upregulated DEGs, which were related to inflammation, and 60% of downregulated DEGs...in the HCH group.” Here, and elsewhere when predicting histone and accessibility connections with differential expression, or in the discussion, authors should consider which other factors may explain the global transcriptional effects. That is, what other epigenetic factors may be of prime interest for future studies?

4) Line 416, page 12. Authors hypothesize that epigenome remodeling may reduce obesity associated rebound effects. Though locus-specific epigenetic editing may be impractical for this study currently, implementation of a global epigenetic modifier such as TSA or similar, may alleviate the rebound effect. A pilot study for weight tracking is reasonable, which would benefit further with epigenomic assessment of related histone modifications. As related, TSA was shown to reverse adipocyte size increases and weight gain by high fat diets in mice. PMID: 34232916.

5) As related to point 3 and 4, authors do not describe data on epigenetic modifiers which may be at play in defining epigenetic obesity memories. Expression data should be presented on primary epigenetic factors (e.g. HMTs, HDACs, DNMTs, TETs) and networks, which are likely altered between obese, lean, and weight loss individuals and mice, and available in their current datasets.

6) In figure 1c, human scRNA-seq data is assessed for interaction strengths between cell types indicating weight loss to diminish the strength of interactions between cell types. Such logic and assessments will also be usefully applied to murine data before and after weight changes and examined for consistency with human data.

Other minor concerns

a. Scale bars need to be clearer in most histology images.

b. Line 198, page 6 regarding adipocyte biology. Authors should provide a standard citation.

c. The authors may want to elaborate more on the decrease in proportion of adipocytes presented in figure 6j. How does this compare to adipocyte concentration overall in fat tissue of CCH and HCH mice?

d. The authors use the word “imprinting” in the abstract (line 22, page 3). They may wish to consider using alternative terminology to avoid confusion with DNA methylation imprinting during early germ cell and embryo development.

e. The authors could cite prior studies that have demonstrated the role of epigenetic changes (like DNA methylation) in metabolic memory of complications in subjects with diabetes.

Referee #4 (Remarks to the Author):

In this manuscript, titled “Adipose tissue retains an epigenetic memory of obesity that persists after weight loss,” Hinte and colleagues explore the possibility that certain cell types in adipose tissue are epigenetically reprogrammed in obese individuals, and that after weight loss there is retention of some of these epigenetic marks which preserve transcriptional signatures of obesity and might make it more challenging to maintain weight loss. The implications of this hypothesis have significant impact on human health and obesity management, and as the authors discuss, other studies have also explored epigenetic memory conferred by obesity. This study includes single-cell measurements of both mouse and human adipose tissue from healthy individuals as well as samples from adipose tissue of obese individuals before and after weight loss. Single-cell measurements allow for cell-type-specific understanding of molecular and cellular perturbations caused by obesity, which makes this study distinct.

Overall, the potential impact of this study is very significant and would be of great interest to a broad scientific community given the unique human cohort study. However, there are some critical issues that would first need to be addressed. The underlying design of the experiments in this study compares adipose tissue cells from healthy individuals (mouse and human) to adipose tissue cells from a different set of individuals before and after weight loss. Epigenetic reprogramming is therefore claimed to be observed if there are epigenetic or transcriptional signatures that are more similar between the samples taken before and after weight loss in obese individuals, than between the healthy individuals and the weight loss samples. The mouse studies in this manuscript are, for the most part, rigorous and well controlled, presenting a comprehensive picture of the mechanism of obesity induced epigenetic reprogramming, as well as a thorough understanding of the epigenetic signals that are preserved after weight loss and the nature of the transcriptional memory that results from these preserved marks. The human data comes from a very valuable set of experiments in which adipose tissue from a cohort of individuals was sampled before bariatric surgery, and two years after the surgery, enabling comparison of obese tissue and post-weight-loss tissue to lean/healthy adipose tissue. The conclusions drawn from the human study, however, are not well supported by the observations, as these experiments are less well controlled. Specifically, the conclusion that there is an epigenetic memory conferred by obesity in human adipose tissue that is retained after weight loss is not supported by the data and should not contribute to the conclusions proposed in this manuscript, as currently presented. In fact, the human studies make up only a very small portion of this manuscript. Before publication, this section should undergo major revisions or be removed completely. Additionally, the comments and questions below should be addressed.

1. In the human studies, transcriptional signatures are measured with single-cell RNAseq, and cell-type specific differential gene expression tests are performed between the healthy/lean individuals and the obese individuals before surgery. Those differentially expressed genes (DEGs) are then compared to DEGs detected between the healthy samples and the obese samples after surgery. The authors observed that many of the DEGs that were downregulated between the obese and lean samples, remained downregulated between the weight loss and lean samples, and concluded that this is evidence of a “transcriptional memory” of obesity in the weight loss samples. However, in these experiments the observed effect is confounded with other sources of variation between the individuals including genotype and behavior. Therefore, these experiments do not control for other

non-epigenetic mechanisms that might cause the observed signal, such as the possibility that the genetic background of the obese individuals determines the DEGs, or that the dietary behavior of the obese individuals produces differential gene expression. Indeed, monogenetic obesity was not formally excluded in this study and the individuals who participated in this study did not adhere to any specific diet before or after surgery. Therefore, it can not be concluded that differential gene expression that is not restored after weight loss results from a “transcriptional memory” of obesity as opposed to being the result of a differential phenotype of the individuals studied.

2. Continuing from point 1, the distinction between “epigenetic memory,” “transcriptional memory,” “obesogenic memory,” and “metabolic memory” needs to be made more clear in this manuscript. The authors seem to use these terms interchangeably, but it can be confusing or even misleading at times. The concept of epigenetic memory is well defined in the literature and refers to a broad range of epigenetic features, including DNA methylation, histone modification, and chromatin organization, that be passed to progeny cells conferring distinct transcriptional phenotypes. However, it is not clear what the authors mean by transcriptional memory in the human study. The term “memory” implies an epigenetic mechanism, but the observation of a persistent transcriptional phenotype before and after weight loss does not necessarily correspond to an epigenetic memory. For example, a lifetime of obesity may create a physiological state that even after bariatric surgery, continues to generate metabolic signals that produce gene expression patterns that are distinct from healthy/lean adipose tissue. While potentially interesting, this phenomenon is distinct from epigenetic memory which is stated in the title of this manuscript and throughout the abstract, introduction and discussion. When specifically addressing the human data, the authors use the term “transcriptional memory” and my concern is that this term is used to evoke the notion of epigenetic memory but simply refers to similar gene expression patterns before and after weight loss.

3. In addition to the fact that the experimental design of the human studies does not control for confounding sources of variation, there is little statistical analysis to support the significance of the observations that there is a set of DEGs that are preserved in obese tissue after weight loss. First, how many individuals were sampled between lean and obese to generate this set of DEGs? Apologies if I missed this, but while samples were collected from 44 obese individuals who experience at least 25% weight loss after surgery, I could not find how many lean individuals were sampled. The authors mention that differences in biopsy sampling could have contributed to the observed variation in cell type abundance. Could sampling effects also have influenced the DEG results? Could sampling effects impact the cellchat interaction results? Is there any statistical test to determine if the results of cellchat analysis are meaningful/statistically significant? Fig 1 g,h,&l shows the relationship between upregulated DEGs and BMI and the authors state a significant negative correlation, but there is no measurement of correlation or significance. This data is used to imply that “individuals with weaker transcriptional memory of obesity experience greater weight loss.” But I am not convinced by this implication given the plots in Fig 1 g,h,&l, which qualitatively present little to no correlation.

4. In the section “Mouse adipocytes retain an obesogenic epigenetic memory after WL” the authors perform multi-omic epigenetic analysis of adipocytes after weight gain and weight loss to identify persistent epigenetic signatures. In order to confirm that the appropriate populations are compared, they use a labeling strategy to track adipocytes in mouse epiAT through a high fat diet. Extended

data figure 5c is used to confirm that the cells that are epigenetically profiled represent the same population of cells profiled with single-nuclei RNAseq. The authors then use Fig 4C to demonstrate a “restoration of the translational profile in adipocytes from HC and HHC mice.” These two measurements seem to be important observations but I am having trouble understanding the analysis. In ED5c, while the spearman coefficients on log2 counts are above 0.7 for all the samples compared, it looks like the genes skew towards the bulk samples, such that a majority of the genes are low counts in the single nuclei samples with a wide range of expression levels in the bulk samples. More importantly, what is the justification for comparing TRAP-seq counts to single nuclei RNAseq counts? In fig 4C how is the conclusion stated above made? The replicates for each condition seem to spread across the PC1 and PC2 space and it is not qualitatively clear that the spatial concordance between any set of replicates indicates an especially similar translome. The analysis and conclusions in lines 213-217 should be clarified.

5. Continuing from point 4, the authors then compare the epigenetic profiles across replicates of each feeding condition. They state that they use multi-omic factor analysis to “overcome limitations of modality-specific analyses” and refer to extended data figure 5d. This figure seems to show broad variation in the positioning of replicates across the first two principal components of cut&tag and atacseq data. However, they then use MOFA to conclude that factor 1 explains the moajor source of variability between conditions which is influenced predominantly by histone modifications. So why then do cut&tag measurements of histone modifications reveal this condition specific variability? I do not understand how MOFA overcomes the modality-specific analysis. Furthermore, much of this principal component analysis is qualitative, identifying regional clustering of replicates. The authors should perform quantitative analysis of the geometric distances between condition means (centers of mass) instead of relying on qualitative interpretations of clustering in PCA space. Because this section provides some of the most compelling observations of epigenetic memory, it is very important that the analysis is clearly motivated and interpreted, and implements rigorous statistical methods.

6. Same points as above for figures 5 b&c and ED6 d&e. The authors observe that certain post-weight loss feeding conditions clustered more closely to pre-weight loss conditions than to controls, indicating retention of epigenetic signals. These distances must be quantified in order to support this conclusion. It is not even qualitatively clear that the HC and HHC conditions are more epigenetically similar to the H and HH conditions than to the controls, based on the PC analysis.

Author Rebuttals to Initial Comments:

Rebuttal letter to the manuscript: “Adipose tissue retains an epigenetic memory of obesity that persists after weight loss”

We thank the editor and all the referees for their careful assessment of our work and their constructive comments, questions, and suggested experiments/analyses. We were very happy to read that the comprehensive studies and epigenetic characterisations in the mouse model were acknowledged by all the referees to be nicely done and support our conclusion that (mouse) adipose tissue retains an epigenetic memory of obesity that persists after weight loss. The referees also found that the use of human adipose tissue samples for single-cell analysis to study cell-type-specific effects of (bariatric surgery induced) weight loss on the transcriptional level was insightful and distinct from prior studies. However, they also voiced some concerns particularly with regard to the human snRNA-seq datasets and analyses. Here, we have now addressed all the referees’ comments and suggestions in full and believe that the manuscript has significantly improved.

In summary, we made these major changes:

- We obtained unique additional human omental and subcutaneous adipose tissue biopsies pre- and post-weight loss from additional studies and verified our original observations (see below).
- We completely revised Figure 1 and the associated Extended Data and text (see below).
- We did not use the term transcriptional or epigenetic “memory” in the human results section to describe the effects of obesity-induced transcriptional (or cellular) changes/differences in obese and post- weight-loss adipose tissue.
- We clarified and updated the Methods, and we provided a new additional table with the clinical metadata.
- We provided additional details and experimental work expanding and supporting our mouse studies.

We hope the reviewers share our enthusiasm for this work and that they find our revisions of the manuscript sufficient to warrant publication in *Nature*.

COMMENT: Several Figures in the rebuttal letter are replicated from the main manuscript and placed directly with the answers to improve readability. We have clearly indicated where the same data is presented in the revised manuscript and sometimes even kept the labels of the panels from the manuscript to make the identification very clear.

We have also updated the numbering of Figures and Tables in the Manuscript and refer in the rebuttal letter always to the new revised numbering scheme.

Additional omental and subcutaneous adipose tissue biopsies

To strengthen our analyses, we have obtained additional omental and subcutaneous adipose tissue biopsies sampled pre- and post-weight loss from two bariatric surgery studies from Germany and Sweden, increasing our sample number from 6 lean and 8 donors living with obesity to a total of **18 lean/normal-weight and 20 donors living with obesity** (each sampled pre and post weight loss). The detailed description of how these biopsies were obtained is included in the **Methods** section and summarized in the table below. We have also included a table with clinical metadata in new **Extended Data Table 1**. Consequently, we have completely revised **Figure 1** and the associated text and the **Extended Data Figures 1,2**.

In total we present data from AT biopsies from these three studies:

Study	Acronym	Hospitals	Surgery at T0	Lean	Obese	Tissues
Multicentre Two Step Surgery (used in original MS)	MTSS	Karlsruhe Municipal Hospital and Dresden-Neustadt Municipal Hospital, Germany	sleeve gastrectomy	1x male 4x female	2x male 6x female	omAT
Leipzig Two Step Surgery	LTSS	Leipzig University Hospital, Germany	sleeve gastrectomy	2x male 3x female	2x male 3x female	omAT and scAT
NEFA	NEFA	Karolinska University Hospital, Sweden	RYGB	8x female	7x female	scAT

In analysing new datasets, we could confirm that cell-specific transcriptional differences in human adipose tissue were retained after weight loss in samples of each study. Adipocytes were amongst the cell types displaying the strongest deregulation of gene expression both before and after bariatric surgery (**Extended Data Fig. 1g-k**). Of note, weight loss in the participants of the NEFA study (RYGB) did result in a complete return to a non-obese or lean state, and control samples were BMI matched to post surgery samples. As such, our analyses of these samples are likely a reflection of obesity-induced differences and their retainment.

SNP based demultiplexing of pooled human snRNA-seq

Further, we performed SNP-based demultiplexing of our pooled snRNA-seq data to obtain nuclei at an individual level for quality control and variability analyses across all donors. Of note, we do not have SNP information for each donor and therefore we cannot assign these “snRNA-seq-individuals” to tissue donors and perform correlation analyses of adipocyte gene expression and clinical parameters.

Kendall correlation analysis

During the revision and on request of the referees, we re-evaluated the conclusions from the original Kendall correlation analysis and concluded that this type of analysis has several drawbacks limiting the interpretability and found that the original conclusions could be misleading. We therefore decided to remove the Kendall correlation analysis of expression of retained DEGs (former Fig. 1h) and associated statements from the manuscript. We also think

that our additional snRNA-seq data are of major relevance, while the Kendall correlation analysis no longer adds substantial value and thus was deleted from the revised manuscript.

Even so, we would like to clarify how this analysis was done and which samples were used for it. The correlation analysis between gene expression at T1 (post weight loss) and clinical parameters was performed with bulk RNA-seq data from 44 obese omAT samples from 44 individuals with at least 25% BMI loss at T0. Of note, omAT at T0 and T1 from 8 individuals was also used for the snRNA-seq analyses (MTSS study). No omAT samples from lean donors were used.

We now performed the correlation analysis again using the new set of DEGs obtained from our new analyses using the MTSS and LTSS snRNA-seq data.

While several adipocyte DEGs correlated to BMI loss similarly to our initial observation (**Reviewer Figure 1**), we are of the opinion this correlation analysis has several drawbacks limiting the interpretability, some of which were also pointed out by reviewers:

- i) Kendall correlation is non-parametric and does not assume linear relationships
- ii) we cannot perform this analysis for the other studies (LTSS and NEFA) as we don't have corresponding bulk RNA-seq data
- iii) all DEGs might be also expressed in other cell types which can confound this analysis
- iv) we cannot address whether environmental, genetic factors, or circulating cytokines etc. influence the correlation

Reviewer Figure 1: Kendall correlation of gene expression at T1 in omAT of the MTSS study and clinical parameters including fasting plasma glucose (FPG) and fasting plasma insulin (FPI). * $P < 0.05$, ** $P < 0.01$, *** $P < 0.001$.

Based on these limitations, we have not included this analysis in the revised manuscript, and we also removed the information related to the human bulk RNA-seq data, which was not used in the revised manuscript.

Referee #1 (macrophage/metabolic physiology):

In this article the authors examine the concept of metabolic memory in the setting of obesity and weight loss. This is a very nicely done study where they examine the role of diet and WL on epigenetic memory and gene programming in adipocytes – both using human samples and murine models. Overall, I think the study is well done, I have the following specific comments:

We thank the referee for their overall positive assessment of our work.

1. For Figure 1, it would be useful to also compare subjects that lost weight through diet and exercise to confirm that some of these findings are not related to the bariatric surgery itself. It is possible that the surgery itself alters the cell phenotypes as opposed to the weight loss.

This is an interesting point raised by the referee. Indeed, the cellular and molecular changes during weight loss induced by dietary interventions or bariatric surgery could be different, and bariatric surgery could result in other systemic effects that can affect transcription. As our comprehensive mouse study was based on a dietary intervention, we would have ideally also collected biopsies obtained from dietary intervention studies leading to significant weight loss. We are, however, only aware of one (ongoing) clinical trial (NCT02706262) that collects subcutaneous adipose tissue biopsies pre- and post-weight loss, although we have no access to these samples. To our knowledge, no dietary study collects or has collected omental adipose tissue. Equally, we ourselves do not have the expertise nor setup to collect such biopsies in the foreseeable future.

Nonetheless, we were able to include further human AT samples in our study from individuals of two additional bariatric surgery studies (LTSS, NEFA; please see above and in the **Methods** section of the manuscript for more details). These include omental adipose tissue and also subcutaneous abdominal adipose tissue samples.

Of note, we have now been able to demultiplex all pooled human snRNA-seq datasets by calling SNPs for each cell and individual (please see above and in the **Methods** section of the manuscript for more details). In agreement with our original findings, pooled snRNA-seq showed for all donors, that adipocytes, APCs and endothelial cells displayed the most deregulated genes during obesity (**Reviewer Figure 2** and new **Extended Data Fig. 1g-j**) and each donor retained transcriptional changes post weight loss.

Reviewer Figure 2 (data also presented in Extended Data Fig. 1g-j): Number of DEGs (up or down) per cell type per donor at T0 scaled by column. From left to right. MTSS study omAT, LTSS study omAT, LTSS study scAT, NEFA study scAT.

2. The conclusion from Figure 1 can be misleading – where they state that the more weight you lost the less metabolic memory you have – this could be due to having less fat tissue (hence less inflammatory cytokines in your adipose tissue milieu) and this could be contributing to the poised cells seen in the snRNA-seq analysis.

It would be useful to understand the relative amount of inflammatory cytokines present in the adipose tissue (as this acts often as the upstream signal for inflammatory macrophages and other cells involved in inflammation).

We agree with the referee, that the conclusions from the original Kendall correlation analysis could be misleading. As this type of analysis has several drawbacks limiting the interpretability (please see above), and we given that are now including additional datasets supporting our original findings related to retained transcriptional changes post weight loss, we have decided to remove the Kendall correlation analysis and associated statements from the manuscript.

The referee also points out that sustained inflammation or increased local cytokine concentrations can contribute to the observed transcriptional retainment. This might even differ from person to person or could be linked to the degree of weight or adipose tissue/fat mass loss. Unfortunately, the human AT samples available for our study were small and not sufficient to measure cytokine concentrations in these human adipose tissue samples.

As the abundance of immune cells might be an (indirect) indicator of the level of inflammation of the tissue and therefore influenced by cytokine concentration, we quantified the relative abundance of immune cells after weight loss. We did not observe significant differences in omAT in the MTSS and LTSS study (**Reviewer Figures 3 and 4** and new **Extended Data Figure 2b,e,h,k**) and a reduction of LAMs and other macrophages only in the scAT samples of the NEFA study but not in scAT samples of the LTSS study.

Reviewer Figure 3 (data also presented in Extended Data Fig. 2b,e,h,k): **a**, Barplot depicting the relative cell type abundance in omAT of the MTSS study per condition. **b**, Barplot depicting the relative cell type abundance in omAT of the LTSS study per condition. **c**, Barplot depicting the relative cell type abundance in scAT of the LTSS study per condition. **d**, Barplot depicting the relative cell type abundance in scAT of the NEFA study per condition.

Reviewer Figure 4: **a**, Relative cell type abundance per donor within immune cells of omAT of the MTSS study. **b**, Relative cell type abundance per donor within immune cells of omAT of the LTSS study. **c**, Relative cell type abundance per donor within immune cells of scAT of the LTSS study. **d**, Relative cell type abundance per donor within immune cells of scAT of the NEFA study. Wilcoxon matched-pairs signed rank test were used to test for significance between T0 and T1. * $P < 0.05$

3. Expression of the key genes related to inflammation or ECM in Fig 4 – that were identified through ATAC and other sequencing methods should be validated – These cells should be sorted and examined to confirm that they are expressing increased levels of some of these key genes identified and called out on the sequencing. The same should be done related to ATAC seq where ChIP PCR could confirm the H3K4me3/H3K27me3 at the promoters of these key genes in cells isolated to confirm the sequencing findings.

We thank the referee for the comment and would like to clarify our methodological approach. All epigenetic and transcriptional data presented in **Figures 4** (and **5**) was generated from “purified” labelled adipocytes (AdipoERCre x NuTRAP; please see **Methods** for details), i.e. in these mice the adipocyte nuclei are marked with biotin and mCherry, the ribosomes are marked with GFP. Using this reporter system, we were able to isolate/purify the biotinylated adipocytes nuclei using a streptavidin pulldown approach and also collect adipocyte-specific

ribosome bound mRNA using anti-GFP antibodies. We added additional details also to the manuscript main text (Lines 234 - 238).

As such, the transcriptional and epigenetic data presented in **Figure 4** and **5** is equivalent to data that would have been generated from sorted adipocytes. We also generated replicates for every assay and the overall patterns and distribution for the different modalities (4 hPTMs and ATAC-seq) are in line with the expectations for these hPTMs. Importantly, RNA-seq, ATAC-seq, Cut&TAG, and ChIP-Seq, while being more expensive, also provide significant advantages over single gene or single locus assays such as ChIP-qPCR or RT-qPCR, i.e. the results are not dependent on the selected control gene or region, provide genome-wide information, and can be normalised internally.

4. As part of Figure 6, would be important to also phenotypically characterize these cells by flow cytometry and/or other methods. It is unclear if the phenotype of these cells is different.

We understand that the referee is referring here to a characterisation of adipocytes after the HFD rechallenge. In line with the remaining work in our manuscript, we performed snRNA-seq to gain first transcriptional insights and relate these to the epigenetic changes prior to the HFD rechallenge (**Fig. 6i-m**). Further, phenotypic characterisation of adipocytes is challenging. Retaining healthy functional mature adipocytes after FACS from HFD fed mice represents a major challenge and we have therefore not been able to phenotype the adipocytes beyond *in vitro* tests. Particularly, large lipid-filled adipocytes are prone to damage during FACS which could result in a bias towards only small and poorly lipid-laden adipocytes surviving this procedure.

Nonetheless, we were able to collect and culture mature adipocytes after the HFD rechallenge using the protocol also used after weight loss and collected data on palmitate and glucose uptake from adipocytes (**Reviewer Fig. 5**). Palmitate uptake was not different between CCH and HCH adipocytes from epididymal or inguinal adipose tissue (**Reviewer Fig. 5a,b**). This might be because HCH adipocytes are larger at this time point (and the assay is normalised to protein content) and cannot take up more than control adipocytes (**Reviewer Fig. 5c,d**). At the same time CCH adipocytes have also been exposed to HFD now and might be as well primed to respond to palmitate. HCH adipocytes of epididymal AT displayed a diminished glucose uptake compared to CCH adipocytes (**Reviewer Fig. 5e**). The same trend was seen for adipocytes from inguinal AT (**Reviewer Fig. 5f**). This could indicate a decreased insulin sensitivity in HCH adipocytes, which would be in line with the observation of slight hyperinsulinemia and hyperglycaemia in HCH mice (**Fig. 6d,e**).

Reviewer Figure 5: a, BODIPY-C16 uptake of isolated, cultured primary adipocytes from epiAT from CCH and HCH mice. Each dot represents an individual biological replicate of a pool of 3 mice. **b**, BODIPY-C16 uptake of isolated, cultured primary adipocytes from ingAT from CCH and HCH mice. Each dot represents an individual biological replicate of a pool of 3 mice. **c**, Distribution of epiAT adipocyte area across conditions. ($n = 4$ mice, 5-8 pictures each). **d**, Distribution of ingAT adipocyte area across conditions. ($n = 4$ mice, 5-8 pictures each). **e**, Glucose uptake in response to 5 nm insulin after 10 minutes of isolated, cultured primary adipocytes from epiAT from CCH and HCH mice. Each dot represents an individual biological replicate of a pool of 3 mice. **f**, Glucose uptake in response to 5 nm insulin after 10 minutes of isolated, cultured primary adipocytes from ingAT from CCH and HCH mice. Each dot represents an individual biological replicate of a pool of 3 mice.

Referee #2 (immune cells/adipose/weight loss):

This is a very well written and interesting article by von Meyenn and colleagues describing mouse and human data on epigenetic memory of adipose tissue cells after weight loss. A strength of the paper is snRNA-seq in both humans and mice, allowing for inclusion of data from adipocytes, precursors, and immune cells. The extensive epigenetic analysis was truly a tour de force. The results will be interesting to a broad audience.

We thank the referee for their positive assessment of our work and its relevance.

Major Comments

1. Do the authors have the weight history of the bariatric surgery patients. They might be able to determine whether the adipocyte memory phenotype correlates with the number of weight cycles.

This is a very good suggestion. Information on weight cycling could provide valuable insights, in particular given our observations regarding long-lasting epigenetic changes in the mouse model. Unfortunately, the clinical data available to us does not provide information related to the weight history of the individuals included. Of note, none of the lean/normal weight donors had been obese. Also, childhood obesity was excluded for all patients, suggesting that none of them has a monogenetic obesity predisposition. Nonetheless, we would speculate that many of the obese tissue donors have experienced weight cycling.

2. The T cell data is interesting and could be followed up on. It would be valuable to subset the T cells so that changes in CD8 and Tregs could be determined.

We thank the referee for their comment and suggestion. Investigating proportional changes of different T-cell subtypes is an interesting experiment in the context of weight loss. The method we used – snRNA-seq – is limiting the depth of our analyses due to technical constraints (i.e. many commonly used cell surface markers of immune cells are not detectable in snRNA-seq). Nevertheless, we have performed analyses of the immune cells in our mouse and human snRNA-seq datasets based on the PBMC reference data set from the Human BioMolecular Atlas Program (<https://portal.hubmapconsortium.org/>). The results are summarized below for each dataset.

snRNA-seq of human omental adipose tissue

MTSS study: Within the snRNA-seq datasets from the MTSS study we detected 735 out of 22,742 nuclei as nuclei of T-cells of which 422 were assigned as CD4 T-cells, 204 Tregs and 109 CD8 T-cells based on reference mapping (**Reviewer Fig. 6a,b**). We did not observe significant changes in abundance of these cells with weight loss (**Reviewer Fig. 6e**).

LTSS study: Within the snRNA-seq datasets from the LTSS study we detected 910 out of 15,347 nuclei as nuclei of T-cells of which 733 were assigned as CD4 T-cells, and 177 CD8 T cells based on reference mapping (**Reviewer Fig. 6c,d**). We did not observe significant changes in abundance of these cells with weight loss (**Reviewer Fig. 6f**).

a

b

c

d

e

f

Reviewer Figure 6: a, UMAP of 3354 nuclei assigned as immune cells from omAT pools of the MTSS study. b, Cluster markers of identified clusters in a. c, UMAP of 2853 nuclei assigned as immune cells from omAT pools of the LTSS study. d, Cluster markers of identified clusters in c. e, Relative abundance of immune cell subtypes per donor in omAT of the MTSS study. Paired Wilcoxon test was used to test for differences between T0 and T1. f, Relative abundance of immune cell subtypes per donor in omAT of the LTSS study. Paired Wilcoxon test was used to test for differences between T0 and T1.

snRNA-seq of human subcutaneous adipose tissue:

LTSS study: Within the snRNA-seq datasets from the LTSS study we detected 194 out of 19,494 nuclei as nuclei of T-cells of which all were assigned as CD4-Tcells (**Reviewer Fig. 7a,b**). We could not annotate other types of T-cells in this dataset. We did not observe significant changes in abundance of these cells with weight loss (**Reviewer Fig. 7e**).

NEFA study: Within the snRNA-seq datasets from NEFA study we detected 232 out of 31,721 nuclei as nuclei of T-cells of which all were assigned as CD4 T-cells (**Reviewer Fig. 7c,d**). We could not annotate other types of T-cells in this dataset. While the CD4 T-cell abundance increased after weight loss (**Reviewer Fig. 7f**), the LTSS study samples did not show this pattern.

Reviewer Figure 7: *a*, UMAP of 1917 nuclei assigned as immune cells from scAT pools of the LTSS study. *b*, Cluster markers of identified clusters in *a*. *c*, UMAP of 5641 nuclei assigned as immune cells from scAT pools of the NEFA study. *d*, Cluster markers of identified clusters in *c*. *e*, Relative abundance of immune cell subtypes per donor in scAT of the LTSS study. Paired Wilcoxon test was used to test for differences between T0 and T1. *f*, Relative abundance of immune cell subtypes per donor in scAT of the NEFA study. Paired Wilcoxon test was used to test for differences between T0 and T1. * $P < 0.05$

snRNA-seq of mouse adipose tissue:

Within our mouse snRNA-seq data of the HFD and weight loss time point (and controls) we could identify 1087 out of ~48,000 nuclei as nuclei of T-cells. Out of these T-cells we can reliably identify 59 as Tregs based on combined expression of *Foxp3* and *Ctla4*, around 48 as CD8 T-cells and only 8 as CD4 (non-regulatory) T-cells. This highlights that at least in our mouse snRNA-seq data these T-cell subtypes are either not detectable due to technical limitations of snRNA-seq or truly low abundant. With ~50 cells across all our conditions, we could not perform meaningful analyses beyond that.

Importantly, despite very sparse, our results regarding T-cell abundance changes are in line with published data. Hata *et al.* (PMID: 36603072) used flow cytometry and T-cell activation assays and could show that the number of T-cells does not change much with weight loss nor does T-cell-mediated cytokine production change: “Although prior exposure to HFD feeding did not grossly affect T cell numbers (Fig. S3, C to G) or T cell-mediated cytokine production (Fig. S3, H to J), a substantial difference in adipose tissue macrophage (ATM) phenotype was observed.”

We did not include the immune cell subtyping analyses in the revised version of the manuscript. We think that an extensive analysis and phenotyping of different T-cells of (human) adipose tissue, while very relevant, is beyond the scope of this publication especially given the results of our analyses and technical limitations of snRNA-seq.

3. Trem2 data should be added to extended figure 1c.

After the reanalysis of the immune compartment of all 4 human snRNA-seq datasets, we did not include *TREM2* as a marker of LAMs because its expression level was lower than that of other markers. This could be due to technical limitations of snRNA-seq. We now identified LAMs via reference mapping based on the PBMC reference data set from the Human BioMolecular Atlas Program (<https://portal.hubmapconsortium.org/>) rather than based on single marker gene expression. We removed the classifications of human macrophage subtypes into lipid scavenging or inflammatory macrophages from the revised manuscript.

4. The entire weight trajectory, both high fat and low fat phases, should be shown in figure 2b and c.

We and many others have repeatedly assessed body weight gain during HFD feeding, but in this study, we decided to focus only on the weight loss period and therefore obtained all wildtype (C57BL/6J) mice from the Jackson Laboratory, including obese and control mice and maintained them for an additional 2 weeks on the corresponding diet to acclimatise them to our environment. Thus, we do not have body weight data during the initial weight gain / HFD feeding period. The exact details are stated in the **Methods** section:

(Lines 550-552): “16- and 29-week-old C57BL/6J DIO (#380050) and DIO control (#380056) male mice were obtained from Jackson Laboratory (USA) and were kept on the respective diets for 2 additional weeks until tissue harvest or diet switch.”

5. *Subsetting of cell populations for mouse transcriptomic data would greatly enhance the conclusions that can be made from this dataset. For example, CD8 T cells and Tregs have drastically different roles in adipose tissue and really can't be assessed as one population.*

We agree with the referee that the composition of the T-cell population could affect our differential gene expression analysis. After reanalysing all human snRNA-seq datasets (see answer to your comment 2) we have removed T-cells from our “transcriptional retention/memory” analyses because the number of nuclei assigned to specific T-cell subtypes is too low to perform any meaningful differential expression analyses. In our mouse snRNA-seq data 115 out of 1087 nuclei were reliably assigned as T-cell subtypes (see answer to your comment 2). Given that each individual T-cell subpopulation comprised only ~50 cells and as such too low to perform any meaningful differential expression analyses, we decided to also remove mouse T-cells from our transcriptional retention analyses.

6. *Extended figure 7d – a 2-way ANOVA should be performed. While true the increment above the curve is not different, the baseline glucose is quite different. The same is true for the ITT data in ext 7f-g. This will change the interpretation of the data and actually makes more sense given that the CCH mice weigh less than the HCH mice.*

We thank the referee for raising this point and we agree that comparing GTTs/ITTs of mice with different body weights or different starting glucose levels can be challenging. This issue has been discussed widely, including between the authors of this manuscript, and also within the scientific community. A recent comment in *Nature Metabolism* (Virtue and Vidal-Puig 2023; PMID: 34117483) provided detailed guidelines for how to interpret GTTs/ITTs in mice. The recommendation is to use the area of the curve to account for different fasting glucose levels and to present the data on fasting glucose levels separately to dissect response to glucose and insulin independently from fasting glucose levels.

Nonetheless we understand the concern of the referee and have performed the requested 2-way ANOVA analysis with Geisser-Greenhouse correction (**Reviewer Fig. 8**). As expected, a 2-way ANOVA analysis of the GTT and ITT data shows significant differences between HCH and CCH mice. This would suggest that HCH mice have impaired glucose tolerance and insulin sensitivity. However, this analysis does not account for the significant different fasting glucose levels (**Fig. 6d**) and this increase in fasting glucose likely explains (or skews) the outcome of the 2-way ANOVA analysis.

Reviewer Figure 8: GTT (a) and ITT (b) from HCH and CCH mice from Extended Data Figure 8 analysed with a 2-way ANOVA.

We ultimately decided to follow the publication mentioned above (Virtue and Vidal-Puig *Nature Metabolism* 2023; PMID: 34117483). Based on that approach, the analysis (now **Extended Data Fig. 8**) shows that glucose dependent insulin secretion and subsequent glucose clearance is not significantly impaired in HCH mice, and HCH mice are still responding to insulin.

7. There is quite a bit of variability in the body weight of the mice. How were mice selected for the transcriptomic and epigenetic analyses?

This is an important point raised by the referee. All weight trajectories shown in the manuscript are obtained from wild-type mice. For the snRNA-seq we always pooled samples from 5 mice making sure the mean body weight reflects the group mean. All epigenetic analysis were performed on samples from AdipoERCre x NuTRAP mice and we selected the samples of animals that were closest to the mean body weight in their group. The bodyweights of these mice at different time points are displayed below (**Reviewer Fig. 9**).

Reviewer Figure 9: *a*, Body weight of AdipoERCre x NuTRAP mice on the day of tissue harvest for each group. *b*, body weight of CC_s, HC, CCC and HHC AdipoERCre x NuTRAP mice from a at the prior time point.

Minor comments

1. The statistical symbols in Figure 1h seem to not always be aligned properly.

Thank you for noting this. We have now completely revised **Figure 1**. We made sure that in the revised **Figure 1** all statistical symbols are aligned in all panels.

2. The naming of the various mouse groups was not intuitive to me and also were not used consistently throughout the manuscript.

Thank you for spotting this inconsistency. We have corrected this mistake in the revised Figures and manuscript.

3. Please be clear as to whether the 10% diet is a true chow diet or is a formulated diet to match the micronutrients of the 60% high fat diet.

This is an important point. Both wild-type DIO mice as well as DIO control mice were kept on micronutrient matched diets. Equally, AdipoERCre x NuTRAP mice received either a 60%

HFD or a matched 10% low fat diet. During the weight loss period, both groups receive the same standard chow diet. We added the necessary information in the **Method** section including the catalogue numbers of the diets or the purchase information from Jackson to the revised manuscript to enhance clarity.

(Lines 557-560): “*The HFD used contained 60% (kcal%) fat (diet no. 2127, Provimi Kliba SA), the low-fat chow diet used contained 10% (kcal%) fat (diet no. 2125, Provimi Kliba SA). During the weight loss period both experimental groups received chow diet (diet no. 3437, Provimi Kliba SA).*”

(Lines 550-552): “*16- and 29-week-old C57BL/6J DIO (#380050) and DIO control (#380056) male mice were obtained from Jackson Laboratory (USA) ...*”

4. Authors should discuss why the HH mice did not have worse glucose tolerance than the H mice. Was glucose dosed to body weight or lean body mass? Others have shown that mice can start to lose epididymal fat mass after such a long time on high fat diet. The authors should reference the work of Strissel and Obin (doi: 10.2337/db07-0767) showing that this is due to adipocyte apoptosis.

We thank the referee for raising this point. For all GTTs in this manuscript glucose was dosed to total body weight given that lean body mass would not account for the differences in the amount of adipose tissue present in DIO or control mice (see **Methods** section). Indeed, our manuscript shows that HH mice had a smaller epiAT than CC_1 mice (**Fig. 2s and Reviewer Fig. 10**) and epiAT displayed infiltration and fibrosis (**Fig. 2t,u and Reviewer Fig. 10**). Therefore, it is possible that this could affect our GTT results. We, however, think that the “non-impaired” glucose tolerance of HH mice could also be explained by the fact that HH mice were very hyperinsulinemic (**Fig. 2o and Extended Data Fig. 3a and Reviewer Fig. 10**) which could result in faster glucose clearance.

Fig 2.

Extended Data Fig. 3

Reviewer Figure 10 (data also presented in Fig. 2s-u,o and Extended Data Fig. 3a): **s**, weights of ingAT, epiAT and BAT, normalized to body weight ($n = 6$ HH and CC_I, $n = 10$ HHC and CCC). **t**, representative photos of epiAT depots. Ruler is in cm. **u**, Tissue histology of epiAT with haematoxylin and eosin (H&E) staining, 20x magnification. Scale bar, 100 μ m. Representative pictures. **o**, fasting insulin levels ($n = 6$ each). **a**, Postprandial insulin. Each dot denotes a biological replicate from 2-3 experiments, $n = 5-8$; Boxplot represents minimum, maximum and median.

We have now also referred to the work of Strissel and Obin (ref. 44) in our manuscript.

Lines 165-167: “Interestingly, the phenomenon of epiAT shrinkage was already observed during obesity in 25-week HFD (HH)-fed mice **as previously reported**⁴¹ and maintained after WL in HHC mice (Fig. 2s,t and Extended Data Fig. 3g,h).”

5. I think the label for Figure 2n should be CC_s instead of CC. Also, it would be helpful to have the y-axis for Fig 2n and 2o on the same scale.

Thank you for noting this. We have adjusted the scales and labels.

6. Fig 2p y-axis should be labelled as Triglycerides before the units.

Thank you for noting this. We have changed the axis label.

7. Subclasses of macrophages in Fig 3c are not provided – i.e. the panel needs a key.

Thank you for noting this. We included the missing colour key.

8. Authors should be careful of their conclusions for difference in HC and HHC adipocytes in extended fig 7a as they only have 3-4 samples per group and are likely underpowered to detect differences.

Thank you for raising this potential limitation. We have changed the text of the manuscript to address this. Indeed, we only had 3-4 independent pools of mice per group.

Lines 338-340: *“In *ingAT*, adipocytes from HHC mice displayed significant increased glucose uptake compared to controls, and for HC adipocytes we observed a trend towards an increased uptake (Extended Data Fig. 8a.)”*

Referee #3 (metabolic physiology/epigenetics): Prof. Dr. Alyssa Hasty

Epigenetic memory represents an important area of research in metabolic disease. In this report Hinte et al. have explored whether adipose tissue retains an epigenetic memory of prior obesity. Data are derived from human adipose tissues and mouse interventions.

Primary strengths of this work include the significance of the topic, use of single-cell transcriptome and epigenome inquiry and the use of “Yo-Yo” diet murine models to support human clinical observations. Overall, the manuscript is well written, the study comprehensive, and further benefits from the use of state-of-the art single cell transcriptome and epigenome analysis techniques. Concepts and approaches should be useful for the study of epigenetic memory in other tissue types and disease models, which the author may consider highlighting in the discussion section of this work. There are however several aspects (described below) that need to be addressed and can help enhance the impact of the study. are described below.

We thank Prof. Alyssa Hasty for her positive assessment of our manuscript and approaches, and for acknowledging the relevance for the scientific community.

Major comments

1) The use of human visceral adipose tissue samples for single-cell analysis is insightful into the effects of bariatric surgery and the retention of obesogenic signatures even after significant weight loss.

a. Authors suggest that weight loss “diminished the strength and numbers of interactions between cell types, suggesting a possible return to a healthier state.”. It will be helpful to present such data for lean individuals in figure 1c, to see the relative resemblance to lean condition.

We thank the referee for the comment. The original CellChat analysis presented was based on snRNA-seq of pooled human AT samples. As inter-individual differences might be significant, particularly in human samples, we believe that a CellChat analysis should be done per individual. We have now been able to demultiplex our datasets and thereby generate person specific snRNA-seq data (albeit these cannot be linked to the clinical data due to a lack of SNP information for each participant). As expected, we find some inter-individual variability. Unfortunately, the number of cells (nuclei) in each cluster per person is too small to perform meaningful CellChat analysis. While we still believe that our global CellChat analysis overall reflects the phenotype, we have opted to remove human CellChat analyses from the manuscript and rather focused on the new additional snRNA-seq of the additional studies.

In contrast to human samples, the datasets generated from our mouse AT samples represent a more controlled setting (genetics, diet, age, environment, etc.) and we have therefore performed CellChat analyses in the mouse snRNA-seq data for macrophages, APCs, adipocytes and FIPs (**Reviewer Fig. 11**) and included this also in **Extended Data Fig. 4d**. We observed that epiAT of lean mice interaction strength was smaller than in epiAT of obese or formerly obese mice. We have included the following sentence in the revised manuscript and believe that this supports our initial findings yet represents a more controlled analysis.

Lines 194-196: *“In line with this, CellChat analysis indicated, that interaction strength between macrophages, adipocytes, APCs and FIPs was increased in epiAT of H, HC, HH and HHC mice compared to epiAT from lean mice (Extended Data Fig. 4d).”*

Reviewer Figure 11 (data also presented in Extended Data Fig. 4d): CellChat analyses of epiAT.

b. In this section, authors state that the change in cell abundances between obese and weight loss conditions might “also be affected by sampling effects.” It is unclear which sampling effects are of potential concern (batch, biopsy differences between lean/obese individuals, etc). This leads one to consider how significant sampling errors may be between visAT biopsies. An estimate of sampling variability may be useful for supplemental information, and additional clarity may be needed here.

We have now performed a more advanced analysis of all human AT samples, in particular we have been able to demultiplex the samples and as such assess cell composition for each sample. This shows that cell composition varies from donor to donor even amongst lean individuals (**Reviewer Fig. 12** and **Extended Data Fig. 2c,f,i,l**). It is worth noting that for humans only a small part of AT can be sampled as opposed to mice where we used the whole epiAT depot. As such sampling proximity to blood vessels or lymph nodes for instance could affect cell composition. Indeed, we identified one lean donor whose nuclei were comprised of 50% B-cells. After confirming with the responsible clinician, we concluded that this is likely caused by partial sampling of a lymph node in the omAT and excluded all nuclei mapped to this donor from the subsequent analysis (see **Methods** section).

We conclude that controlling the exact location of where omAT is sampled during bariatric surgery or other surgeries is challenging as it is sampled in addition to surgery. While surgeons in the various hospitals are asked to sample at the same location, we cannot absolutely control this and have therefore used our demultiplexing approach to identify and remove (in agreement with the clinicians) samples that have a strong sampling bias. For the NEFA study, participants were specifically recruited for scAT sampling prior to and post bariatric surgery (which is also easier) and therefore sampling effects are likely minimal. We have now provided more information in the **Methods** section of the revised manuscript on how biopsies were obtained.

Nonetheless, our overall observation is that the prevailing effects are donor dependent and not surgery type dependent, leading us to conclude that our overall differential expression results are robust.

Reviewer Figure 12 (data also presented in *Extended Data Fig. 2c,f,i,l*): **a**, Barplot depicting the relative cell type abundance in omAT of the MTSS cohort per condition and tissue donor. **b**, Barplot depicting the relative cell type abundance in omAT of the LTSS study per condition and tissue donor. **c**, Barplot depicting the relative cell type abundance in scAT of the LTSS study per condition and tissue donor. **d**, Barplot depicting the relative cell type abundance in scAT of the NEFA study per condition and tissue donor.

c. In figure 1f, it is unclear why Reactome 2022 pathway analysis was utilized for upregulated transcripts, whereas downregulated transcripts were assessed by WikiPathways 2019. To avoid results bias, both up and downregulated sets should be assessed by the same databases, both being fine, and additional results provided in supplemental material. The rationale for choosing these analytical tools should be provided

We agree with the referee that the same pathway databases should have been used and have now repeated (and expanded) the analyses using the WikiPathways database for adipocytes from omAT and scAT as it is a continually evolving community curated pathway database. In omAT, adipocytes persistently “downregulated terms” are metabolism (**Reviewer Fig. 13a,b**) and upregulated inflammatory, apoptotic, fibrotic signaling (**Reviewer Fig. 13c,d**) in both the MTSS and LTSS study biopsies. In adipocytes of scAT adipogenesis related pathways remained downregulated at T1 (**Reviewer Fig. 13e,f**) but for upregulated pathways we did not

obtain significant GSEA results for scAT of the LTSS study (Reviewer Fig. 13g). In adipocytes from scAT of the NEFA study inflammatory and TGFB signaling remained upregulated at T0 (Reviewer Fig. 13h).

Of note, Figure 1 has been completely revised and we decided to focus on the different studies, cell type mapping, and the evidence for transcriptional changes remaining after weight loss in the different datasets. We also now show a few persistently deregulated genes (Fig. 1h&j) rather than showing the results from the WikiPathways analysis as we believe this will be more useful for the reader.

Reviewer Figure 13: GSEA results using WikiPathways database for persistently downregulated DEGs (a,b) and upregulated DEGs (c,d) in adipocytes of omAT from the MTSS and LTSS study and GSEA results using WikiPathways database for persistently downregulated DEGs (e,f) and upregulated DEGs (g,h) in adipocytes of scAT from the NEFA study.

WikiPathways database for persistently downregulated DEGs (**e,f**) and upregulated DEGs (**g,h**) in adipocytes of scAT from the LTSS and NEFA study.

d. Data sets analyzed include samples from 32 females and 12 males. There is significant variability in gene expression of select genes correlated with BMI loss in figure 1i. Are their sex-specific differences? One point of additional clarity. This data indicates that 43 individuals were assessed, but in figure 1a n=22 total samples are analyzed. Was gene expression analyzed for the 43 individuals using a different approach? Methods and figure legend descriptions should be revised for clarity.

We thank the referee for raising this point. Of note, following the comments from the referees and upon re-evaluation of the conclusions from the original Kendall correlation analysis, we concluded that this type of analysis has several drawbacks limiting the interpretability (please see above), and thus we decided to remove the Kendall correlation analysis and associated statements from the manuscript. We believe that the now included additional datasets supporting our original findings related to retained transcriptional changes post weight loss provide much stronger evidence and support our original observation related to long-lasting transcriptional changes in the adipose tissue.

Nonetheless, we would like to address the question raised by the referee here and also clarify which samples were used for the original Kendall correlation analysis using bulk RNAseq and which are used for snRNA-seq.

snRNA-seq (original MTSS study): We selected 8 individuals of the MTSS study that met our selection criteria (BMI loss >25%, no medication, no cancer or other confounding diseases) and used their omAT obtained at T0 and T1 for snRNA-seq. Further, we used omAT from lean donors of the MTSS study. Therefore, we had 22 omAT samples (6x lean, 8x obese pre- and 8x post-surgery; please see below for details) in total for the snRNA-seq. After reanalysis and SNP demultiplexing and confirming with the responsible surgeon we removed nuclei assigned to one lean donor whose sample apparently was comprised of more than 50% B-cells (see answer to Question b above).

Bulk RNAseq: RNAseq was performed of omAT from 44 MTSS study participants (32 females and 12 males), whose omAT was sampled at T1 and T0 and lost 25% BMI from T0 to T1. The 8 individuals whose omAT we used for snRNA-seq are also included in this dataset. We then correlated gene expression of “memory” DEGs identified in adipocytes in omAT at T1 to clinical parameters. Lean samples were not included in this bulk RNAseq dataset.

Sex differences: We agree that investigating sex differences regarding gene expression, metabolic memory and weight loss is important. After demultiplexing by SNPs, we did not observe sex-specific compositional differences (**Reviewer Fig. 14**). This could be because we have fewer biopsies of male donors than of female donors. Further, after demultiplexing we found that each donor and thus both male and female donors retained a changed transcriptional profile from T0 to T1 in both the LTSS and the MTSS studies. In the NEFA study only samples from female donors were included (see below).

Sex distribution among tissue donors:

MTSS lean donors: 1 male, 4 female

MTSS obese/weight loss: 2 male, 6 female

LTSS lean donors: 2 male, 3 female

LTSS obese/weight loss: 2 male, 3 female

NEFA lean donors: 8 female

NEFA obese/weight loss: 7 female

Reviewer Figure 14: **a**, Barplot depicting the relative cell type abundance in omAT of the MTSS cohort per condition and tissue donor. **b**, Barplot depicting the relative cell type abundance in omAT of the LTSS study per condition and tissue donor. **c**, Barplot depicting the relative cell type abundance in scAT of the LTSS study per condition and tissue donor. **d**, Barplot depicting the relative cell type abundance in scAT of the NEFA study per condition and tissue donor. Sex is indicated per donor on top of bars.

e. Line 107, page 4. Authors suggest “individuals with weaker transcriptional memory of obesity experience greater weight loss (WL).” It is recommended that this topic be revisited in the discussion as potential mechanisms that some individuals may erase transcriptional memories more effectively (diet vs exercise vs longevity of initial obesity).

We agree with the referee and have removed both the statement, and the Kendal correlation analyses from the revised manuscript (as also outlined above).

2) As related to the experimental design and results in figure 6c, previously obese mice rebound more rapidly to a state of obesity, and the effect is greater for those mice previously fed high fat diets longest. Though perhaps impractical to repeat most assays, weight loss tracking may be reasonable to establish how long, if possible, reversal of obesogenic memory may take. That is, in both HC and HHC conditions weight normalization is achieved by as early as 4 weeks. Can additional time on a normal chow diet erase the weight regain tendency? What may be the effects of a low-calorie intervention at this point on both populations on such memories?

The referee raises a very interesting point here and it is tantalising to speculate that the obesogenic memory might be erased or diminished with just enough time on a low-caloric diet.

As already noted by the referee, these experiments might be difficult or impractical to perform, especially given the significant length of the experimental setup. We, therefore, after discussions with colleagues and the editor, decided to extend the weight-loss period to match the weight-gain period and performed a new weight loss study with 12 weeks HFD feeding, followed by an equal time (12 weeks) of chow diet feeding before the HFD rechallenge (**Reviewer Fig. 15**). We still observed an accelerated regain of weight in prior HFD exposed mice, even after 12 weeks of chow diet feeding. We cannot exclude that an even longer time of chow diet feeding could diminish this effect, but it would be important to then also consider potential ageing effects in the control and experimental groups, which could confound these experiments.

Reviewer Figure 15: Body weight of HHC_long (n=17) and CHC_long (n=14) mice that were put on chow diet for 12 weeks following 4 weeks of HFD refeeding.

We can only speculate on potential other effects that may influence the rebounding effect. We believe that experiments investigating behavioral or neuronal aspects could reveal additional mechanisms, but these are clearly beyond the scope of our current manuscript.

3) Line 334, page 10. Authors conclude that “4 hPTMs and ATAC-seq could predict or explain 31% of upregulated DEGs, which were related to inflammation, and 60% of downregulated DEGs...in the HCH group.” Here, and elsewhere when predicting histone and accessibility connections with differential expression, or in the discussion, authors should consider which other factors may explain the global transcriptional effects. That is, what other epigenetic factors may be of prime interest for future studies?

We agree with the referee that we are only looking at a selected / limited number of epigenetic modifications and other epigenetic modifications, including DNA methylation, other hPTMs,

etc. are also likely contributing to this epigenetic memory. We have now stated this in the **Discussion**.

Lines 414-416: “*It is possible that other epigenetic modifications that we did not analyse in this study, such as other hPTMs, DNA methylation, or non-coding RNAs, also contribute to the observed phenomena.*”

4) Line 416, page 12. Authors hypothesize that epigenome remodeling may reduce obesity associated rebound effects. Though locus-specific epigenetic editing may be impractical for this study currently, implementation of a global epigenetic modifier such as TSA or similar, may alleviate the rebound effect. A pilot study for weight tracking is reasonable, which would benefit further with epigenomic assessment of related histone modifications. As related, TSA was shown to reverse adipocyte size increases and weight gain by high fat diets in mice. PMID: 34232916.

We thank the referee for this comment. In line with the assessment of the referee, we believe that targeted epigenetic editing would be the ideal approach, yet currently impractical due to lack of suitable *in vivo* methodologies. Our epigenetic analysis suggests that such an approach would have to target 100s or 1000s of sites at the same time.

The alternative global approach has also certain limitations, and *in vivo* administration of “epigenetic drugs” such TSA or similar HDAC inhibitors might result in systemic effects beyond histone acetylation (e.g., acetylation of other proteins) in the adipose tissue but also in other organs. Indeed, the study by Lv *et al.* (PMID: 34232916) did find phenotypic differences but did not show these are related to changes in histone acetylation levels of mature adipocytes or of whole adipose tissue upon *in vivo* administration of TSA.

To first assess the effects of global HDAC inhibitors (TSA and Entinostat) on mature adipocytes and whether they robustly modulate histone acetylation levels, we decided to perform an *in vitro* experiment. This is also in line with the local animal welfare and 3R principles. We isolated mature adipocytes from epiAT, cultured them using the MAAC protocol for 1 week with 500 nM TSA (as in Lv *et al.*), 500 nM Entinostat, or DMSO and performed Western Blot for H3K27ac and H3. Neither TSA nor Entinostat treatment significantly increased H3K27ac levels in primary mature adipocytes (**Reviewer Fig. 16a,b**). This might be because these cells are not dividing and suggests the effects on adipocyte size observed by Lv *et al.* may be independent of H3K27ac changes in adipocytes.

To further assess whether histone lysine acetylation levels are globally changed in our experimental system, we performed WB analysis of primary adipocytes isolated from C, H, CC_s and HC epiAT. We did not observe a difference in global Kac or H3K27ac levels (**Reviewer Fig. 16c-f**). Similarly, histone acetylation of nuclei isolated from human omAT from T0 or T1 used for snRNA-seq was not changed compared to histone acetylation levels in nuclei from lean omAT (**Reviewer Fig. 16g,h**).

Overall, having done these experiments, we believe that untargeted and global modulation of epigenetic modifications is not a therapeutic strategy to alleviate or reduce the obesity associated rebound effects. In addition, we are of the opinion that administration of HDACi *in vivo* during the weight loss period could confound the experiment via indirect effects of TSA or similar drugs in other tissues or on other pathways and as such would not allow to test whether erasure of an epigenetic memory in adipocytes specifically has beneficial effects.

Reviewer Figure 16: **a**, Western Blot of H3K27ac and H3 of isolated adipocytes from *epiAT* treated with DMSO, TSA (500 nm) or Entinostat (500 nm) for 1 week. **b**, Normalization of H3K27ac over H3 from **a**. Lines indicate paired samples (adipocytes from same pool of mice). Paired Wilcoxon tests were used to test for significance. **c**, Western Blot of Kac and H4 and H3K27ac and H3 (**d**) from primary adipocytes isolated from *epiAT* of C, H, CC_s and HC mice. **e**, Normalization of Kac over H4 for **c**. **f**, Normalization of H3K27ac over H3 for **d**. **g**, Western Blot of H3K27ac and H3 of nuclei isolated from whole *oMAT* tissue pools that were used for snRNA-seq from the MTSS and LTSS studies. **h**, H3K27ac over H3 from **h**.

5) As related to point 3 and 4, authors do not describe data on epigenetic modifiers which may be at play in defining epigenetic obesity memories. Expression data should be presented on primary epigenetic factors (e.g. HMTs, HDACs, DNMTs, TETs) and networks, which are likely altered between obese, lean, and weight loss individuals and mice, and available in their current datasets.

We thank the referee for this comment and have now included the data as requested. In mouse adipocytes no epigenetic modifier that is detected in snRNA-seq remained or was differentially expressed after weight loss. However, during obesity some are deregulated such as *Hdac9* which has been reported before. We have plotted these below (**Reviewer Fig. 17**) and included it in the manuscript in the new **Extended Data Fig. 6f**. We conclude that the retained epigenetic memory was likely not caused by increased or decreased abundance of a specific epigenetic modifier.

Lines 287 – 289: “Notably, the expression of relevant epigenetic modifiers was not deregulated in HC or HHC adipocytes (Extended Data Fig. 6f).”

In human AT, we did not detect consistent deregulation of epigenetic modifiers in adipocytes during obesity or after weight loss.

Reviewer Figure 17 (data also presented in Extended Data Fig. 6f): Violin plots of genes encoding for epigenetic modifiers deregulated either in H (*) or HH (#) adipocytes. None of the epigenetic modifiers are deregulated in HC or HHC adipocytes.

6) In figure 1c, human scRNA-seq data is assessed for interaction strengths between cell types indicating weight loss to diminish the strength of interactions between cell types. Such logic and assessments will also be usefully applied to murine data before and after weight changes and examined for consistency with human data.

This is a good suggestion. As written above (see answer to comment 1), we have decided to remove the CellChat analyses of the human snRNA-seq data from the revised manuscript, but we have now performed CellChat analyses for macrophages, APCs, adipocytes, and FIPs for mouse snRNA-seq data (**Reviewer Fig. 18**) and included this in new **Extended Data Fig. 4d**. We observed that in epiAT of lean mice interaction strength was smaller than in epiAT of obese or formerly obese mice. We have included the following sentence in the revised manuscript.

Lines 194-196 “In line with this, CellChat analysis indicated, that interaction strength between macrophages, adipocytes, APCs and FIPs was increased in epiAT of H, HC, HH and HHC mice compared to epiAT from lean mice (Extended Data Fig. 4d).”

Reviewer Figure 18 (data also presented in Extended Data Fig. 4d): *CellChat analyses of epiAT.*

Other minor concerns

a. Scale bars need to be clearer in most histology images.

Thank you for noting this. We have increased the thickness of all scale bars to improve visibility.

b. Line 198, page 6 regarding adipocyte biology. Authors should provide a standard citation.

Thank you for this recommendation. We have added a standard citation (ref. 48; PMID: 24439368).

c. The authors may want to elaborate more on the decrease in proportion of adipocytes presented in figure 6j. How does this compare to adipocyte concentration overall in fat tissue of CCH and HCH mice?

The analysis presented in Fig 6j is a relative analysis of the cell numbers present/identified and mapped to specific cell types in the 4 different snRNA-seq samples. The snRNA-seq data does not allow us to assess or quantify absolute adipocyte numbers of the total respective tissue in each sample. We are not aware of a method to properly calculate adipocyte concentration based on our data or a method to quantify adipocytes in a tissue piece based on our available measurements.

Overall, based on our current data, we cannot make any claims related to the changes of absolute adipocyte number in the tissue, and whether the relative changes observed also translate in changes in “space” occupied by adipocytes. We nonetheless believe that during high-fat diet feeding, individual adipocyte size (and volume) will increase, meaning that the overall adipocyte volume of the AT tissue will not diminish and be replaced by other cell types. But we decided to not include any statement related to this in the revised manuscript, as we cannot confirm them with our current data.

d. The authors use the word “imprinting” in the abstract (line 22, page 3). They may wish to consider using alternative terminology to avoid confusion with DNA methylation imprinting during early germ cell and embryo development.

Thank you for this recommendation. We agree that this could result in confusion and have rephrased this part of the abstract.

Lines 24-26: “*However, maintaining weight loss is a considerable challenge, especially as the body is believed to retain an obesogenic memory that defends against body weight changes.*”

e. The authors could cite prior studies that have demonstrated the role of epigenetic changes (like DNA methylation) in metabolic memory of complications in subjects with diabetes.

Thank you for this recommendation. We mainly included publications that focus on weight loss in our manuscript but have now added references to the legacy effect/metabolic memory in diabetes (Refs 14-16: PMID:25481708, PMID:18843126, PMID:18784090) and a reference highlighting persisting DNA methylation changes after restored glycaemic control in type 1 diabetic humans (Ref 39: PMID:27162351):

Lines 53-55 and 66-68: “*This recurrent pattern may be partially attributable to an (obesogenic) metabolic memory, persisting even after notable WL^{9,10} or metabolic improvements¹⁴⁻¹⁶. (...) Hitherto, human-focused studies have mainly relied on DNA methylation analysis in bulk tissue samples or whole blood to describe putative cellular memory³⁵⁻⁴¹.*”

Referee #4 (adipose/scRNA-seq):

In this manuscript, titled “Adipose tissue retains an epigenetic memory of obesity that persists after weight loss,” Hinte and colleagues explore the possibility that certain cell types in adipose tissue are epigenetically reprogrammed in obese individuals, and that after weight loss there is retention of some of these epigenetic marks which preserve transcriptional signatures of obesity and might make it more challenging to maintain weight loss. The implications of this hypothesis have significant impact on human health and obesity management, and as the authors discuss, other studies have also explored epigenetic memory conferred by obesity. This study includes single-cell measurements of both mouse and human adipose tissue from healthy individuals as well as samples from adipose tissue of obese individuals before and after weight loss. Single-cell measurements allow for cell-type-specific understanding of molecular and cellular perturbations caused by obesity, which makes this study distinct.

Overall, the potential impact of this study is very significant and would be of great interest to a broad scientific community given the unique human cohort study. However, there are some critical issues that would first need to be addressed. The underlying design of the experiments in this study compares adipose tissue cells from healthy individuals (mouse and human) to adipose tissue cells from a different set of individuals before and after weight loss. Epigenetic reprogramming is therefore claimed to be observed if there are epigenetic or transcriptional signatures that are more similar between the samples taken before and after weight loss in obese individuals, than between the healthy individuals and the weight loss samples. The mouse studies in this manuscript are, for the most part, rigorous and well controlled, presenting a comprehensive picture of the mechanism of obesity induced epigenetic reprogramming, as well as a thorough understanding of the epigenetic signals that are preserved after weight loss and the nature of the transcriptional memory that results from these preserved marks. The human data comes from a very valuable set of experiments in which adipose tissue from a cohort of individuals was sampled before bariatric surgery, and two years after the surgery, enabling comparison of obese tissue and post-weight-loss tissue to lean/healthy adipose tissue. The conclusions drawn from the human study, however, are not well supported by the observations, as these experiments are less well controlled. Specifically, the conclusion that there is an epigenetic memory conferred by obesity in human adipose tissue that is retained after weight loss is not supported by the data and should not contribute to the conclusions proposed in this manuscript, as currently presented. In fact, the human studies make up only a very small portion of this manuscript. Before publication, this section should undergo major revisions or be removed completely. Additionally, the comments and questions below should be addressed.

We thank the referee for the comments and the critical evaluation of our work. As recognised and stated by the referee, the mouse studies were performed to gain a comprehensive picture of obesity induced epigenetic reprogramming and how this contributes to transcriptional memory in the adipose tissue after weight loss. Here we had the opportunity to use a well-controlled model of diet-induced weight loss, mitigating potential confounding effects and we believe that we present overall convincing data, demonstrating that (mouse) “adipose tissue retains an epigenetic memory of obesity that persists after weight loss”.

Despite the fact that mice are widely being used as the primary model system to study human (patho)physiology, human studies and datasets are of key importance to draw human-relevant conclusions. We agree with the referee that “the conclusion that there is an epigenetic memory conferred by obesity in human adipose tissue that is retained after weight loss” was ultimately not based on the data we collected in humans but was an extrapolation from our mouse study. We have now addressed this criticism in the revised manuscript and completely revised **Fig. 1** and the associated text (lines 86-139).

Firstly, we agree that we cannot state that there is epigenetic memory in human adipocytes without providing epigenetic data from human donors. Due to technical limitations to isolate sufficient pure human adipocytes from the relevant samples and studies, and the limitations of single-nucleus histone modification profiling, we are not able to generate these datasets at present. We therefore do not state in the results that we observe epigenetic memory in humans. Secondly, we have expanded our transcriptomic work and included additional adipose tissue samples, including both omental adipose tissue and subcutaneous adipose tissue from additional bariatric surgery studies as well as more lean controls (see revised **Fig. 1**). We were even able to demultiplex the snRNA data and assess AT cellular composition and variability per donor based on SNPs found in the data (see answer to question 3 below).

Overall, we believe that the expanded snRNA-seq analysis of the human adipose tissue samples do support and substantiate our original observations that obesity induces transcriptional (and cellular) changes in obese and post-weight-loss adipose tissue. This finding is important and together with the mouse data provides the basis to speculate and discuss whether epigenetic memory might also contribute to human (patho)physiology.

1. In the human studies, transcriptional signatures are measured with single-cell RNAseq, and cell-type specific differential gene expression tests are performed between the healthy/lean individuals and the obese individuals before surgery. Those differentially expressed genes (DEGs) are then compared to DEGs detected between the healthy samples and the obese samples after surgery. The authors observed that many of the DEGs that were downregulated between the obese and lean samples, remained downregulated between the weight loss and lean samples, and concluded that this is evidence of a “transcriptional memory” of obesity in the weight loss samples. However, in these experiments the observed effect is confounded with other sources of variation between the individuals including genotype and behavior. Therefore, these experiments do not control for other non-epigenetic mechanisms that might cause the observed signal, such as the possibility that the genetic background of the obese individuals determines the DEGs, or that the dietary behavior of the obese individuals produces differential gene expression. Indeed, monogenetic obesity was not formally excluded in this study and the individuals who participated in this study did not adhere to any specific diet before or after surgery. Therefore, it can not be concluded that differential gene expression that is not restored after weight loss results from a “transcriptional memory” of obesity as opposed to being the result of a differential phenotype of the individuals studied.

We thank the referee for the comment and agree that a retention of a transcriptional signature (formerly termed “transcriptional memory” in our manuscript) can be influenced or potentially caused by an individual’s initial response to an obesogenic environment, nutritional and physical activity status, unreported medication use, behaviour, or even genetic predispositions. While we cannot fully control all these variables in a human study, we have attempted to limit or report potential confounders.

We cannot fully account for genetic variations in our human data (data is not available), but we can (to our best knowledge) exclude that we used samples of individuals suffering from monogenetic obesity: the clinicians conducting the studies have confirmed that none of the individuals were obese during childhood or experienced early onset obesity, an indicator of monogenetic obesity. We now also report on the available clinical data (new **Extended Data Table 1**).

We have performed SNP based demultiplexing of all our pooled human snRNA-seq datasets and have observed that matching transcriptional changes pre- and post-weight loss in omAT and scAT across are present in every individual in each of the three different studies. While

these may also be influenced by the above-mentioned factors, the results strongly support our observation of retained transcriptional signatures pre- and post-weight loss.

We have performed differential gene expression analysis per cell type and donor and subsequently analysed whether DEGs are also DEGs after weight loss or not. In all cohorts, adipocytes were the most “affected” cell type in terms of number of DEGs (new **Extended Data Fig. 1g-j** and **Reviewer Figure 19**). For each donor in each study, a set of DEGs was also deregulated after weight loss in adipocytes. In all studies, the adipocyte transcriptomes had DEGs that were also deregulated after weight loss (new **Extended Data Fig. 1k** and **Reviewer Figure 20**).

Reviewer Figure 19 (data also presented in Extended Data Figure 1g-j): Number of DEGs (up or down) per cell type per donor at T0 scaled by column. From left to right. MTSS study omAT, LTSS study omAT, LTSS study scAT, NEFA study scAT.

Reviewer Figure 20 (data also presented in Extended Data Figure 1k): Heatmap of number of persistently deregulated genes from T0 to T1 per cell type across AT pools.

2. Continuing from point 1, the distinction between “epigenetic memory,” “transcriptional memory,” “obesogenic memory,” and “metabolic memory” needs to be made more clear in this manuscript. The authors seem to use these terms interchangeably, but it can be confusing or even misleading at times. The concept of epigenetic memory is well defined in the literature and refers to a broad range of epigenetic features, including DNA methylation, histone modification, and chromatin organization, that be passed to progeny cells conferring distinct transcriptional phenotypes. However, it is not clear what the authors mean by transcriptional memory in the human study. The term “memory” implies an epigenetic mechanism, but the observation of a persistent transcriptional phenotype before and after weight loss does not necessarily correspond to an epigenetic memory. For example, a lifetime of obesity may create a physiological state that even after bariatric surgery, continues to generate metabolic signals that produce gene expression patterns that are distinct from healthy/lean adipose tissue. While potentially interesting, this phenomenon is distinct from epigenetic memory which is stated in the title of this manuscript and throughout the abstract, introduction and discussion. When specifically addressing the human data, the authors use the term “transcriptional memory” and my concern is that this term is used to evoke the notion of epigenetic memory but simply refers to similar gene expression patterns before and after weight loss.

We thank the referee for highlighting this potential confusing or misleading terminology. We acknowledge that the terms “transcriptional memory”, “epigenetic memory” or “metabolic memory” are used in different contexts, and their individual meaning might be seen as overlapping. For the understanding and interpretation of our data, it is important to better clarify these terms and we agree that we should aim to refrain from using wording that could be misunderstood. We had originally defined “transcriptional memory” as a transcriptional deregulation that is maintained after return to a lean(er) state (in mice and humans) compared to controls. Indeed, retention of a transcriptional signature does not require to be primarily caused by an underlying epigenetic signature and could also be caused by other factors, such as structural or physiological changes.

As pointed out by the referee above, our mouse work does show that epigenetic changes of obesity in the adipose tissue persist after weight loss. We therefore also state in the manuscript that we observe epigenetic memory in mice. However, we agree that our original wording and the interpretation of “transcriptional memory” in human AT might be misleading and not reflect the presented data. We have therefore changed the wording throughout the revised manuscript and refer to “retained transcriptional differences” or similar and not anymore to “memory” when presenting human data to avoid indicating that our observations are direct evidence of an epigenetic memory in human.

3. In addition to the fact that the experimental design of the human studies does not control for confounding sources of variation, there is little statistical analysis to support the significance of the observations that there is a set of DEGs that are preserved in obese tissue after weight loss. First, how many individuals were sampled between lean and obese to generate this set of DEGs?

We are aware and agree with the referee that controlling for confounding sources of variation such as diet, physical activity, but also genetics, is challenging in human studies, particularly in long-term clinical studies such as those included in our work. We do, however, think that we have performed appropriate statistical analyses to support our claim that in humans AT DEGs under obese conditions match those after weight loss. We address the referee’s question regarding sample number and statistical testing below:

The new **Extended Data Table 1** includes a table summarizing the information on the individuals whose scAT or omAT was used for the generation of snRNA-seq datasets. Also, the figure legends of **Fig. 1** and new **Extended Data Fig. 1** summarise this information. Of note, these numbers are now higher than in the original submission, since we included additional studies.

MTSS study: Total of 21 omAT biopsies

5 (1 male, 4 female) lean donors

8 (2 male, 6 female) obese/weight loss donors (paired samples collected at T1 and T0)

LTSS study: Total of 15 omAT and 15 scAT biopsies

5 (2 male, 3 female) lean donors

5 (2 male, 3 female) obese/weight loss donors (paired samples collected at T1 and T0)

NEFA study: Total of 22 scAT biopsies

8 (all female) lean donors

7 (all female) obese/weight loss donors (paired samples between T1 and T0):

Further, we would like to clarify for the referee how significant DEGs were identified. This description is also included in the **Methods** section of the manuscript. Importantly, in our single-nucleus differential gene expression analysis each cell/nucleus is treated as a replicate.

Lines (817-820): “Differential expression analysis (Wilcoxon rank-sum test with $|\log_2 \text{fold-change}| > 0.5$ and adjusted $p\text{-value} < 0.01$) per cell type between different conditions was done using the FindMarkers function from Seurat.”

Thereafter, we intersected DEGs found at T0 and T1 per cell type to obtain information on how many DEGs remained DEGs with the same regulation (up or down). This is also explained in the **Methods** section of the manuscript.

Lines (838-847): “Transcriptional Retention: DEGs from obese and WL cells from mouse and human were overlaid respectively. A DEG was considered as restored if it was no longer deregulated in WL cells when compared to controls. If not restored, we considered a DEG part of a transcriptional memory. Clusters identified as similar cell types (e.g., three clusters of endothelial cells) were merged for DEG quantification but not DE itself. For human snRNAseq only cell types for which we obtained at least 30 cells per donor were considered for the retention analysis. T-cells were not included in DE or transcriptional retention analysis. For integrated human adipocyte DE quantification non-coding transcripts were excluded.”

In the revised manuscript, we now also performed depot specific analyses of the adipocytes after integrating the adipocytes from MTSS/LTSS and LTSS/NEFA snRNA-seq datasets respectively, thereby increasing the n-number in each dataset. Thereafter, we performed differential analyses again and detected the DEGs that were retained after weight loss (**Fig. 1g-j**, and **Reviewer Fig. 21**). We are convinced that our statistical tests are correct and in agreement with the methods used, allowing us to identify valid DEG sets.

Reviewer Figure 21 (data also presented in Figure 1g-j): **g**, Proportion of downregulated (left) and upregulated genes in adipocytes from integrated omAT of LTSS and MTSS studies that retain an obesity induced transcriptional profile from T0 or change their trajectory toward the profile of lean adipocytes. **h**, Violin plots of selected downregulated memory DEGs in omAT adipocytes across all conditions that did not restore expression profile. **i**, Proportion of downregulated (left) and upregulated genes in adipocytes from integrated scAT LTSS and NEFA studies that retain an obesity induced transcriptional profile from T0 or change their trajectory toward the profile of lean adipocytes. **j**, Violin plots of selected downregulated memory DEGs in scAT adipocytes across all conditions that did not restore expression profile.

Apologies if I missed this, but while samples were collected from 44 obese individuals who experience at least 25% weight loss after surgery, I could not find how many lean individuals were sampled.

We thank the referee for pointing this out. We indeed did not mention the inclusion criteria of the control lean samples in the original manuscript and apologize for this mistake. For each study, we have now supplied more information on how samples were collected in the **Methods** section (see **Methods** and the introductory statement of this rebuttal letter). Samples of lean donors were either obtained during surgeries such as hernia surgery, explorative laparoscopies, or cholecystectomies (MTSS and LTSS studies) or individuals were recruited for scAT sampling specifically (NEFA study).

The authors mention that differences in biopsy sampling could have contributed to the observed variation in cell type abundance. Could sampling effects also have influenced the DEG results?

We thank the referee for raising this question. A similar comment was made by referee #3 (comment 1b) and we are providing here an answer which is in parts also provided above.

To address these questions, we have now performed a more advanced analysis of all human AT samples, in particular we have been able to demultiplex the samples and as such assess cell composition for each sample. We observe that cell composition varies from donor to donor even amongst lean subjects (**Reviewer Fig. 22** (and **12**) and new **Extended Data Fig. 2c,f,i,l**). It is worth noting that for humans only a small part of AT can be sampled as opposed to mice where we used the whole epiAT depot. As such sampling proximity to blood vessels or lymph nodes for instance could affect cell composition. Indeed, we identified one lean donor whose nuclei were comprised of 50% B-cells. After confirming with the responsible clinician, we concluded that this is likely caused by partial sampling of a lymph node in the omAT and excluded all nuclei mapped to this donor from the subsequent analysis (see **Methods** section).

Controlling the exact location of where omAT is sampled during bariatric surgery or other surgeries is challenging since it is sampled in addition to surgery. While surgeons in the various hospitals are asked to sample at the same location, we cannot absolutely control this and have therefore used our demultiplexing approach to identify and remove (in agreement with the clinicians) samples which have a strong sampling bias. For the NEFA study, participants were specifically recruited for scAT sampling prior to surgery as well two years post-surgery (which is also easier) and therefore sampling effects are minimal. We have now provided more information in the **Methods** section of the revised manuscript on how biopsies were obtained.

Finally, we cannot exclude that sampling effects such as exact anatomical location could affect the DEG analyses. However, our overall observation is that the prevailing effects are donor dependent and not surgery type dependent, leading us to conclude that our overall differential expression results are robust and especially the claim that a substantial number of donor specific DEGs present in obesity are also found after weight loss in the same patient (see also Reviewer Figure 19 and Reviewer Figure 20 in reply to your question 1).

Reviewer Figure 22 (data also presented in Extended Data Fig. 2c,f,i,l): **a**, Barplot depicting the relative cell type abundance in omAT of the MTSS cohort per condition and tissue donor. **b**, Barplot depicting the relative cell type abundance in omAT of the LTSS study per condition and tissue donor. **c**, Barplot depicting the relative cell type abundance in scAT of the LTSS study per condition and tissue donor. **d**, Barplot depicting the relative cell type abundance in scAT of the NEFA study per condition and tissue donor.

Could sampling effects impact the cellchat interaction results? Is there any statistical test to determine if the results of cellchat analysis are meaningful/statistically significant?

Like the answer above regarding our human DEG results, sampling effects can affect cellular composition and could influence the CellChat interaction results as well. For our revised manuscript, we have completely revisited the CellChat analyses (see also answer to referee #3, comment 1a). The original CellChat analysis presented was based on snRNA-seq of pooled human AT samples. As inter-individual differences might be significant, particularly in human samples, we believe that a CellChat analysis should be done per individual. We have now been able to demultiplex our datasets and thereby generate person specific snRNA-seq data (albeit these cannot be linked to the clinical data due to a lack of SNP information for each participant). As expected, we find some inter-individual variability. Unfortunately, the number of cells (nuclei) per person is too small to perform meaningful CellChat analysis. Based on this and the

fact that sampling effects could further skew our analyses, we have opted to remove human CellChat analyses from the manuscript and rather focused on the new additional snRNA-seq of the additional studies.

Of note, we now include CellChat analyses of the mouse snRNA-seq data (see **Extended Data Fig. 4d**), where we can control sampling effects (we collect the whole tissue) and many other potential confounders. Unfortunately, we are not aware of a way to perform statistical testing on CellChat interaction results in terms of numbers of interactions to verify if the results are statistically significant. While we could perform more in-depth statistical tests for specific pathways, this is in our opinion beyond the scope of this manuscript.

Fig 1 g,h,&l shows the relationship between upregulated DEGs and BMI and the authors state a significant negative correlation, but there is no measurement of correlation or significance. This data is used to imply that “individuals with weaker transcriptional memory of obesity experience greater weight loss.” But I am not convinced by this implication given the plots in Fig 1 g,h,&l, which qualitatively present little to no correlation.

We thank the referee for their comment. Following the comments from the referees and upon re-evaluation of the conclusions from the original Kendall correlation analysis, we concluded that this type of analysis has several drawbacks limiting the interpretability (please see above), and thus we decided to remove the Kendall correlation analysis and associated statements from the manuscript, including the statement “individuals with weaker transcriptional memory of obesity experience greater weight loss”. We believe that the now included additional sn-RNA-seq datasets supporting our original findings related to retained transcriptional changes post weight loss provide much stronger evidence and support our original observation related to transcriptional changes pre and post weight loss in the adipose tissue.

Nonetheless, we would like to address the question raised by the referee here and to clarify our previous analyses. In **Fig. 1h** of the original manuscript (now removed) we showed Kendall correlation coefficients, which were shown as the dot colour, ranging from 1 to -1. Kendall correlation analysis is based on the ranked data when the sample size is small and does not assume linear relationship between the two variables. The stars on the coloured dots indicated whether the gene expression correlation with the clinical parameter was statistically significant or not. The p-value ranges were indicated in the figure legend. The size of the dot corresponded to the colour scale. Therefore, the individuals DEG expression was, indeed significantly correlated to BMI loss, but not linearly, which is one major drawback limiting the interpretation of Kendall correlations. We agree with the referee, that former Figure 1i did not support the indication that there was a linear relationship between the “transcriptional memory” and BMI loss and could result in confusion or misinterpretation of our Kendall correlation analyses results.

4. In the section “Mouse adipocytes retain an obesogenic epigenetic memory after WL” the authors perform multi-omic epigenetic analysis of adipocytes after weight gain and weight loss to identify persistent epigenetic signatures. In order to confirm that the appropriate populations are compared, they use a labeling strategy to track adipocytes in mouse epiAT through a high fat diet. Extended data figure 5c is used to confirm that the cells that are epigenetically profiled represent the same population of cells profiled with single-nuclei RNAseq. The authors then use Fig 4C to demonstrate a “restoration of the translational profile in adipocytes from HC and HHC mice.” These two measurements seem to be important observations but I am having trouble understanding the analysis. In ED5c, while the spearman

coefficients on log2 counts are above 0.7 for all the samples compared, it looks like the genes skew towards the bulk samples, such that a majority of the genes are low counts in the single nuclei samples with a wide range of expression levels in the bulk samples.

More importantly, what is the justification for comparing TRAP-seq counts to single nuclei RNAseq counts?

We agree that comparing TRAP-seq and snRNA-seq datasets has some limitations. While TRAP-seq will (only) measure translating RNA, snRNA-seq does often only detect high(ly) expressed transcripts and often still unspliced RNA from the nuclei. Furthermore, the two datasets represent either bulk (labelled) adipocytes or pseudobulk data from cells classified as adipocytes. The purpose of our analysis was to check whether our labelled adipocytes would be largely different in their transcriptional profiles from the corresponding snRNA-seq. We can conclude that this is not the case, and we can therefore assume that our epigenetic data can be used in conjunction with the transcriptional datasets generated. Of note, TRAP-seq was the only way to collect adipocyte specific (and matched) RNA from the same samples that we used for our epigenetic characterisation.

Because of the technical differences between the two methodologies, many genes that are detected in the TRAP-seq are not (or only at very low levels) detected in snRNA-seq (lower coverage of the snRNA-seq dataset), and conversely many non-coding RNAs detected in snRNA-seq are not detected in TRAP-seq. Furthermore, gene expression is averaged across all cells mapped to adipocytes in the snRNA-seq data, whereas it is averaged across 2-3 biological replicates in the TRAP-seq datasets. All these limitations and technical differences are skewing the correlation analysis and we believe that the correlation of > 0.7 is high, indicating that snRNA-seq data and TRAP-seq are comparable.

In fig 4C how is the conclusion stated above made? The replicates for each condition seem to spread across the PC1 and PC2 space and it is not qualitatively clear that the spatial concordance between any set of replicates indicates an especially similar transcriptome. The analysis and conclusions in lines 213-217 should be clarified.

The samples of lean and former obese adipocytes (HC and HHC) are indeed – to some degree – scattered across PC1. We did attribute this to variability between individual mice rather than between weight loss and controls. PC2 is nonetheless separating H and HH adipocytes and HC and HHC are closer to their age matched controls in PC2 (please see below for a quantification).

Following the referee's suggestion in the next paragraph, we have calculated the distances in the PCA space to support and “quantify” our statement that HC and HHC adipocytes normalize their transcriptome towards that of age matched controls based on distances of centres of mass.

Centres of Mass for Figure 4C / TRAP-seq data:

Condition	PC1	PC2
C	-42.336036	59.6538952
H	-21.502737	-62.831844
CC_s	61.5314867	35.3417907
HC	-0.0209366	7.41410136
CC_1	18.3110116	54.3249019
HH	-39.580913	-97.440692

CCC	-31.135884	0.74553614
HHC	55.8981201	-16.996662

Distances between centres of mass in PC2 space:

- C to H: 122.49
- CC_s to HC: 27.93
- CC_1 to HH: 151.77
- CCC to HHC: 17.74

In line with our prior qualitative statement, the calculated distances of centres of mass in PC2 for the WL samples are indeed closer to their age matched controls (CC_s to HC and CCC to HHC) than obese adipocytes are to their age matched controls (C to H and CC_1 to HH).

Geometric distances between condition means in the PC1-PC2 space:

	C	CC_1	CC_s	CCC	H	HC	HH	HHC
C	0	159.92651	140.12384	124.647748	149.616494	123.753854	176.794235	173.052263
CC_1	159.92651	0	161.409409	166.103961	172.172338	125.354204	189.433844	178.702077
CC_s	140.12384	161.409409	0	148.642369	148.292231	115.78149	190.05284	137.996436
CCC	124.647748	166.103961	148.642369	0	138.034257	125.383483	145.807554	162.759845
H	149.616494	172.172338	148.292231	138.034257	0	108.729004	100.998587	158.074588
HC	123.753854	125.354204	115.78149	125.383483	108.729004	0	138.374384	133.745254
HH	176.794235	189.433844	190.05284	145.807554	100.998587	138.374384	0	173.011407
HHC	173.052263	178.702077	137.996436	162.759845	158.074588	133.745254	173.011407	0

In the PC1-PC2 space, HC adipocytes samples are closer to age-matched CC_s than H adipocytes are to C adipocytes (115.78 vs 149.62). For CCC adipocytes this is less pronounced, but the distance is still smaller (HHC/CCC vs CC_1/HH: 162.76 vs 189.43).

5. Continuing from point 4, the authors then compare the epigenetic profiles across replicates of each feeding condition. They state that they use multi-omic factor analysis to “overcome limitations of modality-specific analyses” and refer to extended data figure 5d. This figure seems to show broad variation in the positioning of replicates across the first two principal components of cut&tag and atacseq data. However, they then use MOFA to conclude that factor 1 explains the major source of variability between conditions which is influenced predominantly by histone modifications. So why then do cut&tag measurements of histone modifications reveal this condition specific variability? I do not understand how MOFA overcomes the modality-specific analysis.

The tool MOFA has been developed to specifically assess multiomic datasets and identify factors of relevance to explain the variability in the data. It presents advances over single-modality-based analyses. We have used MOFA to identify biological source(s) of variability in our data sets based on all modalities across all conditions in an unsupervised manner. It acts like a statistical generalization of PCA to infer joint low-dimensional representation of multi-omics datasets in terms of latent factors representing the underlying principal axes of heterogeneity across samples. Therefore, it can highlight which modalities are driving the latent

factors capturing most of the observed data variance and moreover provide a factor space to see if (at all) samples cluster according to condition or rather technical effects. Factor 1 in our case was indeed driven by hPTMs but most predominantly by H3K4me1 which is why we later compiled adipocyte specific enhancer sets and performed differential analysis based on them instead of focusing only on promoters.

Further, we agree that referring to Extended Fig 5d (now **Extended Data Fig. 6d**) when explaining the advantages of MOFA is incorrect and have changed this in the manuscript. Now we refer to **Extended Data Fig. 6d** in lines 260-262: “*Motivated by our MOFA findings, we investigated promoters marked by H3K4me3 or H3K27me3 to identify differentially marked promoters for these hPTMs (Extended Data Fig. 6d).*”

Furthermore, much of this principal component analysis is qualitative, identifying regional clustering of replicates. The authors should perform quantitative analysis of the geometric distances between condition means (centers of mass) instead of relying on qualitative interpretations of clustering in PCA space. Because this section provides some of the most compelling observations of epigenetic memory, it is very important that the analysis is clearly motivated and interpreted, and implements rigorous statistical methods.

We thank the referee for their suggestion and have now performed quantitative analyses of the geometric distances between the centres of mass of our conditions. This analysis was valuable to confirm and support our interpretation of an epigenetic memory. When we plot MOFA factors against each other we do not perform PCA and therefore the positions in the latent space are calculated slightly differently. We calculated the Euclidean distance after calculating the centres of mass in the MOFA factor 1 and 2 latent space and compared centres of mass for factor 1.

Geometric distances between condition means in the MOFA factor 1 and 2 latent space:

	C	CC_1	CC_s	CCC	H	HC	HH	HHC
C	0	1.78635432	1.43067598	1.71255309	3.54105514	2.78769306	3.56157891	3.51493444
CC_1	1.78635432	0	1.73732243	1.36976995	2.16829207	1.29998563	2.01939229	2.20499997
CC_s	1.43067598	1.73732243	0	1.38798396	3.32480979	2.65974299	3.46132323	3.64000095
CCC	1.71255309	1.36976995	1.38798396	0	2.87788301	1.74709516	2.71734965	2.49278705
H	3.54105514	2.16829207	3.32480979	2.87788301	0	1.58344965	0.9075197	2.57604816
HC	2.78769306	1.29998563	2.65974299	1.74709516	1.58344965	0	1.14040788	1.29913717
HH	3.56157891	2.01939229	3.46132323	2.71734965	0.9075197	1.14040788	0	1.79060002
HHC	3.51493444	2.20499997	3.64000095	2.49278705	2.57604816	1.29913717	1.79060002	0

Based on this quantification we conclude that indeed HC and HHC samples cluster closer to H and HH samples than to age matched controls (HC: 1.58 vs 2.66 and HHC: 1.79 vs 2.49) and both obese (H to C: 3.54 and HH to CC_s: 2.02) and WL adipocyte samples clustered far away from controls in the MOFA factor 1 and 2 latent space.

Centres of mass in MOFA Factor 1:

Condition	Factor1
C	-1.580118
CC_1	-0.8617194
CC_s	-1.577737
CCC	-0.8936732
H	0.81727376
HC	0.79937423
HH	1.32104444
HHC	1.66381668

Reviewer Figure 23 (data also presented in Figure 4d): Positioning of samples along MOFA Factor 1.

MOFA Factor 1 separates control adipocyte samples from obese and WL adipocytes based on these calculations and visualization along Factor1 (**Fig. 4d** and **Reviewer Fig. 23**). Further, the differential analysis of different hPTMs is statistically valid and is also a strong indicator of an epigenetic memory.

6. Same points as above for figures 5 b&c and ED6 d&e. The authors observe that certain post-weight loss feeding conditions clustered more closely to pre-weight loss conditions than to controls, indicating retention of epigenetic signals. These distances must be quantified in order to support this conclusion. It is not even qualitatively clear that the HC and HHC conditions are more epigenetically similar to the H and HH conditions than to the controls, based on the PC analysis.

We have now calculated the centres of mass and geometric distances between them for PCAs shown in **Fig. 5b,c** and **Extended Data Fig. 7d,e** (this was the former Extended Data Fig. 6d,e) as requested. Indeed, the PCA in **Fig. 5b** (H3K4me1 in enhancers) shows that HC and HHC adipocytes samples are not closer to CC_s and CCC than obese adipocytes. But as we had stated in the manuscript, the enhancer signature of obese and WL adipocytes samples clustered closer to each other than they to control samples for H3K4me1, H3K27ac and ATAC signal.

We are of the opinion that this observation and our differential analysis are both indicators of epigenetic memory and that for enhancers we do not observe a “normalization” of the epigenome post WL.

For PCA in Fig. 5b (H3K4me1 in enhancers) geometric distances between condition means:

	C	CC s	H	HC
C	0	144.403482	224.340597	207.778433
CC_s	144.403482	0	240.01544	233.232768
H	224.340597	240.01544	0	139.3987
HC	207.778433	233.232768	139.3987	0

	CC 1	CCC	HH	HHC

CC 1	0	157.483154	209.164584	245.968799
CCC	157.483154	0	241.211463	295.927407
HH	209.164584	241.211463	0	186.72959
HHC	245.968799	295.927407	186.72959	0

Based on these quantifications we conclude that indeed HC and HHC samples cluster closer to H and HH samples than to age matched controls (HC: 139.4 vs 233.24 and HHC: 186.73 vs 295.92) and both obese and WL adipocyte samples clustered far away from controls in the PC1/PC2 space of H3K4me1 enhancer quantification.

For PCA in Fig. 5c (ATAC in enhancers) geometric distances between condition means:

	C	CC s	H	HC
C	0	151.400118	243.08018	202.662801
CC s	151.400118	0	244.06656	212.51677
H	243.08018	244.06656	0	158.280404
HC	202.662801	212.51677	158.280404	0

	CC 1	CCC	HH	HHC
CC 1	0	183.118733	206.139974	185.423966
CCC	183.118733	0	255.821524	256.19399
HH	206.139974	255.821524	0	171.084296
HHC	185.423966	256.19399	171.084296	0

Based on these quantifications we conclude that indeed HC and HHC samples cluster closer to H and HH samples than to age matched controls (HC: 158.28 vs 212.52 and HHC: 171.08 vs 256.19) and both obese and WL adipocyte samples clustered far away from controls in the PC1/PC2 space of ATAC enhancer quantification.

For PCA in Extended Data Fig. 7d (H3K4me1 in enhancers) geometric distances between condition means:

	C	CC_1	CC_s	CCC	H	HC	HH	HHC
C	0	132.274217	141.723054	136.409499	221.864718	205.770115	232.831608	238.540547
CC_1	132.274217	0	141.187954	121.490456	191.700623	179.638275	194.840662	202.038104
CC_s	141.723054	141.187954	0	123.372984	224.855753	219.827717	242.353089	258.950016
CCC	136.409499	121.490456	123.372984	0	218.361084	209.07934	222.335276	234.14654
H	221.864718	191.700623	224.855753	218.361084	0	143.044975	120.781174	158.440403
HC	205.770115	179.638275	219.827717	209.07934	143.044975	0	133.528387	137.382884
HH	232.831608	194.840662	242.353089	222.335276	120.781174	133.528387	0	113.844406
HHC	238.540547	202.038104	258.950016	234.14654	158.440403	137.382884	113.844406	0

Based on these quantifications we conclude that indeed HC and HHC samples cluster closer to H and HH samples than to age matched controls (HC: 143.05 vs 219.87 and HHC: 113.84 vs

234.15) and both obese and WL adipocyte samples clustered far away from controls in the PC1/PC2 space of **joint H3K4me1** enhancer quantification.

For PCA in Extended Data Fig. 7e (H3K27ac of enhancers) geometric distances between condition means:

	C	CC_s	H	HC
C	0	147.168728	210.79029	197.17311
CC_s	147.168728	0	215.704831	205.709197
H	210.79029	215.704831	0	166.388917
HC	197.17311	205.709197	166.388917	0

	CC_1	CCC	HH	HHC
CC_1	0	194.283763	194.023037	220.718887
CCC	194.283763	0	249.930524	282.801231
HH	194.023037	249.930524	0	146.869828
HHC	220.718887	282.801231	146.869828	0

Based on these quantifications we conclude that indeed HC and HHC samples cluster closer to H and HH samples than to age matched controls (HC: 166.39 vs 205.71 and HHC: 146.87 vs 282.80) and both obese and WL adipocyte samples clustered far away from controls in the PC1/PC2 space of **H3K27ac** enhancer quantification.

Reviewer Reports on the First Revision:

Referees' comments:

Referee #1 (Remarks to the Author):

The authors did a very nice job of addressing all comments and providing new data where requested. I have no further concerns.

Referee #2 (Remarks to the Author):

The authors have addressed my comments.

Referee #3 (Remarks to the Author):

The authors have been highly responsive to my previous review, They have performed new experiments and analyses to further support and strengthen their conclusions.

Referee #5 (Remarks to the Author):

Review

Hinte et al addressed a fascinating topic of epigenetic memory linked to obesity and weight loss in mice and humans in adipose tissue. The findings are interesting and novel, and the transcriptional/epigenetic dataset of the mouse and the new human data would be useful for future studies. While many of the concerns raised by reviewers are largely sufficiently resolved in the revised manuscript, yet I have comments and concerns related to the statistical analysis, lack of controls in the data analysis of the snRNA-seq data as well as clarity/precision of the text and figures. In addition, a comparison between the mouse and human gene signatures is lacking, as well as orthogonal data such as proteins or metabolites to confirm the potential functionality of the expression changes.

My main comments are detailed bellow:

1. The number of samples from the MTSS and LTSS studies per condition should be clearly stated in the main text (i.e. n=xxx), noting the distribution of males/females and distribution of ages. This was also noted multiple times in previous reviews and is not resolved. In addition, please provide the following information on how does age and sex distribution/different across conditions (lean/obese) and the different cohorts. This information should appear as a figure panel and in the main text and in a supplementary table (as provided partially in Reviewer Fig. 14).

2. Regarding the human “memory” genes (i.e. Figure 1) – How does this differ between scAT and omAT? Specifically, IGF1 is shown to be common, yet what about DUSP1 or ID1 etc. Are these genes unique to adipocytes or shared across cell types? Some more details related to these genes and their context from the literature is missing in the main text/discussion, as well as a pathway analysis similar to the mouse dataset. Importantly, a comparison of the mouse dataset to these observed human signatures is generally lacking, yet is of high importance. A direct comparison of gene signatures should be done (beyond the current statement of “mirroring our findings from human adipocytes”).

3. Cell type frequency: The comparison of cell type frequencies across samples is not done properly. The figures presented are of the stacked bar plot are not suitable for this comparison (Figure 3b-c, Figure 6a and Extended Data Fig. 2b,c,e,f,g,i). As this is a claim of interest for the authors to make – they should align cell types across conditions and samples (can be done by bar plot or box plot). Moreover, to directly compare you need to add a p-value to test for the significance of these changes in relative abundance across conditions (which takes into account the variability across samples), and correct for multiple hypothesis.

4. Quality control graphs for the snRNA-seq and p-values are missing:

a. Quality of libraries as quantified by:

i. Number of nuclei per sample/condition.

ii. the distribution of number of genes/UMIs detected – across samples and conditions.

iii. distribution of number of genes/UMIs detected per sample.

b. Cell type identification:

i. How does your atlas compare to recently published data of adipose diversity?

ii. Sub-clusters such as ASPC1-3 should be further explained – showing distinct expression signature and marker genes, as well as distribution of samples per cluster to show if these are dependent on an individual or a condition or a cohort.

c. DEGs:

i. The expression of the major down/up regulated genes retained at T1 should be shown across the different cohorts (this could be a heatmap of genes vs. samples or cells per sample)

5. The loss of TREM2 as a marker of the LAMs is concerning – a figure showing other known LAM markers in lean vs. obese would be helpful to resolve this issue.

6. Statistical analysis: Although widely used, the Wilcoxon rank-sum test applied here is noisy and does not model the background noise in single cell RNA-seq datasets. Other alternatives within the Seurat package would be preferred, such as the poison or MAST test. This can be done as a validation on one of the samples to show the conclusions are robust to the DE test used.

7. Cell-cell interaction analysis – Standard statistical measures used to “score for significance” of interactions are based on permutation tests as outline in the CellPhoneDB method. Yet, as these are hard to implement correctly, and the predicted interactions are hard to validate - an alternative is to apply an orthogonal method, such as NichNet algorithm that also scores interactions based on signaling pathways (or the CellPhoneDB algorithm that integrates statistical analysis). The overlap in the predictions is expected to be partial, given the approach and assumptions made by each algorithm, but could help increase the confidence and point out the most robust interactions and signaling pathways.

8. The following statement is not clear and not supported by the referred figure panels: “By SNP-based demultiplexing we found evidence of inter-individual variations, possibly also affected by sampling during surgery (Extended Data Fig. 2c,f).”

- a. Is this genetic variation?
- b. How do you see the effect of the surgery?
- 9. Many figures are lacking details: please make sure all figures and legends are well annotated and provide enough information to interpret. Specifically:
 - a. Extended Data Figure 1 – cell types names are not clear, and missing also in the legend. For example: MesoCs.
 - b. Extended Data Figure 1g – Missing legend and labels on the figures – what are D0-D7? Which samples are lean/obese? What does the color scale represent (scaled number of DEs?).
 - c. Extended Data Figure 1k – missing x-axis labels.
 - d. Extended Data Fig. 3i: Y-axis label – percent of what?
 - e. Extended Data Fig. 3i: Can't see the difference between CC_i and CCC or C and CC_s – both are purple/dashed lines that are overlapping.
 - f. Extended Data Fig. 4d: dots overlap text, make it impossible to read.
 - g. Extended Data Fig. 4b: The figure was clearly modified as circles are elongated while the legend was not – i.e. the dot scales are meaningless.
 - h. Figure 4f-i: grey labels are impossible to read.
 - i. Figure 6n: the top figure seems to be cropped in a way that chopped the numbers and the bottom title overlaps the graph.
 - j. Thorough out all figure - color scales are missing a title. For example - Extended Data Fig. 5: I assume you are displaying scaled average expression for 5a, and -log-p-value for 5b-e.
 - k. Beyond these clarifications a technical/design point: Figures panels are not aligned and panels are not well spaced within the figure.
 - l. Of note: I did not point out EVERY correction- but I hope you can generalize from these comments.
- 10. Clarity of the text: Cell type names and conditions are not intuitive and challenges the reader (in text, legends and figures) – why not use the full words or the well know cell type names – in some cases this does not even add characters (if this was the consideration):
 - a. MesoCS = mesothelial
 - b. T-cells, B-cells = vascular (why not lymphoid or T/B-cells)
 - c. neuronal like cells - NeurCs1
 - d. In the mice models: HC, CC_s, HHC, CCC – some are not even defined in the main text properly. I'm not even sure I understood correctly the annotations of: C, CC, CCC, and CC_s for example. As previously noted by one of the other reviewers.

Minor comments:

- 11. Multi-Omics Factor Analysis (MOFA) – is not a well-known term and should be clarified in the main text. Specifically, what are you predicting, and what do the factors represent?
 - 1. Supplementary tables with samples, differential genes, and differential pathways are missing.
 - 2. Reviewer Fig. 6 and 7 are interesting and could be added to the supplement.
 - 3. Orthogonal information related to proteins or metabolites is missing in the manuscript, to show the transcriptional/epigenetic changes translate to potentially functional differences.

Author Rebuttals to First Revision:

Rebuttal letter to the manuscript: “Adipose tissue retains an epigenetic memory of obesity that persists after weight loss”

We thank the editor and all the referees for their careful assessment of our revised manuscript and rebuttal. We were very happy to read that referees 1-3 are satisfied with the new data and analyses we have provided. In the following we address and answer the additional points raised by the new Referee #5 (replacement of Referee #4).

Referee #1 (macrophage/metabolic physiology):

The authors did a very nice job of addressing all comments and providing new data where requested. I have no further concerns.

Referee #2 (immune cells/adipose/weight loss):

The authors have addressed my comments.

Referee #3 (metabolic physiology/epigenetics): Alyssa Hasty

The authors have been highly responsive to my previous review. They have performed new experiments and analyses to further support and strengthen their conclusions.

Referee #5 (scRNA-seq) (replacement of Referee #4):

Hinte et al addressed a fascinating topic of epigenetic memory linked to obesity and weight loss in mice and humans in adipose tissue. The findings are interesting and novel, and the transcriptional/epigenetic dataset of the mouse and the new human data would be useful for future studies. While many of the concerns raised by reviewers are largely sufficiently resolved in the revised manuscript, yet I have comments and concerns related to the statistical analysis, lack of controls in the data analysis of the snRNA-seq data as well as clarity/precision of the text and figures. In addition, a comparison between the mouse and human gene signatures is lacking, as well as orthogonal data such as proteins or metabolites to confirm the potential functionality of the expression changes.

We thank the referee for their comments and critical evaluation of our work, and for acknowledging its relevance for the scientific community. In response to the suggestions, we have conducted additional analyses. These analyses have reinforced our confidence in our results and their interpretation. Specifically, we find that similar pathways are persistently altered in both mouse and human adipocytes. Moreover, our “transcriptional retention” analysis results have proven robust, regardless of the differential expression testing method employed.

In summary, we have made these major changes to the manuscript:

- We have added new Extended Data Figures and Tables
- We have included a small graphical representation and table containing age, sex and BMI information in Figure 1
- We included GSEA results of human “memory” DEGs of adipocytes in the manuscript
- We have uploaded lists of differentially expressed genes onto GitHub (https://github.com/vonMeyennLab/AT_memory) that will be available as Source Data

We hope the referee shares our enthusiasm for this work and that they find our revisions of the manuscript sufficient to warrant publication in *Nature*.

COMMENT: Several Figures in the rebuttal letter are replicated from the main manuscript and placed directly with the answers to improve readability. We have clearly indicated where the same data is presented in the revised manuscript and sometimes even kept the labels of the panels from the manuscript to make the identification very clear.

We have also updated the numbering of Figures and Tables in the Manuscript and refer in the rebuttal letter always to the new revised numbering scheme.

My main comments are detailed below:

1. The number of samples from the MTSS and LTSS studies per condition should be clearly stated in the main text (i.e. $n=xxx$), noting the distribution of males/females and distribution of ages. This was also noted multiple times in previous reviews and is not resolved. In addition, please provide the following information on how does age and sex distribution/different across conditions (lean/obese) and the different cohorts. This information should appear as a figure panel and in the main text and in a supplementary table (as provided partially in Reviewer Fig. 14).

We thank the referee for the comment. In Extended Data Table 1 (Reviewer Table 1) we are providing all information regarding sex distribution, age and clinical parameters. Further, we have now included a graphical representation of our studies and table containing BMI, age and sex information in new main Fig. 1a,b (Reviewer Fig. 1). Further, we have included information on sex in the figure legends for each UMAP plot.

Reviewer Table 1 [data also presented in Extended Data Table 1]: Clinical parameters from the MTSS, LTSS and NEFA studies from T0, T1 and lean donors provided as mean \pm SD.

Parameter	lean_MTSS	responder_MTSS	lean_LTSS	responder_LTSS	lean_NEFA	responder_NEFA
BMI Loss (%)	NA	31.26 \pm 4.77	NA	26.21 \pm 0.92	NA	34.22 \pm 7.61
Sex (m/f)	1/4	2/6	2/3	2/3	0/8	0/7
Lipid lowering medication at T0 (yes/no)	0/5	0/8	0/5	0/5	0/8	1/7
Anti-diabetic medication at T0 (yes/no)	0/5	0/8	0/5	0/5	0/8	0/7
Age T0 (years)	37.05 \pm 8.29	39.5 \pm 5.53	45.34 \pm 3.01	48.92 \pm 4.2	49.88 \pm 13.9	38.62 \pm 12.64
BMI T0 (kg/m ²)	24.97 \pm 2.86	48.19 \pm 7.01	25.04 \pm 1.22	46.96 \pm 6.38	26.11 \pm 2.9	40.35 \pm 2.15
Fasting plasma glucose T0 (mmol/l)	NA	NA	4.8 \pm 0.35	5.82 \pm 0.44	5.15 \pm 0.46	5.19 \pm 0.24
HbA1c T0 (%)	5.14 \pm 0.07	5.75 \pm 0.23	5.4 \pm 0.77	5.87 \pm 0.19	5.36 \pm 0.32	5.23 \pm 0.27
HOMA IR T0	NA	NA	1.95 \pm 0.86	5.14 \pm 0.9	1.3 \pm 0.53	3.18 \pm 1.66
Fasting plasma insulin T0 (pmol/l)	NA	NA	64.14 \pm 29.32	137.68 \pm 24.8	35.17 \pm 14.37	86.59 \pm 46.19
Cholesterol T0 (mmol/l)	4.9 \pm 1.18	4.86 \pm 1.07	4.3 \pm 0.75	4.71 \pm 1.59	4.44 \pm 0.9	4.45 \pm 0.91
Triglycerides T0 (mmol/l)	1.23 \pm 0.37	1.98 \pm 0.55	0.82 \pm 0.14	2.12 \pm 0.92	0.76 \pm 0.26	1.18 \pm 0.43
HDL-cholesterol T0 (mmol/l)	2.24 \pm 0.72	1.09 \pm 0.19	1.82 \pm 0.34	1.13 \pm 0.08	1.47 \pm 0.4	1.14 \pm 0.13
LDL-cholesterol T0 (mmol/l)	2.04 \pm 1.54	3.39 \pm 1.04	2.31 \pm 0.8	2.97 \pm 1.17	NA	NA
Age T1 (years)	NA	41.5 \pm 5.53	NA	50.92 \pm 4.14	NA	40.62 \pm 12.64
BMI T1 (kg/m ²)	NA	33.26 \pm 6.25	NA	34.66 \pm 4.86	NA	26.51 \pm 2.04
Fasting plasma glucose T1 (mmol/l)	NA	NA	NA	4.93 \pm 0.43	NA	5.05 \pm 0.14
HbA1c T1 (%)	NA	5.41 \pm 0.54	NA	5.12 \pm 0.45	NA	5.16 \pm 0.2
HOMA IR T1	NA	NA	NA	1.67 \pm 0.25	NA	0.87 \pm 0.18
Fasting plasma insulin T1 (pmol/l)	NA	NA	NA	52.84 \pm 5.78	NA	23.28 \pm 4.61
Cholesterol T1 (mmol/l)	NA	6.37 \pm 2.21	NA	4.07 \pm 0.76	NA	3.72 \pm 0.7
TG T1 (mmol/l)	NA	3.79 \pm 4.05	NA	0.93 \pm 0.17	NA	0.64 \pm 0.14
HDL-cholesterol T1 (mmol/l)	NA	1.16 \pm 0.08	NA	1.65 \pm 0.57	NA	1.54 \pm 0.27
LDL-cholesterol T1 (mmol/l)	NA	4.05 \pm 1.64	NA	2.42 \pm 1.01	NA	NA

Reviewer Figure 1 [data also presented in Figure 1]: **a**, Human study design: omAT and scAT biopsies were collected from people living with obesity during bariatric surgery (T0) and two years post surgery (T1). Only individuals that had at lost at least 25% of BMI compared to T0 were included. omAT and scAT biopsies were

collected from normal-weight/lean individuals from the same studies. Biopsies were sourced from three European studies: MTSS, LTSS and NEFA. **b**, Distribution of sex, age, starting BMI, and BMI loss of lean and obese donors. **across all studies**

We have made the following adjustments to the text:

Lines 90-92: “We first analyzed omAT from the MTSS ($n = 5$ lean, 1 males, 4 females; $n = 8$ obese, 2 males, 6 females) and LTSS ($n = 5$ lean, 2 males, 3 females; $n = 5$ obese, 2 males, 3 females) studies (see Methods for details).”

Lines 111-118: “Intrigued by these findings, we performed the same analysis with subcutaneous AT (scAT) biopsies from never-obese (normal-weight) healthy individuals and people living with obesity (but without diabetes) at T0 and T1 from two independent studies, termed LTSS ($n = 5$ lean, 2 males, 3 females; $n = 5$ obese, 2 males, 3 females) and NEFA ($n = 8$ lean, all female; $n = 7$ obese, all female) (see Methods for details), including only patients exhibiting a minimum reduction of 25% in BMI (Fig. 1 a,b and Extended Data Table 1).”

Overall, the age distribution is comparable between lean donors and individuals (formerly) living with obesity (**Reviewer Fig. 2**) within each cohort but does slightly differ between the cohorts. Of note, all samples used in our work were collected independently in each study. In our analyses we did not perform cross cohort comparisons but focused on the comparison of samples from individuals (formerly) living with obesity to samples from normal weight/lean donors from the same study. We find that in each study cellular transcriptional signatures are retained. Only for adipocytes we performed an analysis from two studies after carefully assessing how well these datasets could be integrated without overcorrection of biological differences. Finally, the requested pathway analysis (please see answer to your question #2) showed that the transcriptional changes in all studies are related to similar pathways.

Reviewer Figure 2: Age distribution per cohort and group. Data points from T0 and T1 are from the same paired individual (T1 being 2 years post-surgery)

2. Regarding the human “memory” genes (i.e. Figure 1) – How does this differ between scAT and omAT? Specifically, IGF1 is shown to be common, yet what about DUSP1 or ID1 etc

We thank the referee for raising this point and we agree that comparing adipocytes of scAT and omAT is very interesting. We have compared the “memory” DEGs between adipocytes from omAT and scAT and have found that some DEGs overlap. Specifically, 13 DEGs that remain downregulated are shared and one DEG that remains upregulated (**Reviewer Fig. 3, Reviewer Table 2**).

The limited exact overlap of “memory” DEGs can be explained by the fact that we are comparing adipocytes from two considerably different adipose tissue depots across cohorts. Massier *et al.* (PMID: 36922516) have shown in an extensive cross-cohort and cross-depot analysis that snRNA-seq data from human adipocytes of different adipose tissue depots are not comparable, supporting our observation that the “memory” DEGs in each depot differ.

Reviewer Figure 3: Overlap of “memory” DEGs between omental and subcutaneous adipocytes.

Reviewer Table 2: Shared memory DEGs between human adipocytes of scAT and omAT

Common downregulated “memory” DEG	Common upregulated “memory” DEG
LRP1B	SAAI
IGF1	
SRPX	
GLUL	
ABHD2	
KIFBP	
FAM13C	
SYBU	
RORB	
ANKRD20A1	
NFKBIA	
GGCT	
CECR2	

Are these genes unique to adipocytes or shared across cell types??

The referee raises an interesting point. We have now checked the expression of *IGF1*, *DUSP1*, *IDH1*, *PDE3A*, *GPX3* and *GLUL* across cell types and whether these genes are also “memory” DEGs in other cell types. We have found that several of these genes are also expressed in other cell types (**Reviewer Fig. 4**). For example, *PDE3A* is also highly expressed in pericytes and smooth muscle cells in omAT and *IGF1* (in line with literature) is also expressed in

macrophages. It has been shown that with obesity, macrophages start expressing *IGF1* while it is downregulated in adipocytes (Chang *et al.* (PMID: 26663512)).

Reviewer Figure 4: a, Dotplot depicting the expression of *PDE3A*, *IDH1*, *LPIN1* and *IGF1* in MTSS and LTSS (b) omAT samples. c, Dotplot depicting the expression of *GPX3*, *DUSP1*, *GLUL* and *IGF1* in NEFA and LTSS (d) scAT samples

We next analyzed whether the genes found to be adipocyte “memory” DEGs, are also “memory” DEGs in other cell types in each study. Besides a few exceptions, the transcriptional “memory” is specific to adipocytes. We have summarized our findings for each DEG below.

IGF1: Is also a memory DEG (same direction) in EndoVCs and MesoCs in the LTSS omAT samples, in EndoVCs in the LTSS scAT samples but is not a memory DEG in other cell types in the MTSS and NEFA samples.

LPIN1: Is only a memory DEG in adipocytes.

IDH1: Is only a memory DEG in adipocytes.

PDE3A: Is also a memory DEG (same direction) in EndoACs and T-cells of omental AT of the LTSS study and in APCs1 in the MTSS study.

GPX3: Is only a memory DEG in adipocytes.

GLUL: Is also downregulated in APCs1 in scAT samples of the LTSS study.

DUSP1: Is also a downregulated in endothelial cells (all three) and macrophages (Macro1) in the scAT samples of the LTSS study.

Moreover, we will include the DEG lists from T0 vs lean and T1 vs lean for each cell type per cohort and for integrated adipocytes as Source Data files and on GitHub (https://github.com/vonMeyennLab/AT_memory). We believe that this will be useful for the reader in case they want to obtain information on gene expression regulation in other cell types. Additionally, the snRNA-seq data will be explorable in a shiny app where cell-type-specific expression can be queried and visualized by the user (link will be available upon acceptance).

Some more details related to these genes and their context from the literature is missing in the main text/discussion, as well as a pathway analysis similar to the mouse dataset.

We have now included the results from our earlier response to referee #3 (former Reviewer Fig. 13) and to this referee in the manuscript (new **Extended Data Fig 5**). We performed GSEA using the WikiPathways database for “memory DEGs” from adipocytes.

Overall, persistently “downregulated terms” in adipocytes are metabolism or adipogenesis related (**Reviewer Fig. 5a-d**) and “upregulated terms” are related to inflammatory, apoptotic, fibrotic (TGF β) signalling (**Reviewer Fig. 5e-h**) across tissues and studies. Of note, we did not obtain significant GSEA results for scAT of the LTSS study (**Reviewer Fig. 5g**). We included the corresponding plot for completeness of the analysis and added a note in the figure to emphasize that the enriched terms are not significant based on the criteria we used for the other plots.

In mice, pathways related to ECM remodeling and inflammation, both linked to TGF β and fibrotic signaling, remained upregulated post weight loss, whereas pathways linked to adipocyte function and metabolism remained downregulated (Fig. 3g,h), indicating that the transcriptional effects in our mouse and human samples are comparable.

We have made the following changes to the text:

Lines 139-143: “Gene set enrichment analysis (GSEA) of retained DEGs in adipocytes of each study showed persistent downregulation of pathways linked to adipocyte metabolism and function (Extended Data Fig. 5a-d.) and persistent upregulation of pathways linked to fibrosis (related to TGF β signalling) and apoptosis (Extended Data Fig. 5e-h).”

Lines 227-230: In summary, post-WL, adipocytes from mice maintained an upregulation of inflammatory- and extracellular matrix (ECM) remodelling-related pathways, while adipocyte-specific metabolic pathways remained downregulated (Fig. 3g,h), mirroring our findings from human adipocytes (Fig. 1h,j and Extended Data Fig. 5).

a

b

c

d

e

f

g

h

Reviewer Figure 5: [data also presented in Extended Data Fig. 5] **a**, Circular plot depicting the top (significant) persistently downregulated (memory) pathway terms in omental adipocytes of the MTSS study based on Wikipathways database. **b**, Circular plot depicting the top (significant) persistently downregulated (memory) pathway terms in omental adipocytes of the LTSS study based on Wikipathways database. **c**, Circular plot depicting the top (significant) persistently downregulated (memory) pathway terms in subcutaneous adipocytes of the LTSS study based on Wikipathways database. **d**, Circular plot depicting the top (significant) persistently downregulated (memory) pathway terms in subcutaneous adipocytes of the NEFA study based on Wikipathways database. **e**, Circular plot depicting the top (significant) persistently upregulated (memory) pathway terms in omental adipocytes of the MTSS study based on Wikipathways database. **f**, Circular plot depicting the top (significant) persistently upregulated (memory) pathway terms in omental adipocytes of the LTSS study based on Wikipathways database. **g**, Circular plot depicting the top persistently upregulated (memory) pathway terms in subcutaneous adipocytes of the LTSS study based on Wikipathways database. These are not significant. **h**, Circular plot depicting the top (significant) persistently upregulated (memory) pathway terms in subcutaneous adipocytes of the NEFA study based on Wikipathways database.

We have now added literature references to the reported memory DEGs, referring to earlier studies that have shown their involvement in adipocyte function or reported deregulation in obesity.

While we decided to present some “memory” DEGs as violin plots throughout the manuscript, as illustrative examples to demonstrate, alongside GSEA, the affected pathways, our study primarily reveals global changes at transcriptomic and epigenomic level, and we prefer to focus on these broader genomic effects rather than reducing our findings to a few specific genes through a “gene-centric” analysis and interpretation.

Importantly, a comparison of the mouse dataset to these observed human signatures is generally lacking, yet is of high importance. A direct comparison of gene signatures should be done (beyond the current statement of “mirroring our findings from human adipocytes”)

As suggested, we performed direct comparison of mouse and human adipocyte “memory” DEGs by analyzing orthologues and identify a number of ortholog genes as memory DEGs in both species, i.e. *SCD/Scd1*, *PDE3A/Pde3b*, *SVIL/Svil*, *ANK1/Ank2*, *PRDM16/Prdm16*, *CTSB/Ctsb*, *FTL/Ftl1*, *LIPA/Lipa* and *FKBP5/Fkbp5*.

It is worth noting, that human and mouse adipocytes are generally not well comparable on a transcriptional level. Emont *et al.* (PMID: 35296864) have reported that “In contrast to the relatively good cross-species concordance between immune cells, vascular cells, and ASPCs, mouse adipocytes do not map cleanly onto human adipocyte subpopulations”.

However, our GSEA results from mouse and human adipocytes do indicate that adipocyte function/metabolism is persistently downregulated (**Extended Data Fig. 5a-d and Fig. 3h**) whereas inflammatory and/or fibrotic signaling are persistently upregulated in both species (**Extended Data Fig. 5e-h and Fig. 3g**). Overall, this indicates that our human and mouse findings are comparable and similar underlying mechanisms are affected, while the specific genes involved might be more species specific.

3. *Cell type frequency: The comparison of cell type frequencies across samples is not done properly. The figures presented are of the stacked bar plot are not suitable for this comparison (Figure 3b-c, Figure 6a and Extended Data Fig. 2b,c,e,f,g,i). As this is a claim of interest for the authors to make – they should align cell types across conditions and samples (can be done by bar plot or box plot). Moreover, to directly compare you need to add a p-value to test for the significance of these changes in relative abundance across conditions (which takes into account the variability across samples), and correct for multiple hypothesis.*

We agree with the referee that stacked bar plots do not allow for a statistical analysis of differences in cellular composition. Therefore, we now performed paired Wilcoxon tests between T0 and T1 for human AT samples of each cohort and corrected for multiple testing using Benjamini, Krieger, and Yekutieli post-hoc corrections to calculate the FDR (**Reviewer Fig. 6**). We have included these results in new **Extended Data Fig. 4**.

While we do find significant cell compositional changes between T0 and T1 in the omAT samples of the MTSS study, we do not find these in the LTSS study's samples (**Reviewer Fig. 6a,b**). In the MTSS samples, we find significant differences in the abundance of MesoCs, NeurCs2 and FAPs (**Reviewer Fig. 6b**) (FDR < 0.05). As we did not find consistent cellular composition differences between T0 and T1 across these cohorts, we have adjusted our sentence in the MS:

Lines 102-104: *“While we did not observe consistent cellular composition differences between T0 and T1 in omAT (Extended Data Fig. 2b,e and Extended Data Fig. 4a,b), ...”*

For the scAT samples, our analysis revealed that there were no consistent cellular composition differences between T0 and T1 across cohorts, consistent with our previous statement (**Reviewer Fig. 6c,d**). Neither did we detect robust differences in cellular composition between T0 and lean or T1 and lean scAT or omAT samples. This might be because our analysis is underpowered, and we observe interindividual variability.

Reviewer Figure 6: [data also presented in Extended Data Fig. 4] **a**, Barplots depicting the relative cell type abundance in human *omAT* of the MTSS study in lean donors and at T0 and T1. Each dot indicates one biological replicate. Lines connecting dot from T0 to T1 indicate a paired sample. **b**, Barplots depicting the relative cell type abundance in human *omAT* of the LTSS study in lean donors and at T0 and T1. Each dot indicates one biological replicate. Lines connecting dot from T0 to T1 indicate a paired sample. **c**, Barplots depicting the relative cell type abundance in human *scAT* of the LTSS study in lean donors and at T0 and T1. Each dot indicates one biological replicate. Lines connecting dot from T0 to T1 indicate a paired sample. **d**, Barplots depicting the relative cell type abundance in human *scAT* of the NEFA study in lean donors and at T0 and T1. Each dot indicates one biological replicate. Lines connecting dot from T0 to T1 indicate a paired sample. Significance between T0 and T1 for a-d was calculated using paired multiple Wilcoxon tests with Benjamini, Krieger and Yekutieli post hoc test for multiple comparisons. Error bars represent SD.

In our mouse samples (**Figure 3b,c and Figure 6i,j**) we cannot perform such analyses because we pooled 5 mice per condition to ensure that our data is well represented. Further, we cannot use SNP based demultiplexing as we did for the human samples to extract information from individual samples because our mice are genetically identical.

Given that prior studies have also observed and reported macrophage infiltration in adipose tissue of obese mice (Sávári *et al.* (PMID: 33378646; Cottam *et al.* (PMID: 35618862); Reinisch *et al.* (PMID: 38360943)) and that our mouse study is well controlled, we think that our statements regarding Figures 3b,c and 6i,j are justified.

4. Quality control graphs for the snRNA-seq and p-values are missing:

We assume the referee is requesting p-values for differential gene expression analysis of our single nucleus datasets. As noted by the referee we used the Wilcoxon test and corrected for multiple testing using the adjusted p-value to identify DEGs. We have added this information to the figure legends. Other p-values are indicated with asterisks in the figures, or we have added the exact p-values to the figures themselves. More information on statistical testing will also be included in the Source Data files.

a. Quality of libraries as quantified by:

i. Number of nuclei per sample/condition.

ii. the distribution of number of genes/UMIs detected – across samples and conditions.

iii. distribution of number of genes/UMIs detected per sample.

We have now included the requested QC metrics as **Extended Data Fig.13 and Extended Data Tables 4-6**. Overall, we found that number of UMIs and features are consistent across samples within single cohorts (**Reviewer Fig. 7a**). The libraries from the MTSS T1 samples have fewer features/UMIs than other samples in this cohort (**Reviewer Fig. 7a,b**), but the differential analysis is not affected by this, and we have concluded that the integration is not affected by this because we can identify a cell type (NeurCs2) that is mainly/only found in these samples. Due to the pooling of individual adipose tissue samples from different donors, the number of nuclei recovered per individual differs within cohorts (**Reviewer Tables 4-5**).

Reviewer Figure 7: [data also presented in Extended Data Fig. 13] **a**, Violinplots depicting the number of genes detected (*nFeature_RNA*) and the number of unique molecular identifiers (*nCount_RNA*) per condition in the omAT samples from the MTSS (left), LTSS (second left) and in scAT samples from the LTSS (second from right) and NEFA (right) study. **b**, Violinplots depicting the number of genes detected (*nFeature_RNA*) and the number of unique molecular identifiers (*nCount_RNA*) per donor in the omAT samples from the MTSS (left) and LTSS (right) study. **c**, Violinplots depicting the number of genes detected (*nFeature_RNA*) and the number of unique molecular identifiers (*nCount_RNA*) per donor in scAT samples from the LTSS (left) and NEFA (right) study. **d**, Violinplots depicting the number of genes detected (*nFeature_RNA*) and the number of unique molecular identifiers (*nCount_RNA*) per assigned cell type in the omAT samples from the MTSS (left) and LTSS (right) study. **e**, Violinplots depicting the number of genes detected (*nFeature_RNA*) and the number of unique molecular identifiers (*nCount_RNA*) per assigned cell type in the scAT samples from the LTSS (left) and NEFA (right) study.

Reviewer Table 3: Number of nuclei passing QC thresholds per condition of human AT samples across cohorts. This table is **Extended Data Table 4**.

No. of nuclei after QC	Condition	Cohort	Tissue
4387	Lean	MTSS	omAT
7287	Res_T0	MTSS	omAT
7820	Res_T1	MTSS	omAT
7316	Lean	LTSS	omAT
7978	Res_T0	LTSS	omAT
7448	Res_T1	LTSS	omAT
4402	Lean	LTSS	scAT
3628	Res_T0	LTSS	scAT
7317	Res_T1	LTSS	scAT
3229	Lean	NEFA	scAT
15258	Res_T0	NEFA	scAT
13234	Res_T1	NEFA	scAT

Reviewer Table 4: Number of nuclei passing QC thresholds per donor in omAT samples. This table is **Extended Data Table 5**.

No. of nuclei after QC	Donor	Study	Tissue
315	Lean_D1	MTSS	omAT
2154	Lean_D2	MTSS	omAT
400	Lean_D3	MTSS	omAT
1125	Lean_D4	MTSS	omAT
393	Lean_D5	MTSS	omAT
562	Res_T0_D0	MTSS	omAT
573	Res_T0_D1	MTSS	omAT
596	Res_T0_D2	MTSS	omAT
2278	Res_T0_D3	MTSS	omAT
603	Res_T0_D4	MTSS	omAT
624	Res_T0_D5	MTSS	omAT
570	Res_T0_D6	MTSS	omAT
1481	Res_T0_D7	MTSS	omAT
703	Res_T1_D0	MTSS	omAT
631	Res_T1_D1	MTSS	omAT
674	Res_T1_D2	MTSS	omAT
534	Res_T1_D3	MTSS	omAT
933	Res_T1_D4	MTSS	omAT
1030	Res_T1_D5	MTSS	omAT
684	Res_T1_D6	MTSS	omAT
2631	Res_T1_D7	MTSS	omAT
323	Lean_D0	LTSS	omAT
1912	Lean_D1	LTSS	omAT
874	Lean_D2	LTSS	omAT
2545	Lean_D3	LTSS	omAT
1662	Lean_D4	LTSS	omAT
1155	Res_T0_D0	LTSS	omAT
1743	Res_T0_D1	LTSS	omAT
1951	Res_T0_D2	LTSS	omAT
1483	Res_T0_D3	LTSS	omAT
1636	Res_T0_D4	LTSS	omAT
1502	Res_T1_D0	LTSS	omAT
3343	Res_T1_D1	LTSS	omAT
1352	Res_T1_D2	LTSS	omAT
468	Res_T1_D3	LTSS	omAT
783	Res_T1_D4	LTSS	omAT

Reviewer Table 5: Number of nuclei passing QC thresholds per donor in scAT samples. This table is **Extended Data Table 6**.

No. of nuclei after QC	Donor	Study	Tissue
672	Lean_D0	LTSS	scAT
1783	Lean_D1	LTSS	scAT
714	Lean_D2	LTSS	scAT
357	Lean_D3	LTSS	scAT
1056	Lean_D4	LTSS	scAT
772	Res_T0_D0	LTSS	scAT
1036	Res_T0_D1	LTSS	scAT
602	Res_T0_D2	LTSS	scAT
776	Res_T0_D3	LTSS	scAT
434	Res_T0_D4	LTSS	scAT
768	Res_T1_D0	LTSS	scAT
2650	Res_T1_D1	LTSS	scAT
2397	Res_T1_D2	LTSS	scAT
1256	Res_T1_D3	LTSS	scAT
245	Res_T1_D4	LTSS	scAT
419	Lean_D0	NEFA	scAT
394	Lean_D1	NEFA	scAT
410	Lean_D2	NEFA	scAT
405	Lean_D3	NEFA	scAT
390	Lean_D4	NEFA	scAT
380	Lean_D5	NEFA	scAT
382	Lean_D6	NEFA	scAT
449	Lean_D7	NEFA	scAT
3305	Res_T0_D0	NEFA	scAT
2965	Res_T0_D1	NEFA	scAT
1528	Res_T0_D2	NEFA	scAT
2048	Res_T0_D3	NEFA	scAT
1857	Res_T0_D4	NEFA	scAT
1511	Res_T0_D5	NEFA	scAT
2090	Res_T0_D6	NEFA	scAT
2825	Res_T1_D0	NEFA	scAT
661	Res_T1_D1	NEFA	scAT
1950	Res_T1_D2	NEFA	scAT
1444	Res_T1_D3	NEFA	scAT
1115	Res_T1_D4	NEFA	scAT
3097	Res_T1_D5	NEFA	scAT
2096	Res_T1_D6	NEFA	scAT

Reviewer Figure 8: [data also presented in Extended Data Fig. 12] **a**, Violinplots depicting the number of genes detected (*nFeature_RNA*) and the number of unique molecular identifiers (*nCount_RNA*) per condition of mouse epiAT samples. **b**, Violinplots depicting the number of genes detected (*nFeature_RNA*) and the number of unique molecular identifiers (*nCount_RNA*) per cell type from mouse epiAT samples.

Moreover, we have added the requested QC plots for our mouse snRNA-seq data (**Reviewer Fig. 8 and Table 6**) as **Extended Data Fig. 12 and Table 3**.

We now refer to all QC plots and tables in the methods section of the revised manuscript.

Reviewer Table 6: Number of nuclei passing QC thresholds per condition of mouse epiAT samples. This table is **Extended Data Table 3**.

No. of nuclei after QC	condition
9491	C
3416	CC
3464	CCC
8499	H
4672	HC
11433	HH
7089	HHC
6711	CCH
8954	HCH

b. Cell type identification:

i. How does your atlas compare to recently published data of adipose diversity?

The referee raises an important point regarding the comparability of our datasets to other published human adipose tissue snRNA-seq data sets. Indeed, to assign major cell types in our data, we used reference mapping against the adipose tissue atlas dataset published by Emont *et al.* (PMID: 35296864). Overall, we find that we can detect the same cell types in our data as in this reference data set (**Reviewer Fig. 9**). The confidence of reference mapping is very high

for most nuclei (**Reviewer Fig. 9**). However, cell subtypes such as specific macrophage subclusters are not always clear and for the immune cell sub-clustering performed in the previous round of revision we also used other databases/literature to identify subtypes. We have added this information in **Extended Data Fig. 3** to our manuscript.

Reviewer Figure 9: [data also presented in Extended Data Fig. 3] **a**, UMAP visualization representing omAT pools from the LTSS study colored by predicted cell subtypes from the Emont et al. visceral AT dataset from Caucasian individuals. Feature plots showing reference mapping scores illustrating how well omAT dataset from LTSS study maps to the Emont et al. dataset. **b**, UMAP visualization representing omAT pools from the MTSS study colored by predicted cell subtypes from the Emont et al. visceral AT dataset from Caucasian individuals. Feature plots showing reference mapping scores illustrating how well omAT dataset from MTSS study maps to the Emont et al. dataset. **c**, UMAP visualization representing scAT pools from the LTSS study colored by predicted cell subtypes from the Emont et al. subcutaneous AT dataset from Caucasian individuals. Feature plots showing reference mapping scores illustrating how well scAT dataset from LTSS study maps to the Emont et al. dataset. **d**, UMAP visualization representing scAT pools from the NEFA study colored by predicted cell subtypes from the Emont et al. subcutaneous AT dataset from Caucasian individuals. Feature plots showing reference mapping scores illustrating how well scAT dataset from NEFA study maps to the Emont et al. dataset.

For our mouse snRNA-seq datasets we also used reference mapping against the dataset from Emont et al. to validate our cell type annotation based on published markers (**Reviewer Figure 10**).

Reviewer Figure 10: [data also presented in Extended Data Fig. 7 and/or 12] UMAP visualization representing integrated epiAT samples from the weight loss study (C, CC, CCC, H, HH, HC, HHC) and from the ‘yo-yo’ study (CCH, HCH) colored by predicted cell subtypes from the Emont et al. mouse epididymal AT dataset. Feature plots showing reference mapping scores illustrating how well these datasets maps to the Emont et al. dataset.

ii. Sub-clusters such as *ASPC1-3* should be further explained – showing distinct expression signature and marker genes, as well as distribution of samples per cluster to show if these are dependent on an individual or a condition or a cohort.

We thank the referee for their suggestion. We have now refined our “APC” subtyping using markers provided in the recent review by Maniyadath *et al.* (PMID: 36889280) and Merrick *et al.* (PMID: 31023895), which we have now also cited in our manuscript. We could now indeed identify fibro-adipogenic progenitors (FAPs) as subclusters in scAT (former APCs2) and omAT (former APCs3) respectively (**Reviewer Fig. 11**) and have therefore renamed them throughout the manuscript and in the figures. For scAT we find that expression of *DPP4* distinguishes the FAPs from other APC clusters more so than other marker genes (**Reviewer Fig. 11c,d**). The other two clusters are not clearly identifiable as a specific subtype but do express markers of preadipocytes/committed progenitors (**Reviewer Fig. 11**).

We have also compared APCs between omAT and scAT and find that apart from the FAPs the other two clusters are not the same. We do find that the APC subtyping between samples from the NEFA cohort and the LTSS cohort of scAT apart from FAPs is not consistent. Therefore, we have decided to keep the current APC nomenclature for the non-FAP clusters.

Reviewer Figure 11: Expression of preadipocyte and FAP marker genes across APC clusters in APCs from MTSS omAT (a), LTSS omAT (b), LTSS scAT (c) and NEFA scAT (d).

In all cohorts we could find APC subclusters including FAPs in most if not all samples (please see **Reviewer Fig. 6** in answer to your question #3).

c. DEGs:

i. The expression of the major down/up regulated genes retained at T1 should be shown across the different cohorts (this could be a heatmap of genes vs. samples or cells per sample)

We have now visualized the expression of major up- and downregulated “memory” DEGs identified in each cohort that are also related to the GSEA results (**Reviewer Fig. 5 and new Extended Data Fig. 5**) and the DEGs identified after integration of adipocytes per tissue, shown in **Fig 1**. These heatmaps (**Reviewer Fig. 12**) display scaled pseudo bulk expression in adipocytes per donor. Overall expression trends are similar between individuals within a cohort and for many DEGs, but not for all, across cohorts.

Reviewer Figure 12: a, Scaled pseudobulk expression of “memory” DEGs in omental adipocytes across donors in the MTSS study. b, Scaled pseudobulk expression of “memory” DEGs in omental adipocytes across donors in the LTSS study. c, Scaled pseudobulk expression of “memory” DEGs in subcutaneous adipocytes across donors in the NEFA study. d, Scaled pseudobulk expression of “memory” DEGs in subcutaneous adipocytes across donors in the LTSS study.

We performed SNP-based demultiplexing of our pooled snRNA-seq data to assign nuclei to individuals mainly for *quality control* and *variability analyses* across all donors. Of note, we do not have SNP information for each donor and therefore we cannot assign these “snRNA-seq-individuals” to tissue donors and perform correlation analyses of adipocyte gene expression and clinical parameters. Further, the pseudobulk approach at the “SNP demultiplexed individual” level does not take cellular heterogeneity into account and the accuracy is highly dependent on technical factors like sequencing depth, quality, pooling strategy, and computational complexity. Given that our study is focusing on cell type specific

changes persistently up/down regulated after weight loss within a single cohort, we decided to not include the gene-centric expression heatmap either at the pseudobulk or at the cellular level.

5. The loss of TREM2 as a marker of the LAMs is concerning – a figure showing other known LAM markers in lean vs. obese would be helpful to resolve this issue.

We thank the referee for their suggestion. We believe that we cannot detect LAMs based on *TREM2* expression due to technical limitations of snRNA-seq of human adipose tissue. To test whether this is the case, we reanalyzed scRNA-seq and snRNA-seq data of human myeloid cells from Massier *et al.* (PMID: 36922516) and used markers for LAMs from Jaitin *et al.* (PMID: 31257031) to identify LAMs in these datasets. While we can identify LAMs in the integrated dataset of scRNA-seq and snRNA-seq data (**Reviewer Fig. 13a**; cluster 2), when we checked *TREM2* expression, we found that only scRNA-seq captures *TREM2* expression reliably (**Reviewer Fig. 13b**).

Reviewer Figure 13: *a*, Dotplot depicting the expression of LAM markers from Jaitin *et al.* in myeloid dataset from Massier *et al.*. *b*, Dotplot depicting the expression of *TREM2* in cluster 2 from a split by technique used.

We were still able to identify LAMs via reference mapping based on the PBMC reference data set from the Human BioMolecular Atlas Program (<https://portal.hubmapconsortium.org/>) and using other markers from Jaitin *et al.* (PMID: 31257031) instead of *TREM2*.

Although we could not detect *TREM2* at a substantial level, other LAM markers like *CD9* and *LPL* (Maniyadath *et al.* (PMID: 36889280) and Jaitin *et al.* (PMID: 31257031)) are expressed and appear to be higher expressed at T0 than samples of lean individuals (**Reviewer Fig. 14**). However, we can only reliably assign very few cells as LAMs (**Reviewer Fig. 14**) and therefore do not refer to LAMs or other macrophage subclasses in our manuscript.

Reviewer Figure 14: *a*, Expression of CD9 and LPL (LAM markers) in cells identified as LAMs across conditions in omAT of the MTSS study. *b*, Expression of CD9 and LPL (LAM markers) in cells identified as LAMs across conditions in omAT of the LTSS study. *c*, Expression of CD9 and LPL (LAM markers) in cells identified as LAMs across conditions in scAT of the LTSS study. *d*, Expression of CD9 and LPL (LAM markers) in cells identified as LAMs across conditions in scAT of the NEFA study.

Of note, we did not include the immune cell subtyping analyses in the revised version of the manuscript. An extensive analysis and phenotyping of different immune cells of (human) adipose tissue in the context of weight loss, while very relevant, is beyond the scope of this manuscript, especially given the above results and technical limitations of snRNA-seq.

6. Statistical analysis: Although widely used, the Wilcoxon rank-sum test applied here is noisy and does not model the background noise in single cell RNA-seq datasets. Other alternatives within the Seurat package would be preferred, such as the poison or MAST test. This can be done as a validation on one of the samples to show the conclusions are robust to the DE test used.

The referee voices an important point regarding the reproducibility and robustness of our cell type specific DE analysis. We have found that we cannot use Poisson or other tests that require raw integer counts as input such as DESeq2 because we used the tool SoupX to adjust for potential ambient RNA.

Therefore, we have now used both MAST (as suggested by the referee) and likelihood-ratio (LR) tests to validate our findings for our cell type specific DE analysis in the NEFA cohort using the same cutoffs for adjusted p-value (<0.01) and absolute log₂FC (>0.5) for all tests.

The DEGs obtained with the MAST test match the DEGs obtained with the Wilcoxon test very well. The MAST test uncovers even more DEGs for some cell types. For instance, 96-97% of upregulated DEGs for adipocytes at T0 in the NEFA cohort found by the Wilcoxon test are also detected by the MAST test (**Reviewer Fig. 15**) and 82-91% of downregulated DEGs for adipocytes in the NEFA cohort are also detected by the MAST test (**Reviewer Fig. 15**). Additionally, the majority of DEGs identified by the Wilcoxon test and MAST test are also detected by using LR (**Reviewer Fig. 15**). We made similar observation for other cell types.

Reviewer Figure 15: Comparison of DE results for adipocytes of NEFA study using MAST or Wilcoxon test at T1 (left) and overlap of upregulated DEGs for adipocytes at T1 from MAST, Wilcoxon or likelihood-ratio tests (right).

Moreover, we performed transcriptional retention analysis with the DEGs obtained from the MAST and LR test. In line with our previous results most cell types show transcriptional retention and for most cell types; the proportion between restored and non-restored DEGs was similar between the three tests (**Reviewer Fig. 16**). Interestingly, using MAST and LR we found more upregulated “memory” DEGs.

Wilcoxon; Extended Data Fig. 1 e,f

MAST

LR

Reviewer Figure 16: a, Bar plot depicting proportion of downregulated and b, upregulated genes in highly abundant cell types that retain a transcriptional profile from T0 to T1 or change their trajectory toward the profile

of lean subjects at T1 of cell populations using Wilcoxon test for identification of DEGs (This is also in Extended Data Fig. 1). **c**, Bar plot depicting proportion of downregulated and **d**, upregulated genes in highly abundant cell types that retain a transcriptional profile from T0 to T1 or change their trajectory toward the profile of lean subjects at T1 of cell populations using MAST test for identification of DEGs. **e**, Bar plot depicting proportion of downregulated and **f**, upregulated genes in highly abundant cell types that retain a transcriptional profile from T0 to T1 or change their trajectory toward the profile of lean subjects at T1 of cell populations using likelihood-ratio for identification of DEGs.

Additionally, we compared the “memory” DEGs we obtained using all three tests and found that most “memory” DEGs from the Wilcoxon analysis are also found by the MAST and/or LR analysis. For adipocytes ~94% of upregulated and ~91% of downregulated Wilcoxon “memory” DEGs are found by all three tests (**Reviewer Fig. 17**).

Reviewer Figure 17: Overlap of downregulated (left) and upregulated (right) “memory” DEGs of NEFA adipocytes identified using MAST, Wilcoxon or LR.

Based on these findings and a recently published benchmarking paper comparing different testing methods integrated in the FindMarkers function of Seurat by Pullin and McCarthy (PMID: 38409056) we have decided to use the Wilcoxon test for our DE analysis as it was (slightly) more stringent and its results are corroborated by the other tests. We have, however, highlighted in the method section that we used MAST and LR to confirm our results.

7. Cell-cell interaction analysis – Standard statistical measures used to “score for significance” of interactions are based on permutation tests as outline in the CellPhoneDB method. Yet, as these are hard to implement correctly, and the predicted interactions are hard to validate - an alternative is to apply an orthogonal method, such as NichNet algorithm that also scores interactions based on signaling pathways (or the CellPhoneDB algorithm that integrates statistical analysis). The overlap in the predictions is expected to be partial, given the approach and assumptions made by each algorithm, but could help increase the confidence and point out the most robust interactions and signaling pathways.

We thank the referee for their suggestion and agree that using alternative methods to analyze cell-cell interactions of snRNA data is very useful to corroborate findings of one such tool. As suggested, we have applied CellChat and CellPhoneDB to the pooled omAT MTSS snRNA-seq dataset to analyze which pathways/interactions are altered during obesity (T0 vs Lean) and after weight loss (T1 vs lean). Thereafter we intersected these results to obtain persistently

altered cell-cell interaction pathways. NicheNet needs to pre-define the microenvironment (niche) in which a cell or group of cells functions. Since we do not have a pre-defined niche to assess in context of weight loss at the cellular level, we did not perform the NicheNet analysis for our comparison.

We do find persistent alterations with both CellChat and CellPhoneDB (**Reviewer Tables 7-8**). As already indicated by the referee, the tools are based on different approaches and assumptions and the predictions can differ. With CellPhone DB we only obtain two persistently altered interactions/affected pathways related to SEMA3A_PlexinA signaling. Using CellChat, however, we can find significant overlapping pathways related to TGFb, collagen or adiponectin (**Reviewer Table 8**), which are in line with our GSEA results (**Reviewer Fig. 5/new Extended Data Fig. 5**).

Having expanded the cell-cell interaction analyses and found that the different tools generate not well overlapping results, we decided to not include any results from these cell-cell interaction analyses in our manuscript. We have observed inter-individual differences in cellular composition of AT samples in all studies. The cell-cell interaction analyses should therefore be conducted per individual which is, unfortunately, not possible in our datasets. While we believe that our global analysis presented below reflects the phenotype, the number of cells (nuclei) in each cluster per individual is too small to perform meaningful cell-cell interaction analyses with any tool.

As a consequence, we have also decided to remove the results from the cell-cell interaction analyses in mice (former **Extended Data Fig. 4d**) and the associated text (Lines 203-205) from the manuscript. We had displayed number of interactions and interaction strength between major cell types in mouse epiAT across conditions. We had used CellChat and could not do statistical tests like CellPhoneDB to show if the observed differences were significant. Given that depending on the tool used for such analyses results can be different, and, indeed, the mere number/strength of interactions are not on their own informative, we think that this plot no longer adds value to our manuscript.

Future studies should analyse and validate cell-cell interactions in the context of weight loss using these and other tools.

Reviewer Table 7: CellPhone DB results of significant overlapping interactions and pathways retained from T0 to T1 compared against lean control samples

Significant overlapping interactions	Res_t0 vs lean (Source: Target)	Res_t1 vs lean (Source: Target)
SEMA3A_PlexinA2_complex1	Adipo: EndoCs	APCs: Adipo
SEMA6A_PlexinA2_complex1	Adipo: EndoCs	APCs: Adipo

Reviewer Table 8: CellChat results of significant overlapping interactions and pathways retained from T0 to T1 compared against lean control samples

Significant overlapping pathways	Res_t0 vs lean (Source: Target)	Res_t1 vs lean (Source: Target)
TGFb	DCs: Adipo	DCs: Adipo
FGF	Adipo: NeurCs	Adipo: NeurCs

PDGF	Adipo:NeurCs	Adipo:NeurCs
ADIPONECTIN	Adipo:DCs	Adipo:DCs
ANGPTL	Adipo: Macro / APCs	Adipo: Macro / APCs
LAMININ	APCs / EndoCs / MesoCs / Macro / Tcells : Adipo	APCs / EndoCs / MesoCs / Macro / Tcells : Adipo
THBS	APCs:Adipo	APCs:Adipo
VISFATIN	MesoCs:Adipo	MesoCs:Adipo
COLLAGEN	Adipo: EndoCs / LECs / Peri / SMCs / NeurCs	Adipo: EndoCs / LECs / Peri / SMCs / NeurCs
NOTCH	Peri:Adipo	Peri:Adipo
PTPRM	Adipo: EndoCs / LECs / Peri / NeurCs	Adipo: EndoCs / LECs / Peri / NeurCs

8. The following statement is not clear and not supported by the referred figure panels: “By SNP-based demultiplexing we found evidence of inter-individual variations, possibly also affected by sampling during surgery (Extended Data Fig. 2c,f).”

a. Is this genetic variation?

b. How do you see the effect of the surgery?

We thank the referee for their comment. We have now included the bar-plots (included in answer to your question #3) as new **Extended Data Fig. 3**. We do observe differences in cellular composition between individuals of the same condition and therefore think that our statement is valid. We do not have genetic information for these individuals and cannot exclude possible genetic variation amongst our donors as an explanation for this. While all individuals of one study underwent the same kind of surgery it is possible that specifically for omental AT the exact anatomic location of sampling during surgery could affect the cellular composition. For example, a sample taken closer to a blood vessel might present with more endothelial cells. However, we must concede that the word “evidence” is not used properly here. We have revised this sentence in the manuscript:

Lines 104-106: “..., **we observed inter-individual variations after SNP-based demultiplexing, possibly also affected by sampling during surgery**”

9. Many figures are lacking details: please make sure all figures and legends are well annotated and provide enough information to interpret.

Specifically:

a. Extended Data Figure 1 – cell types names are not clear, and missing also in the legend. For example: MesoCs.

Following the *Nature* guidelines, we have now included explanations of all cell type abbreviations in the figure legends.

b. Extended Data Figure 1g – Missing legend and labels on the figures – what are D0-D7? Which samples are lean/obese? What does the color scale represent (scaled number of DEs?).

We have added the labelling of the colour scale. D means donor, we have added this information to the legend. All donors are obese and number of DEGs per donor compared to control samples at T0 is depicted. We have adapted the figure legend and figure to enhance clarity on this in **Extended Data Fig. 1g-k**.

c. Extended Data Figure 1k – missing x-axis labels.

We have added the label to the colour scale. For this plot there is no x-axis label because the coloured legend represents which studies/tissues are displayed. We have made sure that all other x/y-axes in the figures are labelled.

d. Extended Data Fig. 3i: Y-axis label – percent of what?

e. Extended Data Fig. 3i: Can't see the difference between CC_i and CCC or C and CC_s – both are purple/dashed lines that are overlapping.

We have changed the label to “Frequency distribution (%)” for all these plots and adapted the colour shading for **Extended Data Fig. 6i**.

f. Extended Data Fig. 4d: dots overlap text, make it impossible to read.

As this figure is no longer part of the manuscript, we have not included the adjusted plot.

g. Extended Data Fig. 4b: The figure was clearly modified as circles are elongated while the legend was not – i.e. the dot scales are meaningless.

We apologize for this. During reformatting the figure was indeed resized. We have corrected this.

h. Figure 4f-i: grey labels are impossible to read.

We have adjusted all alluvial plots (**Fig. 4f-i, 5d,e**) in the manuscript for more clarity. An example is shown below.

i. Figure 6n: the top figure seems to be cropped in a way that chopped the numbers and the bottom title overlaps the graph.

Indeed, the commas of the genomic location were cropped and we have corrected this.

j. Thorough out all figure - color scales are missing a title. For example - Extended Data Fig. 5: I assume you are displaying scaled average expression for 5a, and -log-p-value for 5b-e.

We have added labels to all heat maps in the manuscript. In **Extended Data Fig. 8a** (formerly 5) the number of DEGs is displayed and not scaled average expression. Further, in **Extended Data Fig. 8b-e** the scale represents an Enrichment Score derived from GSEA analysis. This information is also in the figure legends.

Extended Data Fig. 8:

k. Beyond these clarifications a technical/design point: Figures panels are not aligned and panels and not well spaced within the figure.

l. Of note: I did not point out EVERY correction- but I hope you can generalize from these comments.

We have checked and aligned the figures to improve the readability. We expect that during the final editorial process the figures will have to be further edited.

10. Clarity of the text: Cell type names and conditions are not intuitive and challenges the reader (in text, legends and figures) – why not use the full words or the well know cell type names – in some cases this does not even add characters (if this was the consideration):

a. MesoCS = mesothelial

- b. T-cells, B-cells = vascular (why not lymphoid or T/B-cells)
- c. neuronal like cells - NeurCs1

Following the *Nature* guidelines, we have now included explanations of all abbreviations in the figure legends and in the text itself. To keep the figures concise, we prefer to abbreviate cell type names.

d. In the mice models: HC, CC_s, HHC, CCC – some are not even defined in the main text properly. I’m not even sure I understood correctly the annotations of: C, CC, CCC, and CC_s for example. As previously noted by one of the other reviewers.

We understand that the nomenclature may initially not be intuitive and we would like to clarify our nomenclature using the information also included in the main text and the graphical representation used in **Figure 2a**:

“6-week-old male mice were fed a high-fat diet (HFD; 60% kcal from fat) or low-fat chow diet (10% kcal from fat) for 12 (H and C) or 25 weeks (HH and CC_1). Subsequently, we switched the diet to a standard chow diet (HC, CC_s, HHC, CCC)...”

Figure 2a:

C means chow diet/low fat feeding, while H signifies a high fat diet (HFD) feeding period. H followed by C means high fat diet feeding followed by chow diet feeding.

Minor comments:

11. *Multi-Omics Factor Analysis (MOFA) – is not a well-known term and should be clarified in the main text. Specifically, what are you predicting, and what do the factors represent?*

We agree with the referee that MOFA is not (yet) widely used in the adipose tissue research community. The tool MOFA can be used to identify sources of biological variability in multimodal datasets but is not a predictive tool.

To further clarify, we have made the following changes to the text:

Lines 257-261: “Next, to identify sources of biological variability (factors) in our data sets based on all modalities across all conditions we employed Multi-Omics Factor Analysis (MOFA)^{69,70} using our paired epigenetic data, enabling unsupervised integration and

clustering of paired multi-omic (Fig. 4d) datasets to overcome potential limitations of modality-specific analyses.

1. Supplementary tables with samples, differential genes, and differential pathways are missing.

We will include DEG lists as Source Data. They are also already accessible on GitHub for the referee (https://github.com/vonMeyennLab/AT_memory). Further, we will provide the GSEA results associated to figures as Source Data.

2. Reviewer Fig. 6 and 7 are interesting and could be added to the supplement.

We thank the referee for their suggestion. Nonetheless, we have decided to not include immune cell subtyping in our manuscript to retain the main focus of the manuscript on adipocytes and not distract the reader and also due to the technical limitations of snRNA-seq in assigning immune cell subtypes with high confidence.

3. Orthogonal information related to proteins or metabolites is missing in the manuscript, to show the transcriptional/epigenetic changes translate to potentially functional differences.

We thank the referee for their comment. We agree that information on proteomic or metabolomic changes in the context of weight loss would be very interesting and highly relevant, but the generation and analysis of such (human) datasets are beyond the scope of this manuscript should be addressed in subsequent follow-up studies.

Reviewer Reports on the Second Revision:

Referees' comments:

Referee #5 (Remarks to the Author):

The authors have made substantial efforts to address all of my comments, and largely most of the comments are addressed, specifically, the added statistical tests increase the confidence in the reported results. I have some comments that were not sufficiently addressed:

1. Report of the number of samples is still misleading for the mouse samples – while you pooled a significant number of mice you in fact have a single pool of mice per condition (i.e. a single sample). Yet this is not reflected in your report of the number of samples in the main text and legend. For example – you write “n=5 each” or “n = 6 HH and CC_I, n = 10 HHC and CCC “, which suggests that you have 6-10 samples in each condition, while you have a single sample (pooled), and thus it should be reflected, for example: “a single sample per condition from n=5 pooled mice”. This is important as the number of samples affects the robustness and confidence in the reported results - a single samples lacks statistical power to determine if the observed trends emerge due to a single mice or multiple mice, biological differences or technical noise in snRNA-seq.

2. In line with the comment above - unfortunately, the mouse pools (single samples) affect your ability to determine (and test statistically) changes in cell frequency as well as in differentially expressed genes. Fluctuations in cell numbers and library quality and converge can vary significantly, and this especially affects the less abundant cell types (e.g. LAMs) lowly expressed genes, where small technical fluctuations might seem like a large difference. Thus, you need to tone down the text throughout all the mouse sections and in the legends, and adjust the figures to clearly reflect that these results are predictions, and without additional validations they lack the confidence necessary. Specifically, regarding the LAMs- I agree that the myeloid cells observations regarding the LAMs is well documented – but never the less next to the mouse claims you should add a disclaimer stating that while you are limited by the number of samples to define a statistically significant change, you observed a change which is consistent with previous published results.

3. This was a minor comment in my previous report but it was not addressed and it affects the interoperability of the data in Figure 3 and Figure 6 so I raise it again: A stacked bar plot is a misleading way to present changes in cell types (which is the purpose of the figure in these cases), as you can't track and compare changes across all cell types. To be clearer - see for examples Figure 3C – it is hard to test if there are any changes in NPVMs as each bar starts at a different position – if they were aligned next to each other or presented in some other way, such as a heatmap of frequencies, you could directly compare across conditions. When you have more cell types as in Figure 3B or Figure 6j – this is even more apparent that the graph is not informative, and does not enable to see which cell types are changing.

4. Lines 104-106: “we observed inter-individual variations after SNP-based demultiplexing” – I assume you mean “we observed inter-individual variations in cellular compositions after SNP-based demultiplexing

Author Rebuttals to Second Revision:

Rebuttal letter to the manuscript 2023-01-01106C: “Adipose tissue retains an epigenetic memory of obesity that persists after weight loss”

Referee #5 (Remarks to the Author):

The authors have made substantial efforts to address all of my comments, and largely most of the comments are addressed, specifically, the added statistical tests increase the confidence in the reported results.

We thank the referee for reviewing our rebuttal and acknowledging that we have addressed their comments.

I have some comments that were not sufficiently addressed:

1. Report of the number of samples is still misleading for the mouse samples – while you pooled a significant number of mice you in fact have a single pool of mice per condition (I.e. a single sample). Yet this is not reflected in your report of the number of samples in the main text and legend. For example – you write “n=5 each” or “n = 6 HH and CC_1, n = 10 HHC and CCC “, which suggests that you have 6-10 samples in each condition, while you have a single sample (pooled), and thus it should be reflected, for example: “a single sample per condition from n=5 pooled mice”. This is important as the number of samples affects the robustness and confidence in the reported results - a single samples lacks statistical power to determine if the observed trends emerge due to a single mice or multiple mice, biological differences or technical noise in snRNA-seq.

We have included this in the figure legends and in the text regarding the snRNAseq data. For physiological tests, mice were not pooled.

2. In line with the comment above - unfortunately, the mouse pools (single samples) affect your ability to determine (and test statistically) changes in cell frequency as well as in differentially expressed genes. Fluctuations in cell numbers and library quality and converge can vary significantly, and this especially affects the less abundant cell types (e.g. LAMs) lowly expressed genes, where small technical fluctuations might seem like a large difference. Thus, you need to tone down the text throughout all the mouse sections and in the legends, and adjust the figures to clearly reflect that these results are predictions, and without additional validations they lack the confidence necessary. Specifically, regarding the LAMs- I agree that the myeloid cells observations regarding the LAMs is well documented – but never the less next to the mouse claims you should add a disclaimer stating that while you are limited by the number of samples to define a statistically significant change, you observed a change which is consistent with previous published results.

We agree with the referee that we cannot perform statistical tests for relative cell type abundance in our mouse snRNAseq data. Where necessary, we have included statements that our observations are consistent with previous published results. Of note, we did not assess nor state significance in these results, since – as also mentioned by the referee – these analyses in a pooled sample would not be valid.

3. This was a minor comment in my previous report but it was not addressed and it affects the interoperability of the data in Figure 3 and Figure 6 so I raise it again: A stacked bar plot is a misleading way to present changes in cell types (which is the purpose of the figure in these cases), as you can't track and compare changes across all cell types. To be clearer - see for

examples Figure 3C – it is hard to test if there are any changes in NPVMs as each bar starts at a different position – if they were aligned next to each other or presented in some other way, such as a heatmap of frequencies, you could directly compare across conditions. When you have more cell types as in Figure 3B or Figure 6j – this is even more apparent that the graph is not informative, and does not enable to see which cell types are changing.

We respectfully disagree with the referee. In our view, the representation as a heatmap would not improve the comparability across conditions. Based on the feedback we received so far, we think that stacked barplots are not difficult to interpret or misleading and do well represent these (minor) findings of the manuscript

4. Lines 104-106: “we observed inter-individual variations after SNP-based demultiplexing” – I assume you mean “we observed inter-individual variations in cellular compositions after SNP-based demultiplexing

We thank the referee for noticing and have corrected this mistake in the revised manuscript.